# ENERGY-BASED MODEL TRAINING OBJECTIVE ROBUST TO INACCURATE SGLD SAMPLES

## ABSTRACT

We propose a novel technique for training Energy-based Models (EBMs), which are neural network-based models capable of modeling complex probability distributions. The standard approach to EBM training relies on samples generated from the modeled distribution using Stochastic Gradient Langevin Dynamics (SGLD). However, this training method is known to be unstable, as SGLD may fail to provide reliable samples. Compared to other popular generative models, EBMs can directly evaluate unnormalized log-likelihoods for input observations. Unfortunately, trained EBMs typically fail to robustly estimate the likelihoods for distant input observations, as the training procedure only considers the gradients of the log-likelihood with respect to the observations and not the actual log-likelihood values. This paper proposes a generalization of the standard training objective that addresses both issues. The proposed objective explicitly incorporates estimated unscaled log-likelihoods, allowing the EBM to estimate the likelihoods more reliably. Notably, EBMs do not need to (and as we point out, cannot) correctly estimate log-likelihoods to be effective for sampling using the non-convergent SGLD procedure. The proposed objective is controlled by a single hyper-parameter, which balances the trade-off between the quality of the estimated log-likelihoods and the generated samples. A specific setting of this parameter recovers the standard EBM training objective. Moreover, the proposed objective enhances robustness to unreliable SGLD samples by de-weighting contributions from samples that appear inconsistent with the modeled distribution, i.e., samples with very low estimated likelihoods compared to other generated samples or real training data. We demonstrate the improvement in log-likelihood modeling on toy datasets and enhanced stability in a real data scenario, where this stability leads to better performance.

## 1 INTRODUCTION

Unrestricted probabilistic Energy-based Models (EBMs) (Du & Mordatch, 2019; Nijkamp et al., 2019; Xie et al., 2016) are powerful generative models. Theoretically, trained EBMs can generate new data following the Markov Chain Monte Carlo (MCMC) iterative procedure. However, two closely related approaches, score-based generative models (Hyvärinen & Dayan, 2005; Vincent, 2011; Song & Ermon, 2019; 2020; Song et al., 2021) and diffusion models (Sohl-Dickstein et al., 2015; Ho et al., 2020), have shown superior quality in the tasks of generating new data, typically shown on image datasets. Nevertheless, EBMs stand out as they can directly provide unnormalized likelihood values for each input, unlike the alternative approaches. The promise of having reliable unscaled likelihood values could give rise to new approaches requiring them (Du et al., 2023) or improve the performance of existing ones that rely on likelihood from alternative generative models (Kingma & Welling, 2013; Dinh et al., 2016; Kingma & Dhariwal, 2018). The most widely used approach of training EBMs via Maximum Likelihood Estimation (MLE) is based on generated approximate samples from the EBM that are treated as true samples. This EBM training is unstable in most cases, which is considered as currently one of the biggest issues related to EBMs (Grathwohl et al., 2019). An expert can remedy some cases by fine-tuning all the necessary hyperparameters, but usually at the cost of restricting the architecture or causing a potential drop in performance. This makes it not just time-consuming but also impractical as the whole process might need to be repeated when a change in data, model architecture, or hyperparameters is required, let alone joint training, where EBM would represent only a part of the whole system. Unfortunately, except for the stability issues, trained EBMs typically suffer from poor estimates of unscaled likelihood values when comparing the

likelihoods of two distant input observations. Instead, what is considered informative and utilized after training are the local changes in log-likelihood values, also known as score $\nabla_{\mathbf{x}} \log p_{\boldsymbol{\theta}}(\mathbf{x})$.

The standard approach to training EBMs does not directly utilize the actual likelihood values; instead, it relies solely on the score used during the generation of model samples required for training. Our method is motivated by the fact that in real scenarios, only approximate and often unreliable model samples will be generated. To better handle these cases, we propose a generalized training approach (loss) that has a single hyperparameter $\beta$, whose specific setting $\beta = 0$ recovers the standard training. All other settings result in training that explicitly requires values of unscaled log-likelihoods, so log-likelihoods directly play a role in the optimization process. The parameter $\beta$ controls the dynamic range of these weights. We can trade off sample quality for a more credible estimate of likelihood values and improved training stability based on the setting of $\beta$. We demonstrate this effect on the toy dataset. Additionally, we show increased training stability on a real dataset; however, we were unable to train models that could benefit from better-estimated likelihoods because the utilized SGLD sampler was ineffective.

## 2 ENERGY-BASED MODELS

An Energy-Based model (EBM) is a generative model capable of representing complex probability distributions. However, there is no straightforward method for sampling from this distribution. Moreover, given an input observation $\mathbf{x}$ and model parameters $\boldsymbol{\theta}$, typically represented by a neural network, we can only evaluate $p_{\boldsymbol{\theta}}(\mathbf{x})$ up to an unknown normalization constant. For continuous $\mathbf{x}$, the probability distribution is given by

$$p_{\boldsymbol{\theta}}(\mathbf{x}) = \frac{e^{-E_{\boldsymbol{\theta}}(\mathbf{x})}}{Z_{\boldsymbol{\theta}}} = \frac{e^{f_{\boldsymbol{\theta}}(\mathbf{x})}}{Z_{\boldsymbol{\theta}}}. \tag{1}$$

The energy[1] function $E_{\boldsymbol{\theta}}(\mathbf{x})$ assigns a score to each continuous input observation $\mathbf{x} \in \mathbb{R}^{D_x}$. To ensure that $p_{\boldsymbol{\theta}}(\mathbf{x})$ is a properly normalized distribution, the partition function is defined as $Z_{\boldsymbol{\theta}} = \int_{\mathbf{x}} e^{f_{\boldsymbol{\theta}}(\mathbf{x})} d\mathbf{x}$. The gradient of the objective function, which is the expected log-likelihood of training data distribution $p_d(\mathbf{x})$, can be expressed as

$$\nabla_{\boldsymbol{\theta}} \mathbb{E}_{p_d(\mathbf{x})} \left[ \log p_{\boldsymbol{\theta}}(\mathbf{x}) \right] = \nabla_{\boldsymbol{\theta}} \mathbb{E}_{p_d(\mathbf{x})} \left[ f_{\boldsymbol{\theta}}(\mathbf{x}) - \log Z_{\boldsymbol{\theta}} \right] = \mathbb{E}_{p_d(\mathbf{x})} \left[ \nabla_{\boldsymbol{\theta}} f_{\boldsymbol{\theta}}(\mathbf{x}) \right] - \mathbb{E}_{p_{\boldsymbol{\theta}}(\mathbf{x})} \left[ \nabla_{\boldsymbol{\theta}} f_{\boldsymbol{\theta}}(\mathbf{x}) \right]. \tag{2}$$

Minimizing $\nabla_{\boldsymbol{\theta}} \log Z_{\boldsymbol{\theta}} = \mathbb{E}_{p_{\boldsymbol{\theta}}(\mathbf{x})} \left[ \nabla_{\boldsymbol{\theta}} f_{\boldsymbol{\theta}}(\mathbf{x}) \right]$ directly is intractable. An overview of existing approaches for EBMs training is presented in Song & Kingma (2021). This work focuses on a common strategy that addresses the optimization by approximating $\mathbb{E}_{p_{\boldsymbol{\theta}}(\mathbf{x})}$ via sampling. The method first draws $N$ positive samples $\mathbf{x}_i^+ \sim p_d(\mathbf{x})$ and $M$ negative samples $\mathbf{x}_j^- \sim p_{\boldsymbol{\theta}}(\mathbf{x})$ to approximate the computation in Equation 2 by

$$\nabla_{\boldsymbol{\theta}} \mathbb{E}_{p_d(\mathbf{x})} \left[ \log p_{\boldsymbol{\theta}}(\mathbf{x}) \right] \approx \frac{1}{N} \sum_i^N \nabla_{\boldsymbol{\theta}} f_{\boldsymbol{\theta}}(\mathbf{x}_i^+) - \frac{1}{M} \sum_j^M \nabla_{\boldsymbol{\theta}} f_{\boldsymbol{\theta}}(\mathbf{x}_j^-). \tag{3}$$

### 2.1 STOCHASTIC GRADIENT LANGEVIN DYNAMICS

Since $f_{\boldsymbol{\theta}}$ is represented by an unrestricted neural network, there is no direct way to obtain negative samples $\mathbf{x}_j^- \sim p_{\boldsymbol{\theta}}(\mathbf{x})$. Typically, time-consuming Markov chain Monte Carlo (MCMC) sampling methods must be employed. Specifically, Stochastic Gradient Langevin Dynamics (SGLD) (Welling & Teh, 2011) begins with a sample from the initial distribution[2] $\mathbf{x}^0 \sim p(\mathbf{x}^0)$ and aims to generate samples from $p_{\boldsymbol{\theta}}(\mathbf{x})$ through an iterative procedure

$$\mathbf{x}^t = \mathbf{x}^{t-1} + \frac{\alpha^t}{2} \nabla_{\mathbf{x}} f_{\boldsymbol{\theta}}(\mathbf{x}^{t-1}) + \mathbf{u}^t, \qquad u_i^t \sim \mathcal{N}(0, \alpha^t), \ \ 1 \le i \le D_x, \tag{4}$$

where $t$ denotes the time step. In theory, if the conditions $\sum_t \alpha^t = \infty$ and $\sum_t (\alpha^t)^2 < \infty$ hold, it is guaranteed that $\mathbf{x}^t$ will become a sample from $p_{\boldsymbol{\theta}}(\mathbf{x})$ as $t \to \infty$. In practice, we resort to a modified version of SGLD by significantly limiting the number of time steps and hand-tuning the schedule for the step size $\alpha$, which is typically set to a convenient value and kept fixed for all time steps.

---

[1]We will refer to the negative energy $f_{\boldsymbol{\theta}}(\mathbf{x})$ instead of energy $E_{\boldsymbol{\theta}}(\mathbf{x})$ for brevity.

[2]The initial distribution is typically uniform or Gaussian.

The number of steps needed to generate a reasonable sample can be optionally reduced by carefully choosing the initial sample $\mathbf{x}^0$. One commonly used technique builds upon Persistent Contrastive Divergence (PCD) (Tieleman, 2008) by maintaining a buffer of previously generated samples, from which the initial samples are drawn (Du & Mordatch, 2019).

SGLD suffers from known issues, such as a slow mixing rate, the need for a suitable step size, and problematic sampling from areas of constant likelihood values. We describe these in more detail in Appendix A. Similarly, Hamiltonian Monte Carlo (HMC), an alternative MCMC approach to sample from $p_{\boldsymbol{\theta}}(\mathbf{x})$, can result in inaccurate samples.

## 2.2 Source of Potentially Unreliable Likelihood Values

The described training procedure does not directly incorporate likelihood or log-likelihood values, as they are absent in Equation 3, and the negative examples[3] are generated solely based on the score $\nabla_{\mathbf{x}} f_{\boldsymbol{\theta}}(\mathbf{x})$ (Equation 4). In theory, likelihoods do not need to be evaluated to train the model because it is assumed that we have access to true samples from $p_{\boldsymbol{\theta}}(\mathbf{x})$. Nijkamp et al. (2019) demonstrated that even when negative examples are not true samples from $p_{\boldsymbol{\theta}}(\mathbf{x})$, the training can still converge. In such cases, we can generate samples distributed similarly to $p_d(\mathbf{x})$, but the learned distribution $p_{\boldsymbol{\theta}}(\mathbf{x})$ may differ significantly from $p_d(\mathbf{x})$. We can interpret SGLD as a procedure that transforms the initial distribution $p(\mathbf{x}^0)$ into the distribution $\text{SGLD}(p_{\boldsymbol{\theta}}(\mathbf{x}))^4$, using the gradient of the negative energy of $p_{\boldsymbol{\theta}}(\mathbf{x})$, thereby generating samples from this distribution. If the SGLD procedure converges, then $\text{SGLD}(p_{\boldsymbol{\theta}}(\mathbf{x}))$ will be equivalent to $p_{\boldsymbol{\theta}}(\mathbf{x})$. However, when the initial distribution is far from $p_{\boldsymbol{\theta}}(\mathbf{x})$, using a limited number of SGLD steps may not produce samples from $p_{\boldsymbol{\theta}}(\mathbf{x})$ utilizing the gradient of its true negative energy. Nonetheless, a different EBM might exist, potentially far from the desired $p_{\boldsymbol{\theta}}(\mathbf{x})$, that models $p_d(\mathbf{x})$ Its negative energy gradient in the non-convergent SGLD procedure can transform the initial distribution into $p_d(\mathbf{x})$. This scenario is illustrated in the first three columns of Figure 2, which show the true distribution $p_d(\mathbf{x})$, the incorrectly learned distribution $p_{\boldsymbol{\theta}}(\mathbf{x})$, and the distribution of generated examples $\text{SGLD}(p_{\boldsymbol{\theta}}(\mathbf{x}))$. If this occurs, the training converges as $p_d(\mathbf{x}) = \text{SGLD}(p_{\boldsymbol{\theta}}(\mathbf{x}))$. While this approach offers the advantage of constructing an implicit generative model (sampler) of $p_d(\mathbf{x})$, we must avoid relying on unnormalized likelihood values, as $f_{\boldsymbol{\theta}}(\mathbf{x})$ may correspond to a completely different EBM. Additionally, the ability to generate realistic data is sensitive to the specific setting of hyperparameters for the SGLD procedure. Modifications such as increasing the number of steps can significantly degrade the sample quality.

## 2.3 Force Analogy of EBM Training

We liken EBM training via Equation 3 to shaping the surface of the probability density function by balancing two forces as it can provide an intuitive explanation and motivation for our technique. The first (positive) force increases the log-likelihood (via $f_{\boldsymbol{\theta}}(\mathbf{x})$) at the locations of training data $\mathbf{x} \sim p_d(\mathbf{x})$, as illustrated in Figure 1a. The locations of the second (negative) force, which decreases the log-likelihood values, should be determined by $p_{\boldsymbol{\theta}}(\mathbf{x})$ (Figure 1b). The force strength emitted by all positive and negative examples is equivalent. When negative examples are true samples from $p_{\boldsymbol{\theta}}(\mathbf{x})$, both positive and negative forces will be balanced everywhere as the training progresses, and $p_{\boldsymbol{\theta}}(\mathbf{x}) \to p_d(\mathbf{x})$. However, from a force balancing point of view, using the same approach when an excessive number of negative examples originate from regions of low likelihood, positive examples keep increasing $f_{\boldsymbol{\theta}}(\mathbf{x})$, but negative examples decrease $f_{\boldsymbol{\theta}}(\mathbf{x})$ at different locations. This discrepancy can lead to an unconstrained increase or decrease of $f_{\boldsymbol{\theta}}(\mathbf{x})$, possibly causing training divergence. Our approach tries to address this issue by mitigating the damage caused by negative examples with low likelihood values.

## 3 Generalized Training Approach for Energy-based Models

The negative force depicted in Figure 1b can be alternatively obtained by sampling from the uniform distribution $u(\mathbf{x})$ while adjusting the force strength to match $p_{\boldsymbol{\theta}}(\mathbf{x})$ (Figure 1c). However, this approach is impractical for real-world applications involving high-dimensional spaces with low-dimensional support, as it necessitates an extremely large number of samples to locate some with nonnegligible likelihood values. The cases shown in Figure 1b and Figure 1c represent two extremes

---

[3]We further use the terms positive and negative examples instead of samples to highlight that negative examples might not be true samples from the desired distribution.

[4]We denote the distribution of SGLD samples obtained using the negative energy of $p_{\boldsymbol{\theta}}(\mathbf{x})$ as $\text{SGLD}(p_{\boldsymbol{\theta}}(\mathbf{x}))$.

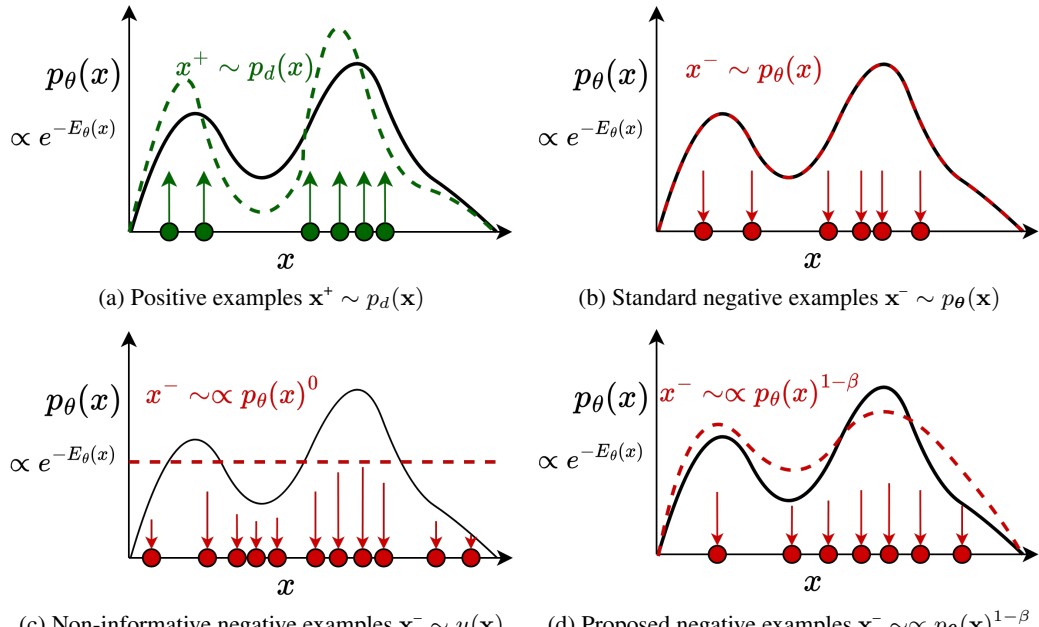

Figure 1: The difference between the standard and proposed EBM training. The standard EBM training is illustrated as forces of (a) positive and (b) negative examples pushing the value of log-likelihood against each other, with each example exerting the same force. The size of the arrows depicts the force acting on the log-likelihood. In the proposed method, we replace the negative force in (b) with (d), which samples $\mathbf{x}^- \sim\propto p_{\boldsymbol{\theta}}(\mathbf{x})^{1-\beta}$, while assigning the force proportional to $p_{\boldsymbol{\theta}}(\mathbf{x})^{\beta}$. Cases (b) and (c) are special instances of (d) for $\beta = 0$ and $\beta = 1$.

of a broader scheme we propose. In essence, we derive negative examples by sampling from a distribution that is more uncertain than $p_{\boldsymbol{\theta}}(\mathbf{x})$ and adjust the force strength accordingly, as illustrated in Figure 1d. Sampling from a more uncertain distribution can enhance the mixing efficiency of the SGLD procedure. However, the primary motivation for this method is that while the SGLD procedure may, and indeed does in practice, produce biased samples, relative adjustments to the force strength across negative examples can be calculated precisely. This enables soft rejection of inaccurate negative examples with low likelihood values.

We propose a new training approach that is a generalization of the standard SGLD-based approach described in Section 2. We want to split the responsibility for the quality of negative examples from possibly biased sampling into the processes of sampling and subsequent reweighting, which we hope to reduce the overall bias. We achieve this by first expressing $p_{\boldsymbol{\theta}}(\mathbf{x})$ via two different distributions $q_{\boldsymbol{\theta}}(\mathbf{x})$ and $r_{\boldsymbol{\theta}}(\mathbf{x})$ as $p_{\boldsymbol{\theta}}(\mathbf{x}) \propto q_{\boldsymbol{\theta}}(\mathbf{x})r_{\boldsymbol{\theta}}(\mathbf{x})$ and leveraging self-normalized importance sampling (SNIS) (Bishop, 2007; Owen, 2013). Applying SNIS to $\mathbb{E}_{p_{\boldsymbol{\theta}}(\mathbf{x})}\left[\nabla_{\boldsymbol{\theta}} f_{\boldsymbol{\theta}}(\mathbf{x})\right]$ in Equation 2, using samples $\mathbf{x}_{\boldsymbol{j}}^- \sim q_{\boldsymbol{\theta}}(\mathbf{x}), 1 \le j \le M$, we have

$$\mathbb{E}_{p_{\boldsymbol{\theta}}(\mathbf{x})}\left[\nabla_{\boldsymbol{\theta}} f_{\boldsymbol{\theta}}(\mathbf{x})\right] = \frac{\mathbb{E}_{q_{\boldsymbol{\theta}}(\mathbf{x})}\left[\tilde{w}(\mathbf{x})\nabla_{\boldsymbol{\theta}} f_{\boldsymbol{\theta}}(\mathbf{x})\right]}{\mathbb{E}_{q_{\boldsymbol{\theta}}(\mathbf{x})}\left[\tilde{w}(\mathbf{x})\right]} \approx \frac{\sum_{j}^{M} \tilde{w}(\mathbf{x}_{\boldsymbol{j}}^-)\nabla_{\boldsymbol{\theta}} f_{\boldsymbol{\theta}}(\mathbf{x}_{\boldsymbol{j}}^-)}{\sum_{j}^{M} \tilde{w}(\mathbf{x}_{\boldsymbol{j}}^-)} = \sum_{j}^{M} w(\mathbf{x}_{\boldsymbol{j}}^-)\nabla_{\boldsymbol{\theta}} f_{\boldsymbol{\theta}}(\mathbf{x}_{\boldsymbol{j}}^-).$$

(5)

The relative (unnormalized) weight is determined as $\tilde{w}(\mathbf{x}) \propto \frac{p_{\boldsymbol{\theta}}(\mathbf{x})}{q_{\boldsymbol{\theta}}(\mathbf{x})} \propto r_{\boldsymbol{\theta}}(\mathbf{x})$, and the absolute (normalized) weight as $w(\mathbf{x}^-) = \frac{\tilde{w}(\mathbf{x}^-)}{\sum_{j}^{M} \tilde{w}(\mathbf{x}_{\boldsymbol{j}}^-)}$. By plugging Equation 5 into Equation 2, we replace Equation 3 with a more general form

$$\nabla_{\boldsymbol{\theta}}\mathbb{E}_{p_d(\mathbf{x})}\left[\log p_{\boldsymbol{\theta}}(\mathbf{x})\right] \approx \frac{1}{N}\sum_{i}^{N}\nabla_{\boldsymbol{\theta}} f_{\boldsymbol{\theta}}(\mathbf{x}_{\boldsymbol{i}}^+) - \sum_{j}^{M} w(\mathbf{x}_{\boldsymbol{j}}^-)\nabla_{\boldsymbol{\theta}} f_{\boldsymbol{\theta}}(\mathbf{x}_{\boldsymbol{j}}^-).$$

(6)

Equation 6 holds in general for any $p_{\boldsymbol{\theta}}(\mathbf{x})$, $q_{\boldsymbol{\theta}}(\mathbf{x})$ and $r_{\boldsymbol{\theta}}(\mathbf{x})$, but in this work, we consider the specific factorization $p_{\boldsymbol{\theta}}(\mathbf{x}) = p_{\boldsymbol{\theta}}(\mathbf{x})^{1-\beta} p_{\boldsymbol{\theta}}(\mathbf{x})^{\beta}$. That corresponds to EBMs $q_{\boldsymbol{\theta}}(\mathbf{x}) \propto e^{(1-\beta)f_{\boldsymbol{\theta}}(\mathbf{x})}$ and $r_{\boldsymbol{\theta}}(\mathbf{x}) \propto e^{\beta f_{\boldsymbol{\theta}}(\mathbf{x})}$. When $\beta$ is set between 0 and 1, the inverse temperature $\beta$ can be interpreted as a proportion of responsibility assigned to reweighting. With the considered factorization, negative examples are drawn as $\mathbf{x}_j^- \sim\propto e^{(1-\beta)f_{\boldsymbol{\theta}}(\mathbf{x}_j^-)}$, and Equation 6 can be expressed as

$$\nabla_{\boldsymbol{\theta}}\mathbb{E}_{p_d(\mathbf{x})}\left[\log p_{\boldsymbol{\theta}}(\mathbf{x})\right] \approx \frac{1}{N}\sum_i^N \nabla_{\boldsymbol{\theta}}f_{\boldsymbol{\theta}}(\mathbf{x}_i^+) - \frac{\sum_j^M e^{\beta f_{\boldsymbol{\theta}}(\mathbf{x}_j^-)}\nabla_{\boldsymbol{\theta}}f_{\boldsymbol{\theta}}(\mathbf{x}_j^-)}{\sum_j^M e^{\beta f_{\boldsymbol{\theta}}(\mathbf{x}_j^-)}}. \tag{7}$$

Equation 3 is a special case of Equation 7 when $\beta = 0$, for which $\tilde{w}(\mathbf{x}^-)$ is proportional to the uniform distribution $u(\mathbf{x})$. Consequently, $w(\mathbf{x}^-) = 1/M$, which is independent of $\mathbf{x}$, unlike in any other setting, where $\beta \neq 0$. Training the EBM using Equation 7 corresponds to the visualization in Figure 1d through force balancing.

### 3.1 Including Positive Example as an Extra Negative Example

When equipped with a true sampler of $p_{\boldsymbol{\theta}}(\mathbf{x})^{1-\beta}$, using $\beta = 0$ is the most rational choice, as increasing $\beta$ only reduces the effective sample size. However, in the context of a biased sampler, the proposed method offers two key benefits. First, negative examples' unnormalized probabilities (evaluated with the inverse temperature $\beta$) are directly integrated into the loss calculation, ensuring that these probabilities can no longer be ignored during optimization. Second, as a consequence, the contribution from negative examples that are out of distribution (OOD) relative to $p_{\boldsymbol{\theta}}(\mathbf{x})$ is diminished, based on the comparison to the other negative examples, which behaves as a soft filtering of untrustworthy negative examples. Nevertheless, its efficacy relies on the presence of at least one negative example that is a true sample or has a likelihood comparable to true samples. When all negative examples are OOD, we would like to filter them all, i.e. set $w(\mathbf{x}^-) \doteq 0$ for all $\mathbf{x}^-$. However, this is impossible as the proposed method is constrained by $\sum_M w(\mathbf{x}^-) = 1$.

Because we train $p_{\boldsymbol{\theta}}(\mathbf{x})$ to approximate $p_d(\mathbf{x})$, it makes positive examples $\mathbf{x}^+ \sim p_d(\mathbf{x})$ suitable candidates to serve as approximate negative examples $\mathbf{x}^- \propto p_{\boldsymbol{\theta}}(\mathbf{x})^{1-\beta}$, with a high likelihood value. Using it as an additional sample in the calculation of the denominator in Equation 5, $\mathbb{E}_{q_{\boldsymbol{\theta}}(\mathbf{x})}[\tilde{w}(\mathbf{x})]$, would allow us to have $w(\mathbf{x}^-) \doteq 0$ for all $\mathbf{x}^-$. At this point, the contributions from all positive ($\sum_i 1/N = 1$) and all negative ($\sum_M w(\mathbf{x}^-) = 1$) examples are balanced. We want to retain this balance to avoid further training instabilities related to the force-balancing analogy of EBM training. However, leveraging positive examples to improve the estimate of the denominator in Equation 5 without affecting its numerator would break this balance. As a result, while we would prevent the likelihood of negative examples from an unconstrained decrease, the likelihood of positive examples would remain unrestricted in its potential growth. Instead, we propose to include positive examples into negative ones, which maintains the balance, as the positive examples used as negative ones are effectively assigned negative weights. We propose a specific method of incorporating positive examples among negative ones. First, we interpret Equation 6 as a contribution from $N$ positive examples, while each positive example has $M$ shared negative examples. With this interpretation, we then include one additional negative example, which is different for each $\mathbf{x}_i^+$, and this additional negative example is $\mathbf{x}_i^+$ itself. As a result of this construction, Equation 6 is replaced with

$$\frac{1}{N}\sum_i^N (1 - \overset{+}{w}(\mathbf{x}_i^+))\left(\nabla_{\boldsymbol{\theta}}f_{\boldsymbol{\theta}}(\mathbf{x}_i^+) - \sum_{j=1}^M w(\mathbf{x}_j^-)\nabla_{\boldsymbol{\theta}}f_{\boldsymbol{\theta}}(\mathbf{x}_j^-)\right), \tag{8}$$

where $\overset{+}{w}(\mathbf{x}^+) = \frac{\tilde{w}(\mathbf{x}^+)}{\tilde{w}(\mathbf{x}^+) + \sum_j^M \tilde{w}(\mathbf{x}_j^-)}$ corresponds to the absolute weight that the positive example would get assigned if it was part of the negative examples. We can see that compared to Equation 6, the mere change is that the individual contributions from these positive examples are rescaled by a factor of $1 - \overset{+}{w}(\mathbf{x}_i^+)$. The effect of this factor is discussed in the next section, and we provide more details and steps to derive Equation 8 from Equation 6 in Appendix B.

### 3.2 Analysis

The particular choice of including positive examples among negative ones, which resulted in Equation 8, might seem arbitrary. However, this decision is motivated by the fact that Equation 8 can

alternatively be viewed as a form of discriminative training. Further insight into this interpretation is provided in Appendix C. Unlike Equation 6, which can replace Equation 2 without affecting the optimized objective, using Equation 8 instead is not equivalent. Importantly, the resulting objective being maximized is a lower bound of the objective that was optimized before including this additional negative example. The proof is provided in Appendix D. We also study the effect of including a positive example among negative ones from a mechanistic point of view. This reveals two effects. First, it rescales the gradient for each batch, depending on the difference between the negative energy of positive and negative examples. As $f_{\boldsymbol{\theta}}(\mathbf{x}^+) - f_{\boldsymbol{\theta}}(\mathbf{x}^-)$ increases, the scale of the gradient becomes smaller, allowing for the possibility of ignoring all negative examples from the batch that do not have competitive likelihoods. Similar to a rescaling of gradient contributions among negative examples introduced by $\beta$, the second effect resulting from replacing Equation 6 with Equation 8 leads the rescaling of the gradient contributions among positive examples. This rescaling effectively slows down the growth of the log-likelihood value for positive examples that have relatively high likelihoods among other positive examples. This effect is magnified as the performance of the SGLD sampler decreases, i.e., when $f_{\boldsymbol{\theta}}(\mathbf{x}^+) - f_{\boldsymbol{\theta}}(\mathbf{x}^-)$ increases. Derivations and more details are provided in Appendix E. We expect these effects to positively influence training. However, as mentioned, they result in a biased objective that is a lower bound of the original objective, which may be undesirable in certain cases. To address this, it is possible to isolate the effect of rescaling the overall gradient in each batch based on how inaccurate the negative examples are, resulting in the optimization of the original unbiased objective. These variants also serve as ablations, and all details and derivations are described in Appendix E.1.

Our proposed modification of EBM training involves two main components. First, we introduced the $\beta$-parameterized objective in Section 3. Second, we incorporated positive examples among negative ones in Section 3.1. Isolating the effect of including a positive example without introducing $\beta$ may seem intriguing; however, this approach is not useful since it corresponds to $\beta = 0$. In this case, including a positive example results in only a slight modification to the learning rate, as demonstrated in Appendix F. Section 3 discusses sampling based on negative energy $1 - \beta$ and subsequent reweighting according to $\beta$. Notably, these two values do not necessarily need to sum up to 1; arbitrary values can be employed instead, corresponding to a different parameterization of the EBM. We derive this in Appendix G, based on an analysis of key aspects of the practical SGLD sampler, including its extension to $\beta \neq 0$ settings.

Section 2.2 explains that when $\beta = 0$, the learned EBM $p_{\boldsymbol{\theta}}(\mathbf{x})$ might be significantly different from $p_d(\mathbf{x})$. With some simplifications, we expect that if training converges for $\beta \neq 0$, $p_{\boldsymbol{\theta}}(\mathbf{x})$ must partially reflect $p_d(\mathbf{x})$, but at the same time, it must compensate for the differences between $\mathrm{SGLD}(p_{\boldsymbol{\theta}}(\mathbf{x}))$ and $p_d(\mathbf{x})$. As a result, for values of $\mathbf{x}$ where $\mathrm{SGLD}(p_{\boldsymbol{\theta}}(\mathbf{x})) > p_d(\mathbf{x})$, we expect $p_{\boldsymbol{\theta}}(\mathbf{x}) < p_d(\mathbf{x})$. We explain our reasoning in Appendix I and discuss additional trade-offs related to the different settings of $\beta$ in Appendix I.1.

If, during many subsequent updates, we fail to generate any reasonable negative examples, the proposed method will not assist with training stabilization when combined with optimizers that apply an adaptive learning rate. We provide more details in Appendix H. Although encountering this failure state necessitates modifying the approach for generating negative examples, we aim to raise awareness about why we cannot prevent instability in this case.

### 3.3 Efficient Implementation

With modern libraries that support automatic differentiation, such as PyTorch (Paszke et al., 2019) and TensorFlow (Abadi et al., 2016), the proposed method can be implemented in a compact form while ensuring efficient computation, with negligible overhead compared to the case when $\beta = 0$. Algorithm 1 demonstrates the difference in the loss calculation in pseudocode using vectors $\mathbf{fp}$ and $\mathbf{fn}$, where $\mathbf{fp}_i = f_{\boldsymbol{\theta}}(\mathbf{x}_i^+)$ and $\mathbf{fn}_j = f_{\boldsymbol{\theta}}(\mathbf{x}_j^-)$. We define methods in their typical sense: $\mathbf{u}.\mathrm{mean}()$ calculates the mean of the vector $\mathbf{u}$; the operation $\mathrm{stack}(\mathbf{u}, \mathbf{v})$ vertically stacks the row vectors $\mathbf{u}$ and $\mathbf{v}$ into a matrix; $\mathbf{u}.\mathrm{LSE}()$ computes $\log \sum e^{\mathbf{u}}$ over the first dimension of the tensor, reducing its dimensionality by one; and $b.\mathrm{expand}(k)$ creates a vector with $k$ elements, where each element is $b$. The justification for using this computation is most apparent from LHS of Equation 22, which is equivalent to Equation 8. Aside from the loss calculation, the only difference is that the SGLD procedure runs with negative energy multiplied by $(1 - \beta)$, making it extremely easy to integrate into existing systems.

---

**Algorithm 1** Loss Calculation in Pseudocode

---

**Input:** $\mathbf{fp} \in \mathbb{R}^N$, $\mathbf{fn} \in \mathbb{R}^M$, $\beta$
1: **if** $\beta = 0$ **then**
2:      $L \leftarrow \mathbf{fn}.\operatorname{mean}() - \mathbf{fp}.\operatorname{mean}()$
3: **else**
4:      $L \leftarrow \operatorname{stack}(\mathbf{fp} * \beta, (\mathbf{fn} * \beta).\operatorname{LSE}().\operatorname{expand}(N)).\operatorname{LSE}().\operatorname{mean}()/\beta - \mathbf{fp}.\operatorname{mean}()$
5: **end if**

---

## 4 EXPERIMENTS

### 4.1 EVALUATION ON TOY DATASETS

To verify whether the expected behavior holds in practice, we focus on experiments using toy datasets, where we can visually compare the learned densities. Specifically, we adopt the 2D toy dataset setup from Nijkamp et al. (2020), which initializes the SGLD procedure with a Gaussian distribution and employs Adam (Kingma & Ba, 2014) as the optimizer. For evaluation, we consider the standard circles dataset and modify the standard Gaussian Mixture Model (GMM) dataset by assigning different weights to each component. This modification is useful for examining the effects of the proposed method. We follow the non-convergent setup described in Nijkamp et al. (2020), where only a limited number of SGLD steps with a constant step size are used to generate negative samples, leading to incorrectly learned energy functions. To expedite the experiments, we reduced the number of SGLD steps from 100 to 20, compensating it with a larger SGLD step size and, in the case of the GMM dataset, by using an adapted learning rate schedule. The rest of the setup, including the architecture, remains unchanged.

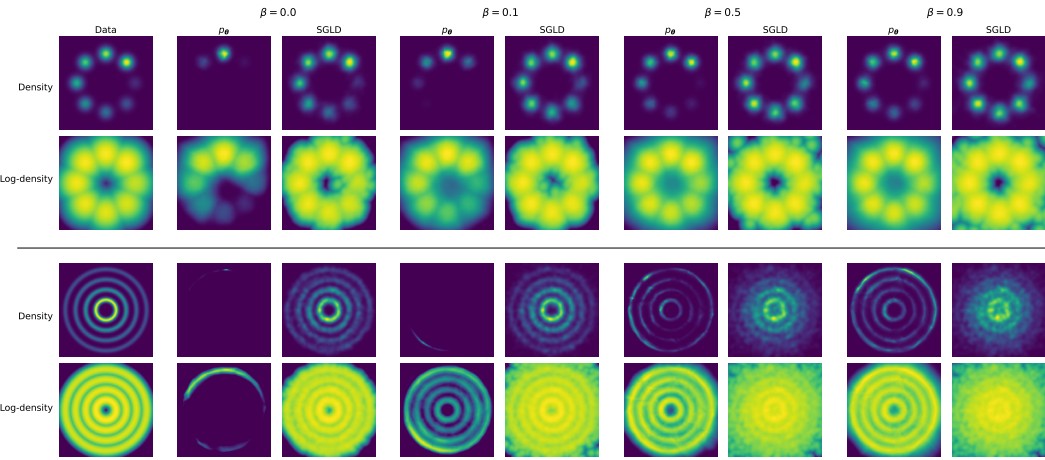

Figure 2: The default setup comparing different values of $\beta$.

We present the results for the default experimental setup in Figure 2. The first two rows correspond to the GMM datasets, while the last two rows relate to the circles dataset. Odd rows visualize the density, whereas even rows visualize the log of density. The first column displays the true density of the data distribution $p_d(\mathbf{x})$. Each subsequent pair of columns corresponds to different $\beta$ settings, showing the learned density $p_\theta(\mathbf{x})$ followed by $\operatorname{SGLD}(p_\theta(\mathbf{x}))^5$. $\operatorname{SGLD}(p_\theta(\mathbf{x}))$ is obtained using the code published by Nijkamp et al. (2020) as the kernel density estimate of the samples generated by the same non-convergent SGLD used during training. With $\beta = 0$, the modeled distribution $p_\theta(\mathbf{x})$ (and visually even its logarithm) significantly deviates from the true data distribution. However, as also observed in Nijkamp et al. (2020), when the same non-convergent SGLD is used for both model training and generation of samples from the trained model, the distribution of the generated samples $\operatorname{SGLD}(p_\theta(\mathbf{x}))$ closely resembles the training data. Even a relatively small value of $\beta = 0.1$ results in the modeled distribution being much closer to the training data distribution compared to $\beta = 0$. Conversely, applying the non-convergent SGLD procedure to the correctly learned density does not

---

[5]It differs from the distribution of negative examples $\operatorname{SGLD}(p_\theta(\mathbf{x})^{1-\beta})$ used during training.

yield true samples, as demonstrated by the GMM dataset with $\beta = 0.9$. However, if we were to sample from this distribution using (convergent) SGLD with a large number of steps, the samples would eventually follow the correctly learned density $p_{\boldsymbol{\theta}}(\mathbf{x})$. These results confirm that we can trade off sample quality for the quality of the learned distribution $p_{\boldsymbol{\theta}}(\mathbf{x})$ by adjusting $\beta$.

We provide additional experiments in Appendix K.1, demonstrating that although the quality of the learned distribution improves with increasing $\beta$, in the case of the non-convergent SGLD sampler, the learned distribution must compensate for this bias. This confirms the expected behavior based on the theoretical analysis discussed in Appendix I. Furthermore, in Appendix K.2, we compare the performance of different variants related to how positive examples are incorporated into negative ones. The results suggest that our default variant, corresponding to the computation in Equation 8, should be preferred over alternatives utilizing Equation 7, Equation 33, and Equation 34. Consequently, in the rest of this work, we will consider only the default variant.

## 4.2 Energy-based Model Applied on Real Dataset

To determine how the improved performance in learning distributions with increasing $\beta$ transfers to real data problems, we extended the setup of Nijkamp et al. (2019) to incorporate the proposed generalized training method and performed experiments on the CIFAR-10 (Krizhevsky et al., 2009) dataset with various settings of $\beta$. Our goal was to examine the influence of $\beta$ on training while exploring a range similar to that in the previous section, i.e., $0.1 < \beta < 0.9$. However, we were unable to train the models with these settings. When $\beta = 0$, it is known that non-convergent SGLD, with a limited number of steps, might not be a good sampler of $p_{\boldsymbol{\theta}}(\mathbf{x})$, but it still provides negative examples that resemble positive examples. The distribution of energies for positive and negative examples is typically similar throughout training, making both negative and positive examples of comparable quality in terms of likelihood under $p_{\boldsymbol{\theta}}(\mathbf{x})$[6]. As $\beta$ increases, the distribution of negative examples changes due to the replacement of $\mathrm{SGLD}(p_{\boldsymbol{\theta}}(\mathbf{x}))$ with $\mathrm{SGLD}(p_{\boldsymbol{\theta}}(\mathbf{x})^{1-\beta})$. However, the difference between samples from $\mathrm{SGLD}(p_{\boldsymbol{\theta}}(\mathbf{x}))$ and $\mathrm{SGLD}(p_{\boldsymbol{\theta}}(\mathbf{x})^{0.9})$ when using the same pseudorandom seed is very small, meaning that the difference in sampling during training for $\beta = 0$ and $\beta = 0.1$ is minimal. Noticeable differences arise for larger values of $\beta$. When we attempted to train the system with $\beta = 0.1$, the gap between the likelihood values of positive and negative examples became substantial after only a few parameter updates. According to true EBM training, these negative examples should not be used. Our approach successfully diminishes the weight of these examples. However, this is not a temporary problem. The non-convergent SGLD systematically fails to provide any negative examples with likelihood values comparable to those of the positive examples. Since we rely on informative negative examples, the training cannot progress. As a result, we are restricted to very small values of $\beta$. Since the SGLD procedures are effectively equivalent in these cases, sampling from more uncertain distributions cannot be the cause of the differences we observe. We provide more details and visualizations in Appendix L.

Our experiments suggest that the practical non-convergent version of SGLD is not an effective sampler for EBM in general, and we elaborate on this further in Section 5. We also tried a few simple modifications to the SGLD sampler in hopes of improving its performance; however, these attempts did not yield satisfactory results, and the development of a more sophisticated approach is beyond the scope of this work. The motivation for our method is to improve the learned density and avoid training instabilities. However, the setup from Nijkamp et al. (2019) does not suffer from training divergence, and we are unable to train models with values of $\beta$ that may positively affect the model's likelihood values. To address this, we shift our attention to a different setup for which training instabilities occur.

## 4.3 Stabilized Training of Joint Energy-based Models

In the experiments conducted on the toy dataset reported in Appendix K, we introduced two setups where $\beta = 0$ led to training divergence, whereas $\beta \neq 0$ did not. To investigate whether increasing $\beta$ can stabilize EBM training on real data, we utilized the Joint Energy-based Model (JEM) (Grathwohl et al., 2019). JEM integrates an EBM with a classifier into a single model and is known for its instability issues. Grathwohl et al. (2019) reported having no solution to training this model directly, as the training would continuously diverge. The only viable option was to repeatedly load the model from the last saved checkpoint before the divergence occurred and change the random seed until the

---

[6]Even though true samples may be quite different.

desired model was obtained. Even the default setup of publicly available code provided by Grathwohl et al. (2019) fails to finish due to training instabilities. We extended this code with the proposed method and performed experiments. Due to space limitations, the detailed description of JEM and the full extent of the experiments are reported in Appendix M, while this section only briefly summarizes them.

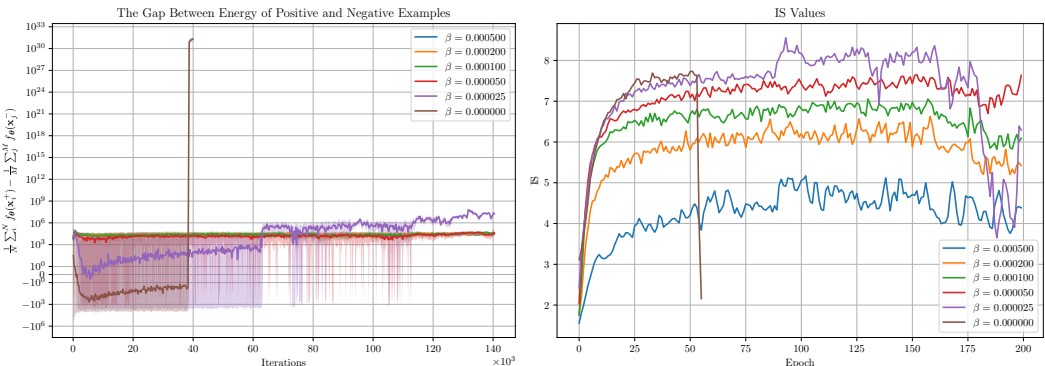

Figure 3: Evolution of Inception Score (IS) and the difference in average negative energy between positive and negative examples during JEM training for various $\beta$.

Similarly to Section 4.2, we are limited to very small values of $\beta$ due to the performance of the SGLD sampler. We present the evolution of the difference between $f_{\boldsymbol{\theta}}(\mathbf{x}^+)$ and $f_{\boldsymbol{\theta}}(\mathbf{x}^-)$, along with the Inception Score (IS) (Salimans et al., 2016), under different $\beta$ settings in Figure 3. While JEM training with $\beta = 0$ diverges around epoch 55, we can postpone or even eliminate these training instabilities by increasing $\beta$. This adjustment allows us to achieve superior performance in terms of classification accuracy, IS, and Fréchet Inception Distance (FID) (Heusel et al., 2017) compared to what was possible with $\beta = 0$, as shown in Table 1. Negative examples used during training for $\beta = 0.000025$ are depicted in Figure 4. When further modifying the default hyperparameters related to the generative component of JEM, we observe that training with $\beta = 0$ can result in divergence even in the first few epochs of training, while increasing $\beta$ prevents it. More details are provided in Appendix M.1. While we demonstrated that the proposed method can achieve better performance, we believe that the ability to stabilize the training process is far more valuable. In practice, managing training instabilities can be more discouraging than experiencing a slight performance degradation.

Table 1: Comparison of classification accuracy on the CIFAR-10 test set, Inception Score (IS), and Fréchet Inception Distance (FID) for various $\beta$ values using the default JEM setup.

| $\beta$ | Accuracy↑ [%] | IS↑ | FID↓ |
|---|---|---|---|
| 0.000000 | 90.5 | 7.7 | 39.6 |
| 0.000025 | 91.2 | **8.6** | **38.9** |
| 0.000050 | 91.2 | 7.7 | 44.1 |
| 0.000100 | 91.6 | 7.1 | 50.9 |
| 0.000200 | **91.7** | 6.6 | 59.6 |
| 0.000500 | 91.5 | 5.2 | 85.3 |

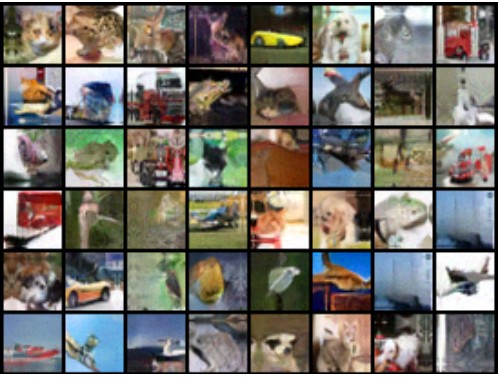

Figure 4: Negative examples generated during epoch 107 of JEM training ($\beta = 2.5 \times 10^{-5}$).

## 5 DISCUSSION AND CONCLUSION

We proposed a new training approach, parameterized by $\beta$, for training Energy-based Models using negative examples generated by SGLD. This method generalizes the standard approach, which corresponds to setting $\beta = 0$. This work focuses on establishing a theoretical foundation for

the proposed approach and includes a non-trivial analysis. Due to space limitations, most of the theoretical details are presented in the Appendix. The motivation behind this method is to mitigate the negative influence of inaccurate negative examples that are not true samples from $p_{\boldsymbol{\theta}}(\mathbf{x})$, which should, in turn, improve likelihoods and reduce training instabilities. However, this approach may lead to a decrease in the quality of generated examples when using a practical non-convergent SGLD sampler. Fortunately, we can balance these effects by tuning $\beta$. Our method extends standard EBM training by effectively deweighting unreliable training samples as analyzed in Appendix E. This effect can be split into three components: the weight that rescales the contribution of negative examples, preventing $f_{\boldsymbol{\theta}}(\mathbf{x}^-)$ from shrinking to large negative values; the weight that adjusts the contribution of positive examples, preventing $f_{\boldsymbol{\theta}}(\mathbf{x}^+)$ from growing excessively; and the weight that scales the overall contribution of a single batch based on the energy difference between positive and negative examples, reducing the importance of batches with a significant gap between $f_{\boldsymbol{\theta}}(\mathbf{x}^+)$ and $f_{\boldsymbol{\theta}}(\mathbf{x}^-)$.

On a toy dataset, we confirmed our theoretical hypothesis regarding the behavior of $\beta$ and demonstrated that the learned density improves as $\beta$ increases. However, with the real dataset, we were unable to fully realize this potential, successfully training models only with extremely small values of $\beta$, such us $\beta = 10-6$. This limitation arose because, with small values of $\beta$ such as $\beta = 0.01$, the SGLD sampler struggled to produce negative examples $\mathbf{x}^-$ with energies comparable to those of positive examples. In contrast, when $\beta = 0$, both negative and positive examples exhibited comparable energies; however, this gap gradually widened as $\beta$ increased. To explain this behavior, we propose a hypothesis in Appendix J, suggesting that training with $\beta = 0$ follows a more general approach of "attractor-repeller training". Fortunately, we demonstrated that we can mitigate training instabilities by using very small values of $\beta$. This was illustrated in the training of JEM, where setting $\beta = 0$ can lead to divergence, which may occur even within the first few epochs.

Even though training stabilization is an important aspect, we believe that improving likelihood could potentially be even more valuable. Unfortunately, achieving this requires a different sampling method that produces negative examples with higher $f_{\boldsymbol{\theta}}(\mathbf{x}^-)$, enabling the training of models with values in the range $0.1 < \beta < 0.5$. It is unclear whether existing approaches could be adapted for this purpose, whether a new method needs to be developed, or if the issue could be resolved through alternative strategies, such as better initialization of the SGLD or a specialized architecture.

Our method is applicable in any context where SGLD samples are used to train EBMs with the standard loss function ($\beta = 0$). This integration is straightforward and can potentially enhance performance or address stability issues. However, since this work primarily focuses on building a theoretical foundation, our method introduces additional possibilities. By efficiently filtering out negative examples with low $f_{\boldsymbol{\theta}}(\mathbf{x}^-)$, we shift from the theoretical requirement that all negative examples must be valid to a more flexible condition—that at least one negative example is good. This flexibility allows for the combination of multiple techniques for generating negative examples, such as using both SGLD and HMC or deploying multiple SGLD samplers with varying hyperparameters.

Furthermore, in the case of JEM, we demonstrated that the proposed method could overcome phases of training where most, if not all, generated examples were unreliable. This flexibility is a powerful tool, as it enables adaptive HMC or SGLD sampling that can optimize parameters during training, which has been reported to be challenging for $\beta = 0$ due to instabilities limiting exploration. Additionally, the introduced weights can also serve as effective monitoring tools for EBM training.

Our approach can also help assess how much a system relies on the introduced attractor-repeller scheme by comparing systems trained with $\beta = 0$ to those trained with, for example, $\beta = 0.01$. In this work, we addressed the issue of samplers frequently producing samples with low $f_{\boldsymbol{\theta}}(\mathbf{x})$. Similarly, if a sampler tends to produce samples with excessively high $f_{\boldsymbol{\theta}}(\mathbf{x}^-)$ values (e.g., sampling from $\propto p_{\boldsymbol{\theta}}(\mathbf{x}^-)^4$), the proposed approach could be adapted by using negative values of $\beta$. Lastly, all values of $\beta$ correspond to EBM training, allowing for dynamic adjustments of $\beta$ during training or the simultaneous optimization of objectives with varying $\beta$ values.

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

## A SGLD ISSUES

### A.1 AREAS OF CONSTANT OR UNDEFINED LIKELIHOOD

SGLD (MCMC chain) can travel through areas of constant likelihood only based on diffusion, i.e. Gaussian noise added at each step. It requires a large step size or a large number of steps to escape these areas. Practical applications with a limited number of SGLD steps will likely result in an SGLD sample originating in this area, which artificially increases the probability of producing samples from these areas, therefore, resulting in sampling from a modified distribution.

According to the manifold hypothesis, in many real applications, the support of the data distribution forms a low-dimensional manifold in the observed space. In other words, the true data distribution $p_d(\mathbf{x}) = 0$ for most $\mathbf{x}$. It can be interpreted as the score $\nabla_{\mathbf{x}} f_{\boldsymbol{\theta}}(\mathbf{x})$ being either 0 or undefined. The training objective minimizes KLD between $p_d(\mathbf{x})$ and $p_{\boldsymbol{\theta}}(\mathbf{x})$. When $p_{\boldsymbol{\theta}}(\mathbf{x})$ starts converging to $p_d(\mathbf{x})$, the importance of this flaw increases. Convolving the data distribution with a Gaussian filter (adding Gaussian noise to each $\mathbf{x}$) can alleviate it, but it would require a large amount of noise to remove that issue entirely. This is not suitable as the resulting distribution would be very noisy.

### A.2 MIXING RATE

SGLD has a slow mixing rate. Once the MCMC chain imposed by SGLD enters a particular mode of multimodal distribution, it will likely stay in that mode when only a limited number of steps with suitable step size is performed. Within a single distribution mode, Langevin Dynamics are not

affected by the weight (probability mass) of that mode, and it is believed that SGLD might provide genuine samples from the underlying distribution mode. However, in a real scenario, it results in the probability of obtaining a sample from each mode m not being affected by the weight of each mode, i.e. the mode mixture weight $p_{\boldsymbol{\theta}}(\mathrm{m}) = \int_{\mathbf{x}} p_{\boldsymbol{\theta}}(\mathbf{x}, \mathrm{m})$. Instead, we believe that the weight will roughly correspond to a probability of the MCMC chain being initialized in that mode $p_0(\mathrm{m}) = \int_{\mathbf{x}^0} p(\mathbf{x}^0, \mathrm{m})$, where $p(\mathbf{x}^0)$ is the distribution used to initialize Markov chain of SGLD.

### A.3 SGLD STEP SIZE

As training progresses, the optimal SGLD step size can vary. Since we perform only a limited number of steps, typically with a fixed size, too small step sizes will not cause enough movement, while too large step sizes will only cause uninformative jumping over the space under the exploration. Too-large step sizes could be avoided or at least detected by incorporating Metropolis-Hastings acceptance probability, which gives rise to Metropolis-adjusted Langevin algorithm (Besag, 1994), but it is typically not applied during training.

## B DERIVATIONS FOR INCLUDING POSITIVE EXAMPLE AS NEGATIVE ONE

We propose including an extra negative example, which will differ for each positive example. Specifically, we propose that every positive example acts as its negative example with index $M + 1$, i.e. $\mathbf{x}_{M+1}^- = \mathbf{x}_i^+$ and this section describes how to derive Equation 8. We begin by interpreting Equation 6 as a contribution from $N$ parts, given by

$$\frac{1}{N} \sum_i^N \left( \nabla_{\boldsymbol{\theta}} f_{\boldsymbol{\theta}}(\mathbf{x}_i^+) - \frac{\sum_j^M \tilde{w}(\mathbf{x}_j^-) \nabla_{\boldsymbol{\theta}} f_{\boldsymbol{\theta}}(\mathbf{x}_j^-)}{\sum_j^M \tilde{w}(\mathbf{x}_j^-)} \right). \tag{9}$$

Next, we include an extra negative example, specific for each $\mathbf{x}_i^+$, resulting in

$$\frac{1}{N} \sum_i^N \left( \nabla_{\boldsymbol{\theta}} f_{\boldsymbol{\theta}}(\mathbf{x}_i^+) - \frac{\sum_j^{M+1} \tilde{w}(\mathbf{x}_j^-) \nabla_{\boldsymbol{\theta}} f_{\boldsymbol{\theta}}(\mathbf{x}_j^-)}{\sum_j^{M+1} \tilde{w}(\mathbf{x}_j^-)} \right). \tag{10}$$

Substituting that additional negative example $\mathbf{x}_{M+1}^- = \mathbf{x}_i^+$, we obtain

$$\frac{1}{N} \sum_i^N \left( \nabla_{\boldsymbol{\theta}} f_{\boldsymbol{\theta}}(\mathbf{x}_i^+) - \frac{\tilde{w}(\mathbf{x}_i^+) \nabla_{\boldsymbol{\theta}} f_{\boldsymbol{\theta}}(\mathbf{x}_i^+) + \sum_j^M \tilde{w}(\mathbf{x}_j^-) \nabla_{\boldsymbol{\theta}} f_{\boldsymbol{\theta}}(\mathbf{x}_j^-)}{\tilde{w}(\mathbf{x}_i^+) + \sum_j^M \tilde{w}(\mathbf{x}_j^-)} \right). \tag{11}$$

For brevity, we denote $\sum_j^M \tilde{w}(\mathbf{x}_j^-)$ as $s$, then

$$\frac{1}{N} \sum_i^N \frac{1}{\tilde{w}(\mathbf{x}_i^+) + s} \left( (\tilde{w}(\mathbf{x}_i^+) + s) \nabla_{\boldsymbol{\theta}} f_{\boldsymbol{\theta}}(\mathbf{x}_i^+) - \tilde{w}(\mathbf{x}_i^+) \nabla_{\boldsymbol{\theta}} f_{\boldsymbol{\theta}}(\mathbf{x}_i^+) - \sum_j^M \tilde{w}(\mathbf{x}_j^-) \nabla_{\boldsymbol{\theta}} f_{\boldsymbol{\theta}}(\mathbf{x}_j^-) \right). \tag{12}$$

This simplifies to

$$\frac{1}{N} \sum_i^N \frac{1}{\tilde{w}(\mathbf{x}_i^+) + s} \left( s \nabla_{\boldsymbol{\theta}} f_{\boldsymbol{\theta}}(\mathbf{x}_i^+) - \sum_j^M \tilde{w}(\mathbf{x}_j^-) \nabla_{\boldsymbol{\theta}} f_{\boldsymbol{\theta}}(\mathbf{x}_j^-) \right). \tag{13}$$

Substituting $s$ back and using the absolute weight $w(\mathbf{x}^-) = \frac{\tilde{w}(\mathbf{x}^-)}{\sum_j^M \tilde{w}(\mathbf{x}_j^-)}$ instead of the relative weight, we get

$$\frac{1}{N} \sum_i^N \frac{\sum_j^M \tilde{w}(\mathbf{x}_j^-)}{\tilde{w}(\mathbf{x}_i^+) + \sum_j^M \tilde{w}(\mathbf{x}_j^-)} \left( \nabla_{\boldsymbol{\theta}} f_{\boldsymbol{\theta}}(\mathbf{x}_i^+) - \sum_j^M w(\mathbf{x}_j^-) \nabla_{\boldsymbol{\theta}} f_{\boldsymbol{\theta}}(\mathbf{x}_j^-) \right). \tag{14}$$

Finally, plugging in the absolute weight, which positive example would get assigned if it was part of the negative examples $\overset{+}{w}(\mathbf{x}^+) = \frac{\tilde{w}(\mathbf{x}^+)}{\tilde{w}(\mathbf{x}^+) + \sum_j^M \tilde{w}(\mathbf{x}_j^-)}$ recovers Equation 8 as

$$\frac{1}{N} \sum_i^N (1 - \overset{+}{w}(\mathbf{x}_i^+)) \left( \nabla_{\boldsymbol{\theta}} f_{\boldsymbol{\theta}}(\mathbf{x}_i^+) - \sum_{j=1}^M w(\mathbf{x}_j^-) \nabla_{\boldsymbol{\theta}} f_{\boldsymbol{\theta}}(\mathbf{x}_j^-) \right). \tag{15}$$

## C   INCLUDING POSITIVE EXAMPLE RESULTS IN DISCRIMINATIVE TRAINING

We can interpret the proposed training in Equation 8 as a special case of discriminative training and this section aims to provide more details.

### C.1   DISCRIMINATIVE TRAINING

For simplicity, we use the same symbols for parameters, training data, distributions, and inverse temperature for both generative and discriminative models. Training a classifier of $\mathbf{x}$ parameterized by $\boldsymbol{\theta}$ is typically achieved by minimizing expected cross-entropy H between the true posterior distribution $p$ and modeled posterior distribution $p_{\boldsymbol{\theta}}$ over the possible $D_y$ values of label $y$. In practice, we collect tuples $(\mathbf{x^i}, y^i) \in \mathrm{DS}$, where $p(y = y^i \mid \mathbf{x} = \mathbf{x^i}) = 1$ and search for $\boldsymbol{\theta}$ maximizing

$$-\mathbb{E}_{p(\mathbf{x})}\left[H(p, p_{\boldsymbol{\theta}})\right] = -\mathbb{E}_{p(\mathbf{x})}\left[-\sum_{y \in \mathrm{Y}} p(y \mid \mathbf{x}) \log p_{\boldsymbol{\theta}}(y \mid \mathbf{x})\right] \approx \frac{1}{|\mathrm{DS}|} \sum_{i=0}^{|\mathrm{DS}|} \log p_{\boldsymbol{\theta}}(y^i \mid \mathbf{x^i}). \quad (16)$$

It is common to transform $\mathbf{x}$ by a neural neural network $g_{\boldsymbol{\theta}}(\mathbf{x})$ to $D_y$-dimensional vector and model $p_{\boldsymbol{\theta}}(y \mid \mathbf{x})$ via Softmax function $\mathrm{SM}(g_{\boldsymbol{\theta}}(\mathbf{x}); \beta) : \mathbb{R}^{D_y} \to (0, 1)^{D_y}$ as

$$p_{\boldsymbol{\theta}}(y = i \mid \mathbf{x}) = \mathrm{SM}(g_{\boldsymbol{\theta}}(\mathbf{x}); \beta)_i = \frac{e^{\beta g_{\boldsymbol{\theta}}(\mathbf{x})_i}}{\sum_{j=1}^{D_y} e^{\beta g_{\boldsymbol{\theta}}(\mathbf{x})_j}}, \quad (17)$$

where $\sum_i \mathrm{SM}(\cdot)_i = 1$. While the inverse temperature $\beta$ is typically set to 1, we keep it general. The gradient of the maximized objective (Equation 16) for a single tuple $(\mathbf{x}, y = i)$ is

$$\nabla_{\boldsymbol{\theta}} \log p_{\boldsymbol{\theta}}(y = i \mid \mathbf{x}) = \nabla_{\boldsymbol{\theta}} \log \mathrm{SM}(g_{\boldsymbol{\theta}}(\mathbf{x}); \beta)_i = \nabla_{\boldsymbol{\theta}} \beta g_{\boldsymbol{\theta}}(\mathbf{x})_i - \nabla_{\boldsymbol{\theta}} \log \sum_{j=1}^{D_y} e^{\beta g_{\boldsymbol{\theta}}(\mathbf{x})_j}. \quad (18)$$

### C.2   DERIVATION

For EBM training with $\beta \neq 0$ (i.e. excluding the standard approach $\beta = 0$), we have

$$\frac{\sum_j^M e^{\beta f_{\boldsymbol{\theta}}(\mathbf{x_j^-})} \nabla_{\boldsymbol{\theta}} f_{\boldsymbol{\theta}}(\mathbf{x_j^-})}{\sum_j^M \left[e^{\beta f_{\boldsymbol{\theta}}(\mathbf{x_j^-})}\right]} = \frac{1}{\beta} \frac{\sum_j^M \nabla_{\boldsymbol{\theta}} e^{\beta f_{\boldsymbol{\theta}}(\mathbf{x_j^-})}}{\sum_j^M \left[e^{\beta f_{\boldsymbol{\theta}}(\mathbf{x_j^-})}\right]} = \frac{1}{\beta} \nabla_{\boldsymbol{\theta}} \log \sum_j^M e^{\beta f_{\boldsymbol{\theta}}(\mathbf{x_j^-})}. \quad (19)$$

Using this result, Equation 7 can be further reformulated as

$$\frac{1}{N} \sum_i^N \nabla_{\boldsymbol{\theta}} f_{\boldsymbol{\theta}}(\mathbf{x_i^+}) - \frac{1}{\beta} \nabla_{\boldsymbol{\theta}} \log \sum_j^M e^{\beta f_{\boldsymbol{\theta}}(\mathbf{x_j^-})} = \frac{1}{N} \sum_i^N \frac{1}{\beta} \nabla_{\boldsymbol{\theta}} \left(\beta f_{\boldsymbol{\theta}}(\mathbf{x_i^+}) - \log \sum_j^M e^{\beta f_{\boldsymbol{\theta}}(\mathbf{x_j^-})}\right). \quad (20)$$

Considering an extra negative example $\mathbf{x_j^-} = \mathbf{x_i^+}$, Equation 20 becomes

$$\frac{1}{N} \sum_i^N \frac{1}{\beta} \nabla_{\boldsymbol{\theta}} \left(\beta f_{\boldsymbol{\theta}}(\mathbf{x_i^+}) - \log \sum_j^{M+1} e^{\beta f_{\boldsymbol{\theta}}(\mathbf{x_j^-})}\right). \quad (21)$$

We aim to show that the content of the parenthesis in Equation 21 and Equation 18 can be mapped to each other. To establish this mapping, for each $\mathbf{x_i^+}$, we construct $\mathbf{X}^i = [\mathbf{x_i^+}, \mathbf{x_1^-}, \ldots, \mathbf{x_M^-}]$. Instead of having a $g_{\boldsymbol{\theta}}(\mathbf{x})$ that maps $\mathbf{x}$ to a vector with $D_y = M + 1$ dimensions, we interpret $f_{\boldsymbol{\theta}}(\cdot)$ as a vector function mapping $\mathbf{X}^i$ to $M + 1$-dimensional vector, where $y$-th element of this mapping is $f_{\boldsymbol{\theta}}(\mathbf{X}_y^i)$. Then, Equation 21 is equivalent to

$$\frac{1}{N} \sum_i^N \frac{1}{\beta} \left(\nabla_{\boldsymbol{\theta}} \beta f_{\boldsymbol{\theta}}(\mathbf{x_i^+}) - \nabla_{\boldsymbol{\theta}} \log(e^{\beta f_{\boldsymbol{\theta}}(\mathbf{x_i^+})} + \sum_j^M e^{\beta f_{\boldsymbol{\theta}}(\mathbf{x_j^-})})\right) = \frac{1}{N} \sum_i^N \frac{1}{\beta} \nabla_{\boldsymbol{\theta}} \log \mathrm{SM}(f_{\boldsymbol{\theta}}(\mathbf{X}^i); \beta)_0,$$

$$(22)$$

which corresponds to the task of indicating in $\mathbf{X}^i$ where the positive example resides. Denoting the index within $\mathbf{X}^i$ as $y$, we have $\mathrm{SM}(f_{\boldsymbol{\theta}}(\mathbf{X}^i); \beta)_0 = p_{\boldsymbol{\theta}}(y = 0 \mid \mathbf{X}^i)$, revealing how Equation 8 relates to discriminative training.

# D  EFFECT OF INCLUDING POSITIVE EXAMPLE AMONG NEGATIVE ONES ON OBJECTIVE

To understand the implications of including a positive example among negative ones, we first show that setting $\beta \neq 0$ allows for an alternative way of expressing the objective. Based on Equation 19 and Equation 20, the gradient of the proposed objective (Equation 7) can be alternatively expressed as

$$\frac{1}{N} \sum_i^N \left( \nabla_{\boldsymbol{\theta}} f_{\boldsymbol{\theta}}(\mathbf{x}_i^+) - \frac{1}{\beta} \nabla_{\boldsymbol{\theta}} \log \sum_j^M e^{\beta f_{\boldsymbol{\theta}}(\mathbf{x}_j^-)} \right), \tag{23}$$

which is also equivalent to

$$\frac{1}{N} \sum_i^N \frac{1}{\beta} \left( \nabla_{\boldsymbol{\theta}} \log e^{\beta f_{\boldsymbol{\theta}}(\mathbf{x}_i^+)} - \nabla_{\boldsymbol{\theta}} \log \sum_j^M e^{\beta f_{\boldsymbol{\theta}}(\mathbf{x}_j^-)} \right) = \nabla_{\boldsymbol{\theta}} \frac{1}{\beta} \frac{1}{N} \sum_i^N \log \left( \frac{p_{\boldsymbol{\theta}}(\mathbf{x}_i^+)^\beta}{\sum_j^M p_{\boldsymbol{\theta}}(\mathbf{x}_j^-)^\beta} \right). \tag{24}$$

Therefore, we can alternatively express the objective as

$$\frac{1}{\beta} \mathbb{E}_{\mathbf{x}^+ \sim p_d(\mathbf{x})} \left[ \log \left( \frac{p_{\boldsymbol{\theta}}(\mathbf{x}^+)^\beta}{\sum_j^M p_{\boldsymbol{\theta}}(\mathbf{x}_j^-)^\beta} \right) \right]. \tag{25}$$

If $\mathbf{x}^- \sim\propto p_{\boldsymbol{\theta}}(\mathbf{x}^-)^{1-\beta}$, the term $1/M \sum_j^M p_{\boldsymbol{\theta}}(\mathbf{x}_j^-)^\beta \approx\propto \int_{\mathbf{x}} p_{\boldsymbol{\theta}}(\mathbf{x})^{1-\beta} p_{\boldsymbol{\theta}}(\mathbf{x})^\beta = 1$, resulting in $\nabla_{\boldsymbol{\theta}} \log \left( \sum_j^M p_{\boldsymbol{\theta}}(\mathbf{x}_j^-)^\beta \right) \approx 0$. Substituting it into Equation 25 confirms that it reduces to the negative cross-entropy between $p_d(\mathbf{x})$ and $p_{\boldsymbol{\theta}}(\mathbf{x})$, which is the original objective. This confirms the legitimacy of maximizing $\frac{1}{\beta} \mathbb{E}_{\mathbf{x}^+ \sim p_d(\mathbf{x})} \left[ \log \left( \frac{p_{\boldsymbol{\theta}}(\mathbf{x}^+)^\beta}{\sum_j^M p_{\boldsymbol{\theta}}(\mathbf{x}_j^-)^\beta} \right) \right]$, which we use to investigate the impact of introducing an additional negative example.

The addition of an extra negative example can be expressed alternatively as

$$\frac{p_{\boldsymbol{\theta}}(\mathbf{x}^+)^\beta}{\sum_j^{M+1} p_{\boldsymbol{\theta}}(\mathbf{x}_j^-)^\beta} = \frac{p_{\boldsymbol{\theta}}(\mathbf{x}^+)^\beta}{p_{\boldsymbol{\theta}}(\mathbf{x}^+)^\beta + \sum_j^M p_{\boldsymbol{\theta}}(\mathbf{x}_j^-)^\beta} = \frac{1}{1 + \frac{\sum_j^M p_{\boldsymbol{\theta}}(\mathbf{x}_j^-)^\beta}{p_{\boldsymbol{\theta}}(\mathbf{x}^+)^\beta}} = \sigma \left( \log \left( \frac{p_{\boldsymbol{\theta}}(\mathbf{x}^+)^\beta}{\sum_j^M p_{\boldsymbol{\theta}}(\mathbf{x}_j^-)^\beta} \right) \right), \tag{26}$$

where $\sigma(\cdot)$ denotes the logistic sigmoid. Using the RHS of Equation 26 to express the addition of the positive example among negative ones applied to Equation 25 yields

$$\frac{1}{\beta} \mathbb{E}_{p_d(\mathbf{x})} \left[ \log \sigma \left( \log \frac{p_{\boldsymbol{\theta}}(\mathbf{x})^\beta}{\sum_j^M p_{\boldsymbol{\theta}}(\mathbf{x}_j^-)^\beta} \right) \right]. \tag{27}$$

Since $\log(\sigma(\cdot))$ is a concave function, applying Jensen's inequality yields

$$\frac{1}{\beta} \mathbb{E}_{p_d(\mathbf{x})} \left[ \log \sigma \left( \log \frac{p_{\boldsymbol{\theta}}(\mathbf{x})^\beta}{\sum_j^M p_{\boldsymbol{\theta}}(\mathbf{x}_j^-)^\beta} \right) \right] \leq \frac{1}{\beta} \log \sigma \left( \mathbb{E}_{p_d(\mathbf{x})} \left[ \log \frac{p_{\boldsymbol{\theta}}(\mathbf{x})^\beta}{\sum_j^M p_{\boldsymbol{\theta}}(\mathbf{x}_j^-)^\beta} \right] \right). \tag{28}$$

By considering $M+1$ negative examples, we maximize a lower bound of the loss function on the RHS. Since $\log(\sigma(\cdot))$ is also a monotonic function, it follows that if $f_{\boldsymbol{\theta}}(\mathbf{x})$ is powerful enough, then

$$\arg\max_{\boldsymbol{\theta}} \frac{1}{\beta} \log \sigma \left( \mathbb{E}_{p_d(\mathbf{x})} \left[ \log \frac{p_{\boldsymbol{\theta}}(\mathbf{x})^\beta}{\sum_j^M p_{\boldsymbol{\theta}}(\mathbf{x}_j^-)^\beta} \right] \right) = \arg\max_{\boldsymbol{\theta}} \frac{1}{\beta} \mathbb{E}_{p_d(\mathbf{x})} \left[ \log \frac{p_{\boldsymbol{\theta}}(\mathbf{x})^\beta}{\sum_j^M p_{\boldsymbol{\theta}}(\mathbf{x}_j^-)^\beta} \right]. \tag{29}$$

Thus, by including an extra positive example among negative ones, we maximize the lower bound of the objective function with $M$ negative examples.

## E   IS BIASED OBJECTIVE PREFERRED?

We manipulate Equation 8 to isolate 2 effects that were introduced by adding positive examples among negative ones as

$$
\underbrace{\left(\frac{1}{N}\sum_i^N 1 - \overset{+}{w}(\mathbf{x}_i^+)\right)}_{\text{GradScale}}\left(\sum_i^N \underbrace{\left(\frac{1 - \overset{+}{w}(\mathbf{x}_i^+)}{\frac{1}{N}\sum_l^N 1 - \overset{+}{w}(\mathbf{x}_l^+)}\right)}_{v(\mathbf{x}_i^+)}\nabla_{\boldsymbol{\theta}} f_{\boldsymbol{\theta}}(\mathbf{x}^+) - \sum_{j=1}^M w(\mathbf{x}_j^-)\nabla_{\boldsymbol{\theta}} f_{\boldsymbol{\theta}}(\mathbf{x}_j^-)\right). \quad (30)
$$

GradScale rescales the overall gradient computed in each batch, while $v(\mathbf{x}^+)$ can be interpreted as an absolute weight that rescales the gradient of positive examples similarly as $w(\mathbf{x}^-)$ rescaled gradients of negative examples. If $v(\mathbf{x}^+) = 1/N$, then the effect of applying Equation 30 compared to Equation 6 would only be rescaling the overall gradient computed in each batch. The value of GradScale decreases as the difference between $f_{\boldsymbol{\theta}}(\mathbf{x}^+)$ and $f_{\boldsymbol{\theta}}(\mathbf{x}^-)$ increases, which is what we want to achieve. Conversely when $f_{\boldsymbol{\theta}}(\mathbf{x}^+) \approx f_{\boldsymbol{\theta}}(\mathbf{x}^-)$, which can be interpreted as SGLD procedure is producing reliable examples, then GradScale $\approx \frac{M}{M+1}$ and it will have a negligible effect on the optimization process.

Now we investigate the effect of the absolute weight for positive example $v(\mathbf{x}^+)$. Rewriting it as

$$
v(\mathbf{x}_i^+) = \frac{1 - \overset{+}{w}(\mathbf{x}_i^+)}{\frac{1}{N}\sum_l^N 1 - \overset{+}{w}(\mathbf{x}_l^+)} = \frac{\frac{\sum_j^M \tilde{w}(\mathbf{x}_j^-)}{\tilde{w}(\mathbf{x}_i^+)+\sum_j^M \tilde{w}(\mathbf{x}_j^-)}}{\frac{1}{N}\sum_l^N \frac{\sum_j^M \tilde{w}(\mathbf{x}_j^-)}{\tilde{w}(\mathbf{x}_l^+)+\sum_j^M \tilde{w}(\mathbf{x}_j^-)}} = \frac{\frac{1}{\tilde{w}(\mathbf{x}_i^+)+\sum_j^M \tilde{w}(\mathbf{x}_j^-)}}{\frac{1}{N}\sum_l^N \frac{1}{\tilde{w}(\mathbf{x}_l^+)+\sum_j^M \tilde{w}(\mathbf{x}_j^-)}} \quad (31)
$$

demonstrates that the absolute weight for positive example $v(\mathbf{x}^+)$ corresponds to relative (unnormalized) weight for positive example $\tilde{v}(\mathbf{x}^+) = \frac{1}{\tilde{w}(\mathbf{x}^+)+\sum_j^M \tilde{w}(\mathbf{x}_j^-)}$. Comparing the ratio between the relative weights of two positive examples $\tilde{v}(\mathbf{x}_1^+)$ and $\tilde{v}(\mathbf{x}_2^+)$, we get

$$
\frac{\tilde{v}(\mathbf{x}_1^+)}{\tilde{v}(\mathbf{x}_2^+)} = \frac{\tilde{w}(\mathbf{x}_2^+)+\sum_j^M \tilde{w}(\mathbf{x}_j^-)}{\tilde{w}(\mathbf{x}_1^+)+\sum_j^M \tilde{w}(\mathbf{x}_j^-)} = \frac{e^{\beta f_{\boldsymbol{\theta}}(\mathbf{x}_2^+)}+\sum_j^M \tilde{w}(\mathbf{x}_j^-)}{e^{\beta f_{\boldsymbol{\theta}}(\mathbf{x}_1^+)}+\sum_j^M \tilde{w}(\mathbf{x}_j^-)}. \quad (32)
$$

When $f_{\boldsymbol{\theta}}(\mathbf{x}^+) \gg f_{\boldsymbol{\theta}}(\mathbf{x}^-)$, i.e. SGLD procedure is not producing competitive examples, then $\tilde{v}(\mathbf{x}^+) \approx 1/p_{\boldsymbol{\theta}}(\mathbf{x}^+)^\beta$. In the context of force analysis (Figure 1a), this can be interpreted as increasing the log-likelihood values of positive examples with already high likelihoods to a lesser extent compared to those positive examples with low likelihood values. We suggested that when $f_{\boldsymbol{\theta}}(\mathbf{x}^+)$ grows without constraints, it might cause training divergence. Therefore, $\tilde{v}(\mathbf{x}^+)$ helps mitigate this kind of training divergence. As the gap between $f_{\boldsymbol{\theta}}(\mathbf{x}^+)$ and $f_{\boldsymbol{\theta}}(\mathbf{x}^-)$ decreases, the effect of this rescaling decreases. This is the desired behavior because when the quality of the sampler of negative examples improves, the effect of the bias decreases.

### E.1   REMOVING THE BIAS INTRODUCED BY AN EXTRA NEGATIVE EXAMPLE

We incorporated the positive example among the negative ones primarily to prevent unwanted changes in $\boldsymbol{\theta}$ when negative examples are not appropriate, as such updates would lack informativeness. We demonstrated that it is, indeed, happening, by an automatic decrease of the gradient magnitude GradScale defined in Equation 30. At the same time, it introduces the weight $v(\mathbf{x}^+)$ that rescales the gradients within positive examples. It causes paying more attention to $\mathbf{x}^+$ with a smaller $p_{\boldsymbol{\theta}}(\mathbf{x}^+)$. As discussed in Appendix D, this results in a biased objective. This section describes the way to keep the gradient rescaling of each batch similarly to GradScale, but remove its dependency on positive examples, which eliminates the bias. This results in the introduction of two additional variants, which can be used for an ablation study.

Within each batch, we can correct the bias by making $\overset{+}{w}(\mathbf{x}_i^+)$ independent of $i$ through averaging as $\frac{1}{N}\sum_i^N \overset{+}{w}(\mathbf{x}_i^+)$. This is equivalent to hard-wiring $v(\mathbf{x}_i^+) = 1/N$ in Equation 30, so we calculate

$$
\underbrace{\left(\frac{1}{N}\sum_i^N 1 - \overset{+}{w}(\mathbf{x}_i^+)\right)}_{\text{GradScale}}\left(\sum_i^N \nabla_{\boldsymbol{\theta}} f_{\boldsymbol{\theta}}(\mathbf{x}^+) - \sum_{j=1}^M w(\mathbf{x}_j^-)\nabla_{\boldsymbol{\theta}} f_{\boldsymbol{\theta}}(\mathbf{x}_j^-)\right). \quad (33)
$$

Even though this removes the within-batch bias, $\mathrm{GradScale}$ still depends on $\mathbf{x}^+$. To achieve that gradient scaling does not get influenced by $\overset{+}{w}(\mathbf{x}^+)$, we further propose to split $N$ positive examples into two parts, $A$ and $B$, each with $L = N/2$ examples. $A$ contains $\mathbf{x}_i^{\mathrm{a}+}$ and $B$ contains $\mathbf{x}_i^{\mathrm{b}+}$. Then we replace $\overset{+}{w}(\mathbf{x}_i^{\mathrm{a}+})$ by $\overset{+}{w}(\mathbf{x}_A^+)$ and $\overset{+}{w}(\mathbf{x}_i^{\mathrm{b}+})$ by $\overset{+}{w}(\mathbf{x}_B^+)$, which are defined as $\overset{+}{w}(\mathbf{x}_A^+) = \frac{1}{L}\sum_i^L \overset{+}{w}(\mathbf{x}_i^{\mathrm{b}+})$ and $\overset{+}{w}(\mathbf{x}_B^+) = \frac{1}{L}\sum_i^L \overset{+}{w}(\mathbf{x}_i^{\mathrm{a}+})$. We perform the computation for $A$ and $B$ in parallel and average the contributions. The computation for part $A$ then becomes

$$\left( \frac{1}{L}\sum_i^L 1 - \overset{+}{w}(\mathbf{x}_i^{\mathrm{b}+}) \right) \left( \sum_i^L \nabla_{\boldsymbol{\theta}} f_{\boldsymbol{\theta}}(\mathbf{x}_i^{\mathrm{a}+}) - \sum_{j=1}^M w(\mathbf{x}_j^-)\nabla_{\boldsymbol{\theta}} f_{\boldsymbol{\theta}}(\mathbf{x}_j^-) \right). \tag{34}$$

This makes the gradient multiplier $1 - \overset{+}{w}(\mathbf{x}_A^+)$ independent of $\mathbf{x}_i^{\mathrm{a}+}$, resulting in an unbiased estimator of the gradient computed in Equation 2. Negative examples in the batch have no dependency on positive examples from that batch, so we can simply perceive it as extra-added stochasticity through varying learning rates in each batch when using true samples.

## F   INCLUDING POSITIVE EXAMPLE DOES NOT HELP THE STANDARD EBM TRAINING

The trick of adding the positive example as an extra negative (Section 3.1) cannot be applied to the standard training approach corresponding to $\beta = 0$. It would only result in a slightly decreased learning rate and no other effect. Since we assume that the extra negative example $\mathbf{x}_{M+1}^-$ is different for each positive example, the negative energy corresponds to

$$f_{\boldsymbol{\theta}}(\mathbf{x}_{M+1}^-) = \frac{1}{N}\sum_i^N f_{\boldsymbol{\theta}}(\mathbf{x}_i^+). \tag{35}$$

Plugging this result into Equation 3 yields

$$\frac{1}{N}\sum_i^N \nabla_{\boldsymbol{\theta}} f_{\boldsymbol{\theta}}(\mathbf{x}_i^+) - \frac{1}{M+1}\sum_j^{M+1} \nabla_{\boldsymbol{\theta}} f_{\boldsymbol{\theta}}(\mathbf{x}_j^-) = \frac{M}{M+1}\left( \frac{1}{N}\sum_i^N \nabla_{\boldsymbol{\theta}} f_{\boldsymbol{\theta}}(\mathbf{x}_i^+) - \frac{1}{M}\sum_j^M \nabla_{\boldsymbol{\theta}} f_{\boldsymbol{\theta}}(\mathbf{x}_j^-) \right). \tag{36}$$

The same result can can also be derived from Equation 8 by setting $\beta = 0$, for which $1 - \overset{+}{w}(\mathbf{x}^+) = M/M+1$ and $w(\mathbf{x}_j^-) = 1/M$.

## G   PRACTICAL SGLD SAMPLER

Reducing the amount of noise added at each SGLD step (Equation 4) has been a common practice in prior works, often deemed necessary for generating negative examples of reasonable quality within a limited number of steps. Instead of $u_i^t \sim \mathcal{N}(0, \alpha^t)$, typically $u_i^t \sim 0.01 * \mathcal{N}(0, \alpha^t)$ is used and $1/2$ multiplying $f_{\boldsymbol{\theta}}(\mathbf{x}^{t-1})$ is ommitted as well. This adjustment is sometimes incorrectly referred to as a practical trick, as it is a legitimate technique corresponding to EBM parametrization with a different temperature. This modification can be rewritten as a proper sampling from EBM having negative energy $20000 f_{\boldsymbol{\theta}}(\mathbf{x})$, i.e. $p_{\boldsymbol{\theta}}(\mathbf{x}) \propto e^{20000 f_{\boldsymbol{\theta}}(\mathbf{x})}$. Consequently, it is crucial to scale $f_{\boldsymbol{\theta}}(\mathbf{x})$ by 20000 when comparing unnormalized log-likelihood values. We can still apply the same objective function, as the reparametrization from $f_{\boldsymbol{\theta}}(\mathbf{x})$ to $20000 f_{\boldsymbol{\theta}}(\mathbf{x})$ only causes the overall scaling of the loss. Alternatively, we could achieve the same outcome by keeping the proper noise level in the SGLD procedure, but scaling the output of the last NN layer by 20000 and possibly adjusting the learning rate, which reveals the true essence of the trick. For optimizers such as Adam, this parameterization (both explicit and implicit via the change of SGLD amount of noise) effectively multiplies the learning rate in the last layer of NN modeling $f_{\boldsymbol{\theta}}(\mathbf{x})$ by 20000.

### G.1   THE SAME PARAMETER UPDATE TRAINS DIFFERENT EBMS

In Section 3, we explained the proposed method via sampling from $q_{\boldsymbol{\theta}}(\mathbf{x}) \propto e^{(1-\beta) f_{\boldsymbol{\theta}}(\mathbf{x})}$ and then using $r_{\boldsymbol{\theta}}(\mathbf{x}) \propto e^{\beta f_{\boldsymbol{\theta}}(\mathbf{x})}$ for reweighting the gradient. For illustration, we set $\beta = 0.5$, then

$q_{\boldsymbol{\theta}}(\mathbf{x}) = r_{\boldsymbol{\theta}}(\mathbf{x}) = s_{\boldsymbol{\theta}}(\mathbf{x}) \propto e^{1/2 f_{\boldsymbol{\theta}}(\mathbf{x})}$. The sampling from $s_{\boldsymbol{\theta}}(\mathbf{x})$, followed by reweighting based on $s_{\boldsymbol{\theta}}(\mathbf{x})$ should train $p_{\boldsymbol{\theta}}(\mathbf{x})$ with $\beta = 0.5$. If we would like to train $s_{\boldsymbol{\theta}}(\mathbf{x})$ with the standard EBM training approach ($\beta = 0$) instead, the gradient computation would be

$$\nabla_{\boldsymbol{\theta}} \mathbb{E}_{p_d(\mathbf{x})} \left[ \log s_{\boldsymbol{\theta}}(\mathbf{x}) \right] = 1/2 \left( \mathbb{E}_{p_d(\mathbf{x})} \left[ \nabla_{\boldsymbol{\theta}} f_{\boldsymbol{\theta}}(\mathbf{x}) \right] - \mathbb{E}_{s_{\boldsymbol{\theta}}(\mathbf{x})} \left[ \nabla_{\boldsymbol{\theta}} f_{\boldsymbol{\theta}}(\mathbf{x}) \right] \right). \tag{37}$$

A comparison with Equation 2 reveals that the difference between the equations for $p_{\boldsymbol{\theta}}(\mathbf{x})$ and $s_{\boldsymbol{\theta}}(\mathbf{x})$ lies solely in the magnitude of the gradient, and the distribution from which SGLD samples. Consequently, as long as the negative examples are samples from $p_{\boldsymbol{\theta}}(x)^k$ for an arbitrary $k$, the same computation used to update $\boldsymbol{\theta}$ trains an EBM with a negative energy of $k f_{\boldsymbol{\theta}}(\mathbf{x})$. This flexibility enables us to potentially train some EBM $p_{\boldsymbol{\theta}}(x) \propto e^{k f_{\boldsymbol{\theta}}(\mathbf{x})}$ using the same $\boldsymbol{\theta}$ update rule (Equation 2) as long as SGLD provides samples from the current $p_{\boldsymbol{\theta}}(x)^k$ for any $k$. However, the drawback is that we will not know the exact value of $k$.

We mention this in the context of the proposed method for two main reasons. First, we suggested that sampling should be based on $(1 - \beta) f_{\boldsymbol{\theta}}(x)$ and then reweighting should be based on $\beta f_{\boldsymbol{\theta}}(x)$. This approach can be also related to discriminative training using softmax with inverse temperature $\beta$ when considering a variant with an included extra negative example. In terms of explained parametrization, $1 - \beta$ and $\beta$ can be replaced by arbitrary values $a$ and $b$ as they will correspond to $a = k(1 - \beta)$ and $b = k\beta$ for some $k$. Second, the argument about the flexibility of training remains valid as long as SGLD provides samples from $p_{\boldsymbol{\theta}}(x)^k$ for any $k$ even for $\beta \neq 0$. So the possibility of training different EBMs using Equation 2 extends to Equation 7.

## H    THE NEGATIVE INFLUENCE OF ADAPTIVE LEARNING RATE

When all generated negative examples have low likelihood values, and for sufficiently large $\beta$, the parameter ($\boldsymbol{\theta}$) updates during training are effectively disregarded due to the presence of the additional positive example. The positive example multiplies the gradient by $(1 - \overset{+}{w}(\mathbf{x}^+)) \approx 0$ in Equation 8. When a certain $\theta$ is reached during training and then suddenly all negative examples stop being proper, the proposed updates should almost not modify $\boldsymbol{\theta}$, thus preserving the currently achieved solution $\boldsymbol{\theta}$. Conversely, the standard training approach ($\beta = 0$) continues to modify $\boldsymbol{\theta}$, essentially erasing previously learned information.

However, it is important to note a potential complication that could prevent the described behavior. If negative examples are consistently inadequate for many iterations, we should observe very low gradient values as $(1 - \overset{+}{w}(\mathbf{x}^+)) \approx 0$. Despite this, certain optimizers like AdaGrad (Duchi et al., 2011) or Adam (Kingma & Ba, 2014) might adapt to this situation and effectively amplify the gradient values. In such cases, the proposed solution would only slow down the effect of diverging from the current $\boldsymbol{\theta}$, but it would not prevent it. Nevertheless, if no competitive negative example is generated for an extended duration, proactive measures should be taken. This involves adjusting the procedure for generating negative examples to ensure that at least some have comparable likelihood values. Failing to do so would compromise the effectiveness of training EBM.

## I    REQUIREMENTS FOR SOLUTION OBTAINED WHEN TRAINING CONVERGES

In Section 2.2, we discussed that for setting $\beta = 0$ if $\text{SGLD}(p_{\boldsymbol{\theta}}(\mathbf{x})) = p_d(\mathbf{x})$, it is a solution to the optimization problem. In general, the sufficient condition for convergence is satisfied based on a moment matching and not necessarily distribution matching (Nijkamp et al., 2019), i.e. $\mathbb{E}_{\text{SGLD}(p_{\boldsymbol{\theta}}(\mathbf{x}))} \left[ \nabla_{\boldsymbol{\theta}} f_{\boldsymbol{\theta}}(\mathbf{x}) \right] = \mathbb{E}_{p_d(\mathbf{x})} \left[ \nabla_{\boldsymbol{\theta}} f_{\boldsymbol{\theta}}(\mathbf{x}) \right]$, but for the sake of the following discussion, we will not distinguish between these two cases. When $\beta \neq 0$, this is no longer valid. The larger the $\beta$, the more important $f_{\boldsymbol{\theta}}(\mathbf{x})$ is over $\nabla_{\mathbf{x}} f_{\boldsymbol{\theta}}(\mathbf{x})$, which is employed in SGLD procedure. Considering an arbitrary value of $\beta$, the convergence is reached when $p_d(\mathbf{x}) \propto \text{SGLD}(p_{\boldsymbol{\theta}}(\mathbf{x})^{1-\beta}) e^{\beta f_{\boldsymbol{\theta}}(\mathbf{x})}$ or alternatively $p_d(\mathbf{x}) \propto e^{\beta f_{\boldsymbol{\theta}}(\mathbf{x}) + \log \text{SGLD}(p_{\boldsymbol{\theta}}(\mathbf{x})^{1-\beta})}$. If[7] $\text{SGLD}(p_{\boldsymbol{\theta}}(\mathbf{x})^{1-\beta}) = \text{SGLD}(p_{\boldsymbol{\theta}}(\mathbf{x}))^{1-\beta}$, then we could interpret the result as $p_d(\mathbf{x}) \propto e^{\beta f_{\boldsymbol{\theta}}(\mathbf{x}) + (1-\beta) \log \text{SGLD}(p_{\boldsymbol{\theta}}(\mathbf{x}))}$, therefore, for $\beta \neq 0$ : $f_{\boldsymbol{\theta}}(\mathbf{x}) = 1/\beta \log p_d(\mathbf{x}) + (1 - 1/\beta) \log \text{SGLD}(p_{\boldsymbol{\theta}}(\mathbf{x})) + c = \log p_d(\mathbf{x}) + (1 - 1/\beta)(\log \text{SGLD}(p_{\boldsymbol{\theta}}(\mathbf{x})) - \log p_d(\mathbf{x})) + c$, where c is an arbitrary constant. The learned negative energy will have to compensate

---

[7]Even though it might not hold in practice, it helps as an analysis tool.

for the difference between $\log \mathrm{SGLD}(p_{\boldsymbol{\theta}}(\mathbf{x}))$ and $\log p_d(\mathbf{x})$. For values of $\beta$ between 0 and 1, $1 - 1/\beta$ will be negative. Consequently, for $\mathbf{x}$ where $\mathrm{SGLD}(p_{\boldsymbol{\theta}}(\mathbf{x})) > p_d(\mathbf{x})$, we will have $p_{\boldsymbol{\theta}}(\mathbf{x}) < p_d(\mathbf{x})$, and vice versa. The gap between $p_d(\mathbf{x})$ and $p_{\boldsymbol{\theta}}(\mathbf{x})$ increases as $\beta$ grows; however, it is also expected that the difficulty of reaching a convergent solution will increase with larger values of $\beta$. This analysis also shows that when $\beta \neq 0$, the negative energy $f_{\boldsymbol{\theta}}(\mathbf{x})$ must encode some information about $p_d(\mathbf{x})$.

When using a buffer of previously stored generated examples (PCD) for initial distribution in SGLD, we might effectively increase the number of performed SGLD steps. This would cause the desirable behavior as $\mathrm{SGLD}(p_{\boldsymbol{\theta}}(\mathbf{x}))$ gets closer to $p_{\boldsymbol{\theta}}(\mathbf{x})$. As a result, even in the case $\beta = 0$, $f_{\boldsymbol{\theta}}(\mathbf{x})$ would contain some information about $p_d(\mathbf{x})$. However, setting $\beta \neq 0$ would not break this logic. It is fair to mention that the buffer is used in some cases while it is not used in others. The disadvantage of using the buffer is the problematic sampling at the inference time.

### I.1 THE ROLE OF $\beta$ VALUE

We highlighted the drawbacks of setting $\beta = 0$, on the other hand, setting $\beta \approx 1$ presents its challenges. In this scenario, negative examples are drawn from the uniform distribution $u(\mathbf{x}) \propto e^{0 f_{\boldsymbol{\theta}}(\mathbf{x})}$ as shown in Figure 1c. In the context of the manifold hypothesis, most (practically all) of these examples would reside in low-likelihood regions carrying little information about the desired expectation and we would like them to be filtered out. Increasing $\beta$ makes SGLD steps more driven by sampled noise than $\nabla_{\mathbf{x}} f_{\boldsymbol{\theta}}(\mathbf{x})$. This will have an unwanted effect on the number of required SGLD steps needed to travel from low-likelihood to high-likelihood regions. On the other hand, the mixing rate (traveling across distribution modes) should improve. The effectiveness of filtering out unwanted negative examples (sensitivity to detect outliers) increases with increasing value of $\beta$ as the weight is determined based on $e^{\beta f_{\boldsymbol{\theta}}(\mathbf{x})}$.

The automatic decrease of gradient magnitude is based on the likelihood comparison between a single positive and the sum of all negative examples. However, especially when considering a small number of negative examples $M$ and a large $\beta$, the likelihood of a positive example can be much higher, even with a proper sampler of negative examples. This is because a positive example is drawn from $p_d(\mathbf{x})$, while negative from $p_{\boldsymbol{\theta}}(\mathbf{x})^{1-\beta}$ rather than $p_{\boldsymbol{\theta}}(\mathbf{x})$, but the effect can be partially counteracted by adjusting the learning rate.

These arguments demonstrate a trade-off between larger and smaller values of $\beta$. However, we want to emphasize that as an extension of this work, it is also possible to arbitrarily change the value of $\beta$ during training as long as it is fixed for all negative examples within each batch.

## J ATTRACTOR-REPELLER DYNAMICS: AN EBM TRAINING HYPOTHESIS

We initially expected that the difference between EBM training with $\beta = 0$ and $\beta < 0.01$ would be imperceptible or, at most, result in minimal differences. However, when applied to real data, the difference is pronounced. While the SGLD sampler is capable of providing negative examples $\mathbf{x}^-$ with $f_{\boldsymbol{\theta}}(\mathbf{x}^-)$ close to $f_{\boldsymbol{\theta}}(\mathbf{x}^+)$ for $\beta = 0$, it fails for $\beta = 0.01$, resulting in inefficient training. From a theoretical point of view, updating parameters based on calculations with $\beta = 0.01$ should be more precise. However, we propose a hypothesis that explains why this less precise update can help the SGLD sampler be more effective. This hypothesis is based on the concepts of attractors and repellers, which we refer to as "attractor-repeller training".

As $\beta$ increases, the weight of negative examples with lower likelihood values decreases to the point where they are effectively ignored. This approach is in line with the principles of EBM training, as these negative examples should not be produced in the first place. However, as a consequence, $f_{\boldsymbol{\theta}}(\mathbf{x}^-)$ remains unaffected around this $\mathbf{x}^-$, and the SGLD procedure also remains unchanged. Consequently, repeating the SGLD procedure with the same initialization is likely to yield a similar negative example. In contrast, when $\beta = 0$, such negative examples are treated as true samples, further incorrectly decreasing already small $f_{\boldsymbol{\theta}}(\mathbf{x}^-)$, which causes the trained EBM to deviate from modeling the correct distribution. Therefore, this process reduces the probability of generating the same negative example again.

We can interpret this by conceptualizing the practical SGLD procedure as an implicit multi-step generator that iteratively explores the domain of $\mathbf{x}$. As $f_{\boldsymbol{\theta}}(\mathbf{x}^+)$ increases for positive examples $\mathbf{x}^+$, local

maxima may form, serving as attractors for the SGLD procedure. Simultaneously, negative examples function as repellers, as updates can create local minima at those locations. The establishment of a local minimum forces SGLD to search for another negative example in the subsequent iteration. This explains why the effectiveness of the SGLD procedure is significantly greater when $\beta = 0$ compared to $\beta = 0.01$. The setting $\beta = 0$ produces an effective implicit sampler, resembling GAN (Goodfellow et al., 2020) training; however, as we have shown, it does not correspond to the training of EBM modeling the true data distribution.

Moreover, since increasing $\beta$ eliminates the divergent behavior, the attractor-repeller training appears to be responsible for training instabilities. It tends to decrease $f_{\boldsymbol{\theta}}(\mathbf{x}^-)$ too rapidly in regions where $f_{\boldsymbol{\theta}}(\mathbf{x}^-)$ is already small, which consequently leads to unrestricted growth of $f_{\boldsymbol{\theta}}(\mathbf{x}^+)$—a phenomenon that would not occur in true EBM training. While true EBM training can also be viewed as an attractor-repeller scheme, we use this term to describe a more general training method that does not require negative examples to be sampled from the model. Instead, it only requires that increasing $f_{\boldsymbol{\theta}}(\mathbf{x})$ raises the probability of $\mathbf{x}$ being sampled, while decreasing $f_{\boldsymbol{\theta}}(\mathbf{x})$ lowers this probability.

## K    DETAILED ANALYSIS OF DENSITY LEARNED ON TOY DATASETS

This section extends Section 4.1 by providing additional scenarios for the toy datasets.

### K.1    IMPACT OF SGLD SAMPLER BIAS ON LEARNED LIKELIHOOD

The training with the default setup for the circles dataset didn't converge, and running twice as many training steps results in learned $p_{\boldsymbol{\theta}}(\mathbf{x})$ that is more similar to $p_d(\mathbf{x})$ when $\beta > 0$. For the reference, we provide the comparison in Figure 5. In some cases, especially for $\beta = 0$, the likelihood or even log-likelihood might appear to have no variations, i.e. score $\nabla_{\mathbf{x}} f_{\boldsymbol{\theta}}(\mathbf{x})$ appears to be 0. However, the local estimates of scores are usually still informative as they guide the SGLD procedure, but the values of likelihood are way too low to be shown by different colors. Investigating the variant $\beta = 0.9$ illustrates that even when twice as many iterations are performed, $p_{\boldsymbol{\theta}}(x)$ does not converge to $p_d(\mathbf{x})$ as the weight of each circle is estimated incorrectly. This corresponds to the analysis provided in Appendix I. The analysis suggests that the learned $f_{\boldsymbol{\theta}}(\mathbf{x})$ must compensate for incorrect SGLD samples, i.e. decreasing the log-likelihood at places where $\mathrm{SGLD}(p_{\boldsymbol{\theta}}(\mathbf{x})^{1-\beta}) > p_d(\mathbf{x})^{1-\beta}$ and vice versa. In this case, SGLD tends to produce negative examples more often in the central part of the plot because it is closer to the initial distribution of SGLD. To confirm this behavior, we created an additional experimental setup with a different initial Gaussian distribution, having half the size of the standard deviation and mean shifted to the bottom right quadrant of the default initial distribution. At the same time, we increased the SGLD step size 10 times to compensate for the additional distance needed to travel. Examination of the results shown in Figure 6 confirms the same pattern. As the SGLD procedure with limited steps needs to transport probability mass from the bottom right corner into the top left corner of the plot, it affects the learned density. When $\beta = 0$, SGLD still learns to generate samples resembling the data distribution, although $f_{\boldsymbol{\theta}}(\mathbf{x})$ corresponds to another EBM. Settings $\beta \neq 0$ again correspond to $f_{\boldsymbol{\theta}}(\mathbf{x})$ that better reflects the data distribution. Similarly, learned $p_{\boldsymbol{\theta}}(\mathbf{x})$ needs to compensate for the shift between $\mathrm{SGLD}(p_{\boldsymbol{\theta}}(\mathbf{x}))$ and $p_{\boldsymbol{\theta}}(\mathbf{x})$. This adjusted setup resulted in training divergence for $\beta = 0$ and the circles dataset.

### K.2    COMPARISON OF APPROACHES FOR INCORPORATING POSITIVE EXAMPLE

We compare our default variant that incorporates a positive example among negative ones (Equation 8), to the variant without including it (No pos, Equation 7). Additionally, we compare it to variants, where we remove the objective bias caused by including positive example within the batch (Batch corr, Equation 33) or within the whole training (True obj, Equation 34) as described in Appendix E.1. Note that all variants are equivalent for the baseline system ($\beta = 0$).

The performance of all considered proposed variants appears to be very similar in the default setup (Figure 7). To see the difference, we introduce two additional setups. First, we limit the number of generated negative examples per batch to 3, which are additionally generated using only 3 SGLD steps, and show how both the temperature and variants affect learned density on the GMM dataset in Figure 10 and the circles dataset in Figure 11. The results suggest that the default variant should be preferred over the others (No pos, True obj, Batch corr), so the bias it causes appears to positively

affect the training. The motivation for introducing our default variant was to handle cases when all negative examples might be uninformative. To examine the limits of this setup, we further resort to a single negative example per batch that directly comes from the initial distribution (0 SGLD steps). Figure 8 comparing different variants illustrates that the default variant is still able to learn some characteristics of the data distribution. Moreover, we show that as $\beta$ increases, $p_{\theta}(x)$ gets closer to $p_d(\mathbf{x})$ in Figure 9. This result should be taken with a grain of salt, as this behavior is probably limited to low-dimensional (toy) datasets. However, as we did not observe any case when the default variant performs worse than other variants, we only consider the default variant in the following experiments. Since we did not perform any SGLD steps in this setup, the column corresponding to $\mathrm{SGLD}(p_{\theta}(\mathbf{x}))$ visualizes the default initial distribution.

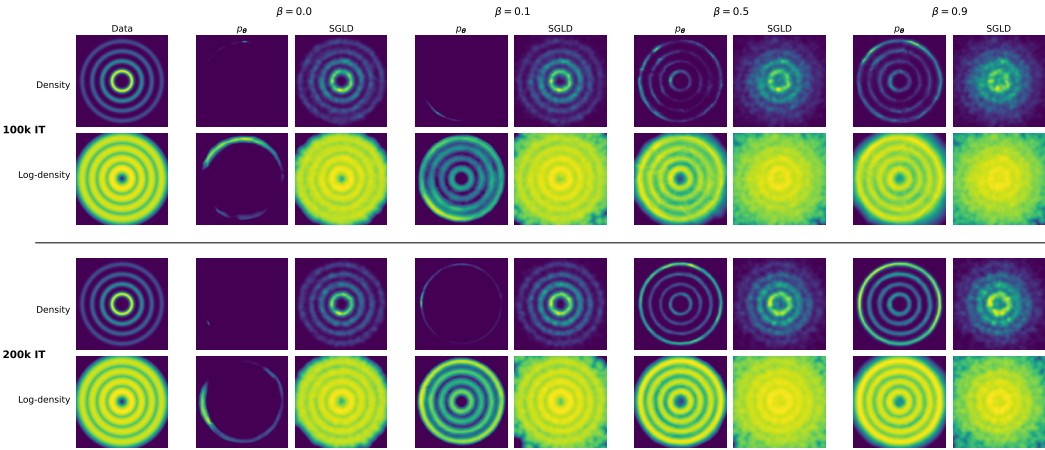

Figure 5: The default setup on the circles dataset compared to twice as long run (200k iterations).

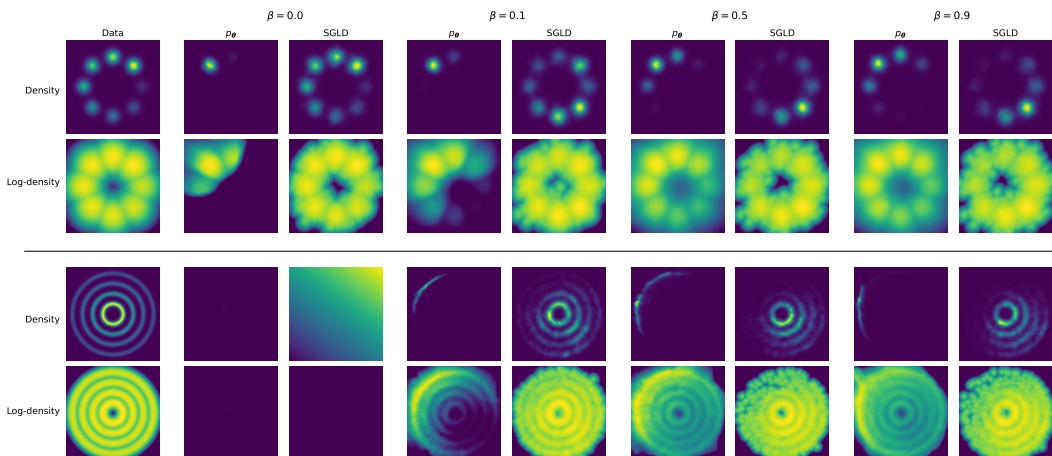

Figure 6: The default setting with the modified initial distribution. The initial samples now originate from the bottom right part of the plots.

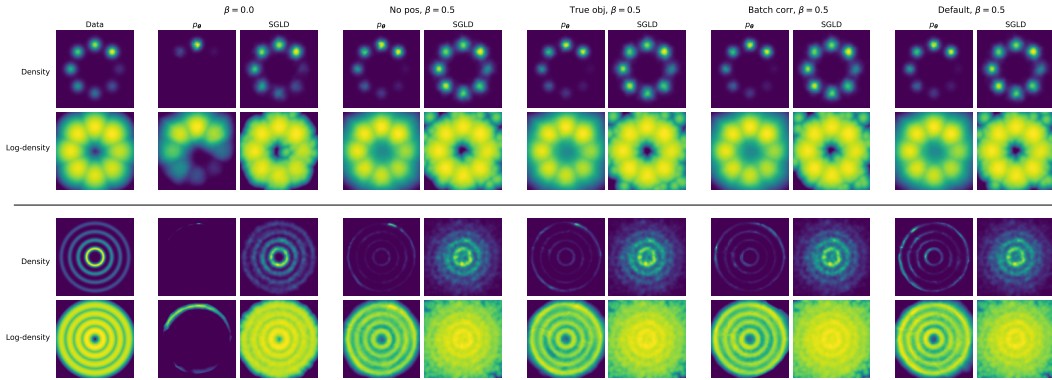

Figure 7: Different proposed variants compared in the default setup. With enough negative examples and SGLD steps, all proposed variants perform similarly.

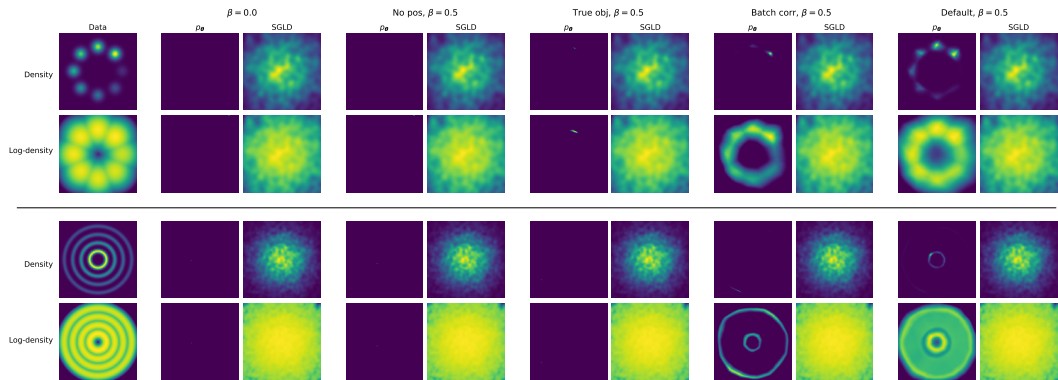

Figure 8: Degenerated setup, where we use only a single negative example initialized from Gaussian distribution per each batch and do not perform any SGLD steps. The learned negative energy of the default proposed method partially reflects the true log-likelihood of data.

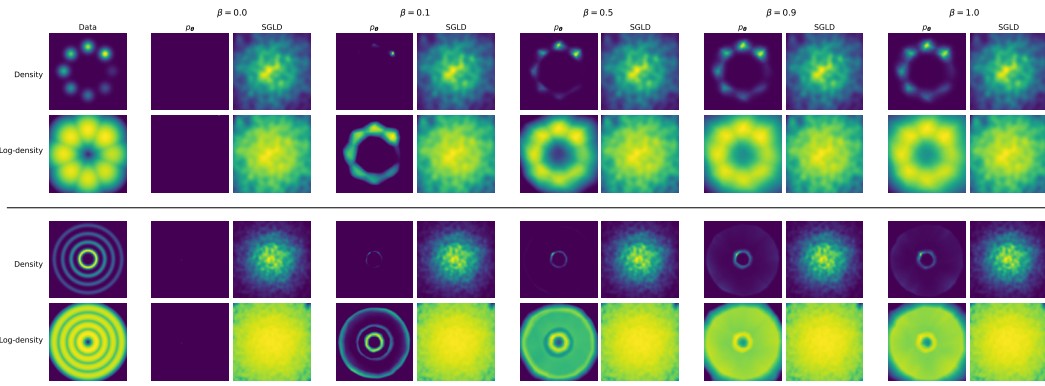

Figure 9: Degenerated setup, where we use only a single negative example initialized from Gaussian distribution per each batch and do not perform any SGLD steps. The learned log-likelihood contains some information about the true log-likelihood of data when $\beta \neq 0$.

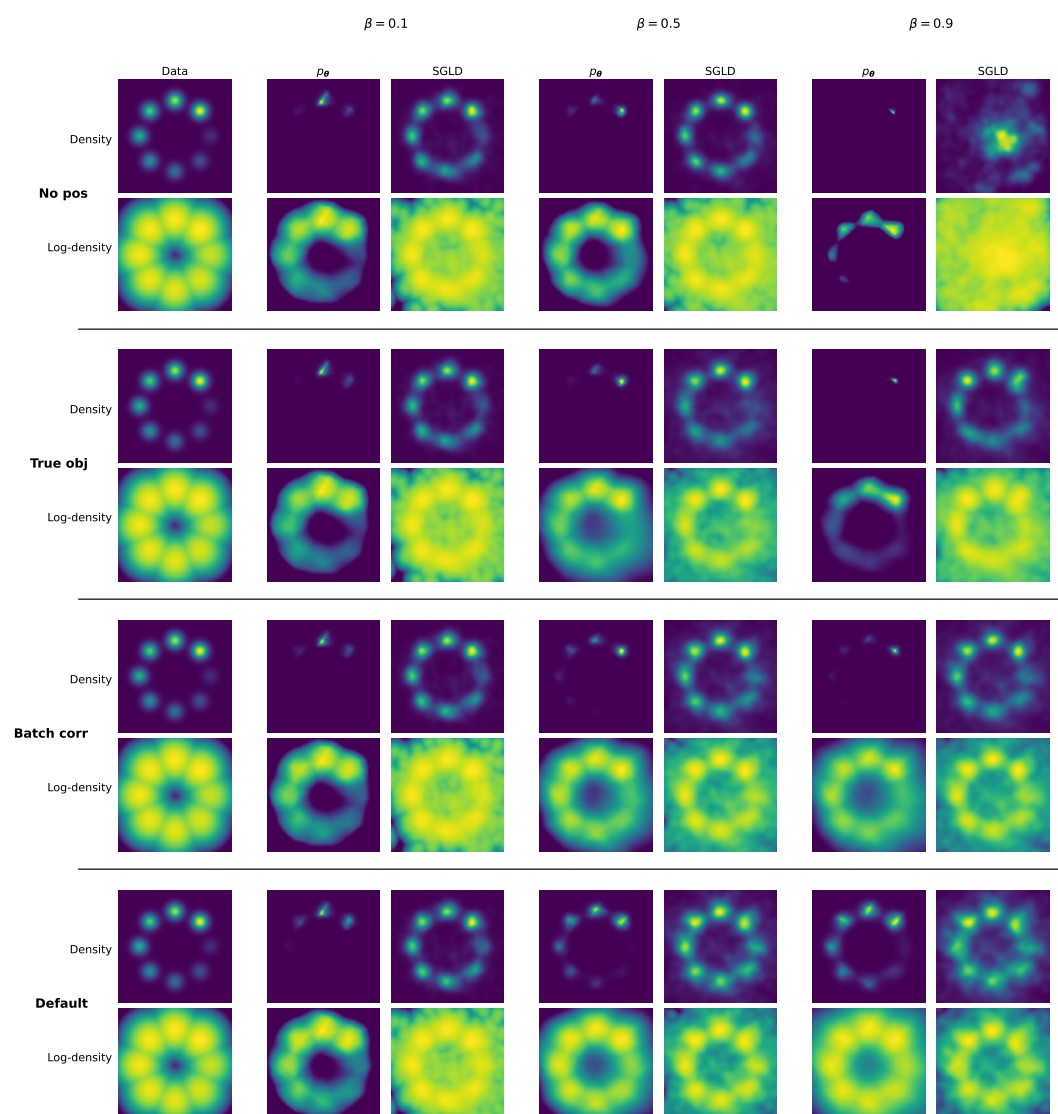

Figure 10: Comparison of different variants and different values of $\beta$ in the restricted setup of the GMM toy dataset. We allow only 3 (default 20) negative examples per each batch of 50 positive examples and perform 3 (default 20) SGLD steps. The default variant learns the most similar energy function.

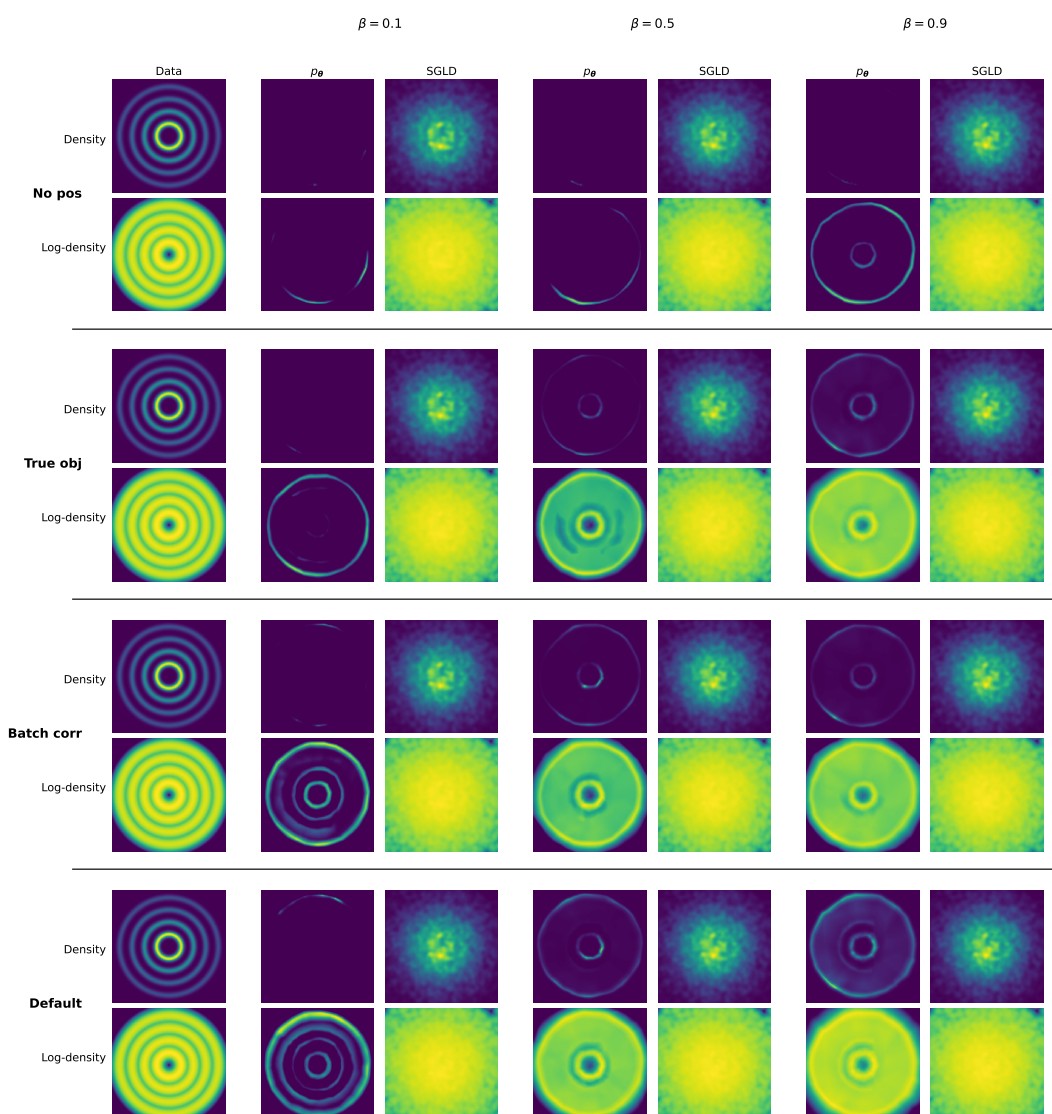

Figure 11: Comparison of different variants and different values of $\beta$ in the restricted setup of the circles toy dataset. We allow only 3 (default 100) negative examples per each batch of 100 positive examples and perform 3 (default 100) SGLD steps. The default variant learns the most similar energy function.

## L   DETAILED ANALYSIS OF ENERGY-BASED MODEL TRAINING ON CIFAR-10

We compare the difference in the training of EBMs based on the value of $\beta$ using the setup of Nijkamp et al. (2019). We conduct experiments with very small values of $\beta$. For $\beta$ in this range, the SGLD procedure remains effectively unaffected. In Figure 12, we visualize the evolution of the negative energy means for both negative and positive examples, with means calculated based on the examples in a single batch. As $\beta$ increases, the gap between $f_{\boldsymbol{\theta}}(\mathbf{x}^+)$ and $f_{\boldsymbol{\theta}}(\mathbf{x}^-)$ widens. Since efficient training requires negative examples with comparable likelihood values, this slows the training down. Note that, in the case of an inaccurate sampler, reweighting the negative examples due to $\beta > 0$ provides a more reliable estimate of the true weight that should be used in the correct EBM parameter updates. However, once we begin to "properly" reduce the weights of these examples, the SGLD procedure fails to generate negative examples with likelihoods comparable to those of the positive examples. This suggests we are performing a less biased update of the model parameters, but as a result, the gap

between $f_{\boldsymbol{\theta}}(\mathbf{x}^+)$ and $f_{\boldsymbol{\theta}}(\mathbf{x}^-)$ increases further. In contrast, for $\beta = 0$, the model parameter update is less precise, but it does not widen the gap between $f_{\boldsymbol{\theta}}(\mathbf{x}^+)$ and $f_{\boldsymbol{\theta}}(\mathbf{x}^-)$.

We further investigate the behavior of the trained models[8]. We generate negative examples $\mathbf{x}^-$ using the SGLD procedure with the hyperparameters used during training and compare the histograms of $f_{\boldsymbol{\theta}}(\mathbf{x}^-)$ and $f_{\boldsymbol{\theta}}(\mathbf{x}^+)$ in Figure 13. This confirms the difference between the distribution of negative and positive examples. The goal of EBM training is to reduce the likelihood of negative examples while increasing the likelihood of positive examples. From this perspective, increasing $\beta$ would lead to better-trained models, if the negative examples were representative. We verify that they are not representative by running SGLD with a different setting. We increase the SGLD step size by a factor of four and show the results in Figure 14. From a theoretical standpoint, this change should not alter the distribution of generated examples. However, we observe that it is indeed possible to produce negative examples with much higher likelihood values, confirming that the model is poorly trained in terms of likelihood values. Finally, we demonstrate that the visual quality of the negative examples generated from trained models using the unmodified SGLD degrades as $\beta$ increases, as shown in Figure 15.

---

[8]We refer to models as trained after performing $100,000$ updates, although models with larger $\beta$ may not be fully trained.

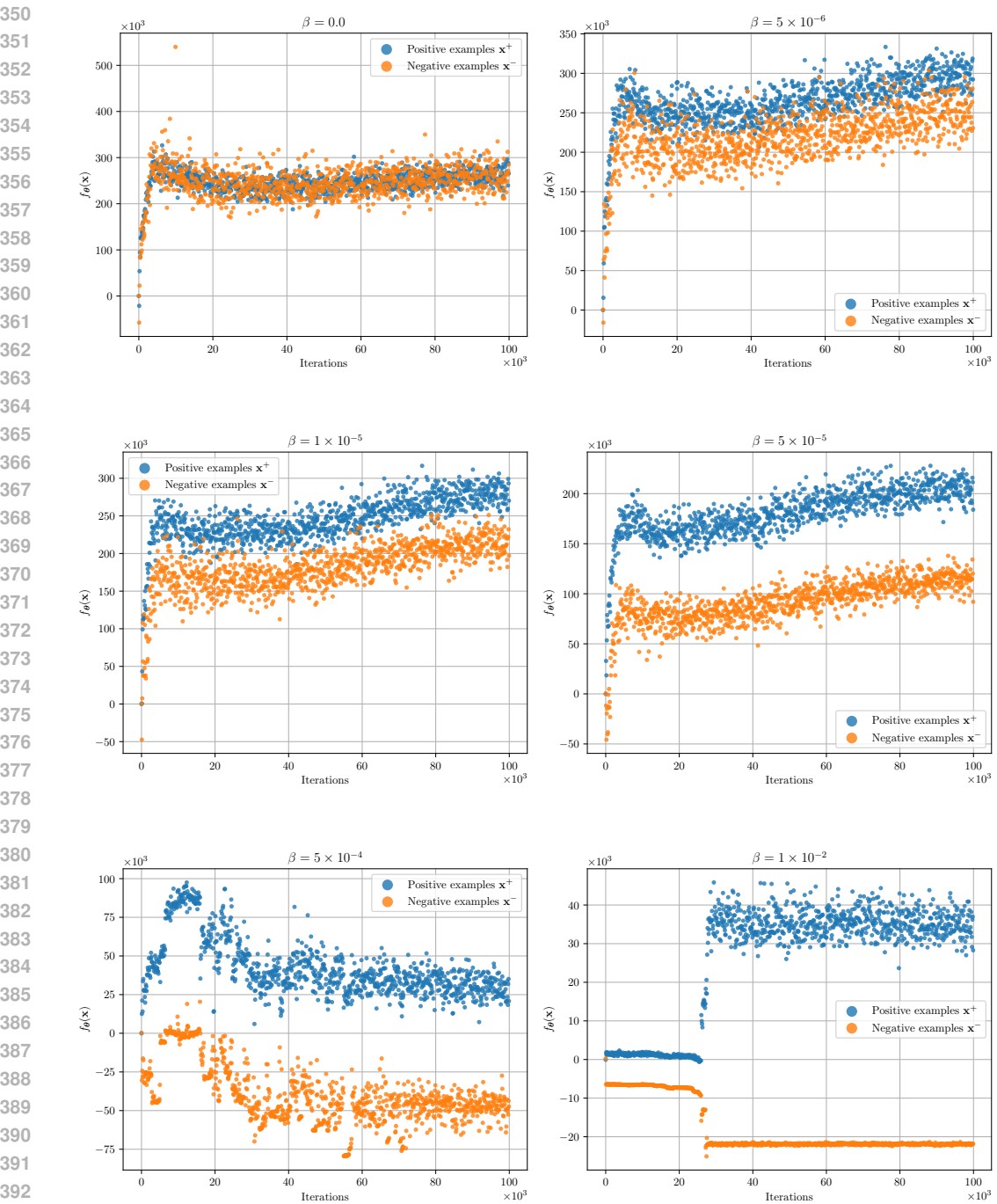

Figure 12: The evolution of the average value of $f_{\boldsymbol{\theta}}(\mathbf{x}^+)$ and $f_{\boldsymbol{\theta}}(\mathbf{x}^-)$ during training EBM. The averages are calculated over positive and negative examples within a single batch, sampled once every 100 iterations.

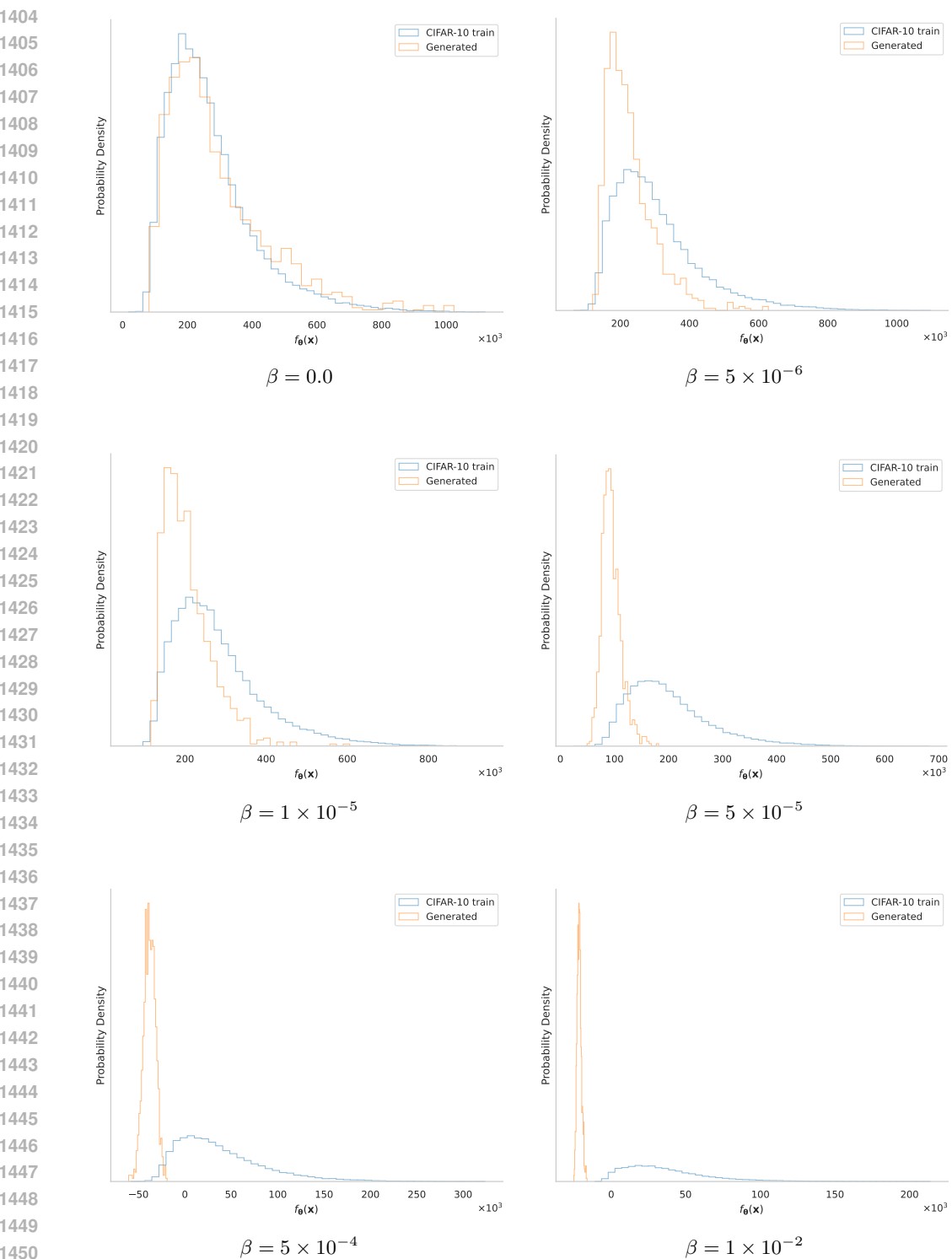

Figure 13: Visualization of the distribution of $f_{\boldsymbol{\theta}}(\mathbf{x})$ values for trained EBMs using CIFAR-10 dataset. We compare CIFAR-10 training data with generated negative examples. Each plot corresponds to a model trained with a different $\beta$, indicated below the respective plot.

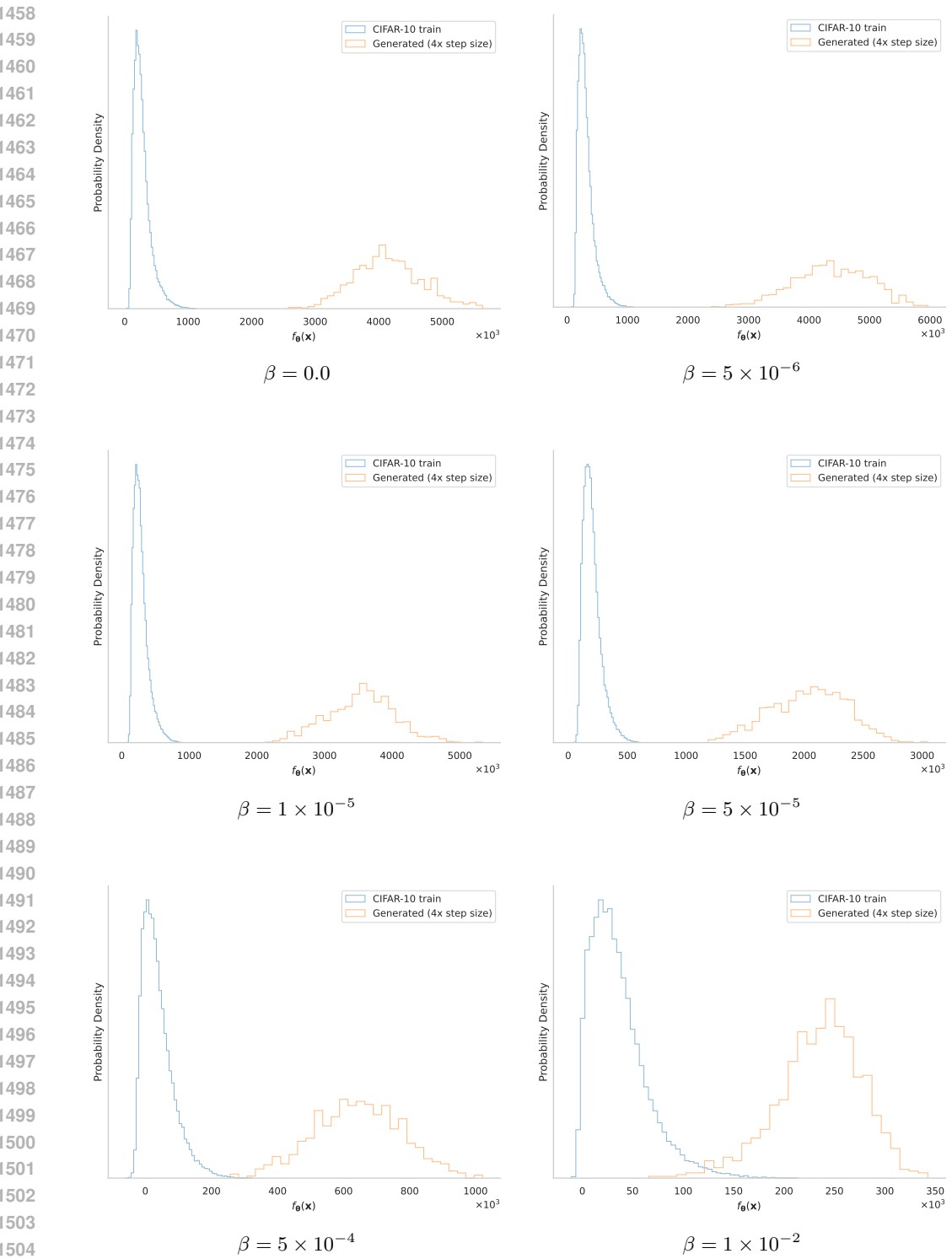

Figure 14: Visualization of the distribution of $f_{\boldsymbol{\theta}}(\mathbf{x})$ values for trained models using CIFAR-10 dataset. We compare CIFAR-10 training data with negative examples generated with a modified SGLD procedure. The modification lies in performing $4\times$ larger step sizes. Each plot corresponds to a model trained with a different $\beta$, indicated below the respective plot.

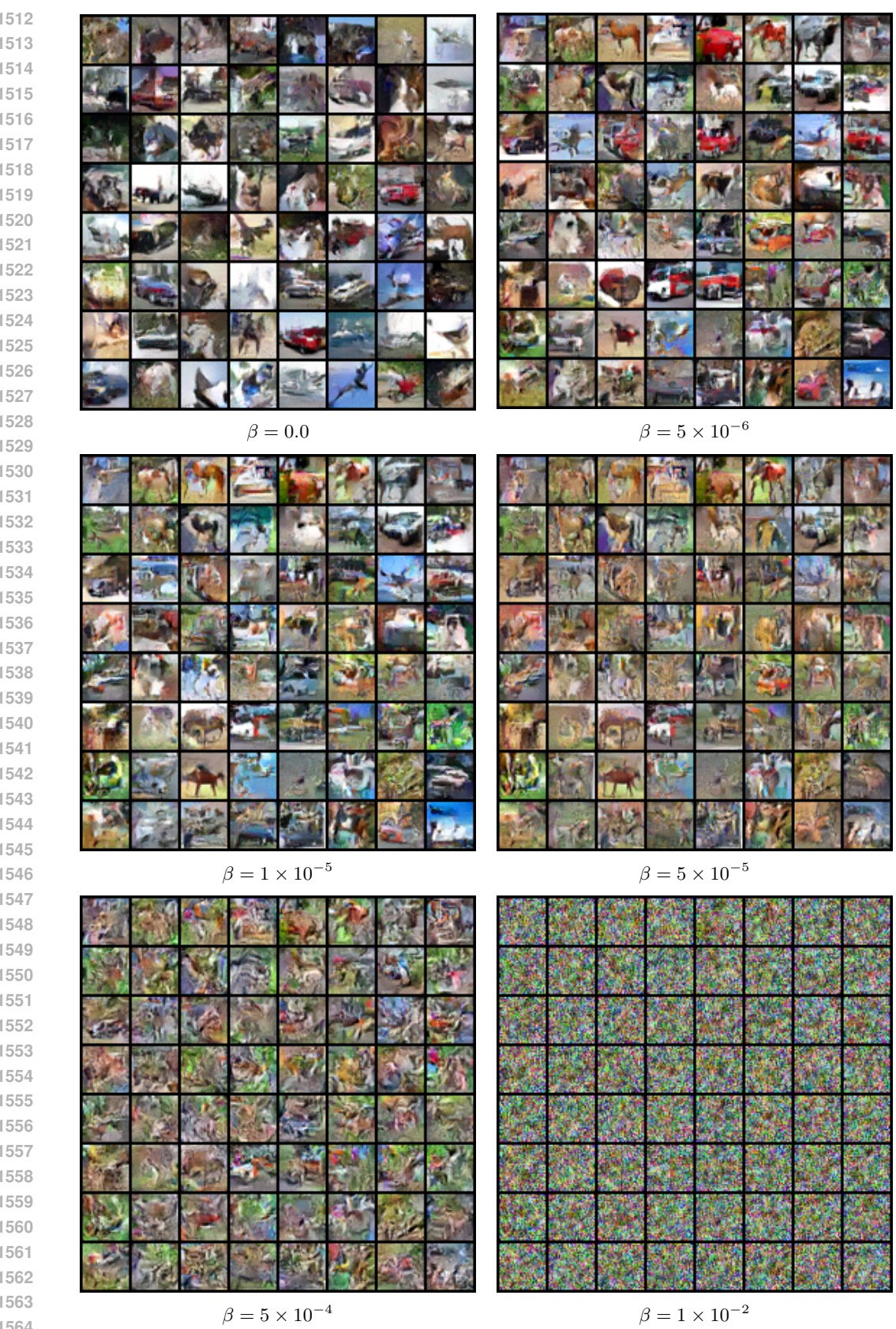

Figure 15: The negative examples generated during the final (100k) training iteration of EBM. Each image grid corresponds to a model trained with a different $\beta$, indicated below the respective grid.

## M   DETAILED ANALYSIS OF JOINT ENERGY-BASED MODEL TRAINING

The Joint Energy-based Model (JEM) (Grathwohl et al., 2019) combines an energy-based model with a classifier by modeling the negative energy $h_{\boldsymbol{\theta}}(\mathbf{x})[y]$ of the joint distribution $p_{\boldsymbol{\theta}}(\mathbf{x}, y)$, where $y$ represents the class in the classification model. In this formulation, $e^{h_{\boldsymbol{\theta}}(\mathbf{x})[y]} \propto p_{\boldsymbol{\theta}}(\mathbf{x}, y)$. Since

$$\log p_{\boldsymbol{\theta}}(y \mid \mathbf{x}) = \log \frac{p_{\boldsymbol{\theta}}(\mathbf{x}, y)}{p_{\boldsymbol{\theta}}(\mathbf{x})} = \log \frac{p_{\boldsymbol{\theta}}(\mathbf{x}, y)}{\sum_y p_{\boldsymbol{\theta}}(\mathbf{x}, y)} = \log \frac{e^{h_{\boldsymbol{\theta}}(\mathbf{x})[y]}}{\sum_y e^{h_{\boldsymbol{\theta}}(\mathbf{x})[y]}}, \tag{38}$$

the negative energy $h_{\boldsymbol{\theta}}(\mathbf{x})[y]$ can be directly used as the logits for the classifier. The negative energy $f_{\boldsymbol{\theta}}(\mathbf{x})$ of $p_{\boldsymbol{\theta}}(\mathbf{x})$ is related to the negative energy of $p_{\boldsymbol{\theta}}(\mathbf{x}, y)$ as

$$e^{f_{\boldsymbol{\theta}}(\mathbf{x})} \propto p_{\boldsymbol{\theta}}(\mathbf{x}) = \sum_y p_{\boldsymbol{\theta}}(\mathbf{x}, y) \propto \sum_y e^{h_{\boldsymbol{\theta}}(\mathbf{x})[y]}. \tag{39}$$

Given that the negative energy of a distribution is the logarithm of a likelihood plus an arbitrary constant, it can be obtained as $f_{\boldsymbol{\theta}}(\mathbf{x}) = \log \sum_y e^{h_{\boldsymbol{\theta}}(\mathbf{x})[y]}$. We maximize $\log p_{\boldsymbol{\theta}}(x, y)$ by factorizing it as $\log p_{\boldsymbol{\theta}}(x, y) = \log p_{\boldsymbol{\theta}}(\mathbf{x}) + \log p_{\boldsymbol{\theta}}(y \mid \mathbf{x})$. This allows us to separately maximize $\log p_{\boldsymbol{\theta}}(\mathbf{x})$ as a standard energy-based model (Equation 3) and $\log p_{\boldsymbol{\theta}}(y \mid \mathbf{x})$ as a classifier by minimizing the cross-entropy, as described in Appendix C.1.

We selected this work to demonstrate the usefulness of our method for two key reasons. First, it uses a standard architecture, Wide Residual Networks (Zagoruyko & Komodakis, 2016), rather than a specialized one designed for generative performance. This choice leads to frequent training instabilities, and the authors report that the only remedy is restarting training from the last checkpoint and resetting the random seed. A further increase in the number of SGLD steps becomes necessary if that proves ineffective. These instabilities worsen with any changes to the hyperparameters, making model development particularly challenging. We demonstrate that an appropriate choice of $\beta$ can mitigate these training divergences. Second, demonstrating the model's utility becomes challenging when the SGLD procedure ceases to provide good samples. To address this, we leverage JEM's ability to function as a classifier, which allows us to evaluate its classification accuracy.

### M.1   EXPERIMENTS

The experiments were conducted using the publicly available code from Grathwohl et al. (2019), extended with the proposed method. Their implementation effectively introduces an additional hyperparameter, the output scale, with a default value of 20,000. Details are provided in Appendix M.2, though understanding these details is not essential for this section. The rest of their setup remained unchanged, except for continuing the training for two additional epochs after detecting divergence[9], for analysis purposes. The setup uses only 20 SGLD steps[10], but incorporates persistent contrastive divergence. The replay buffer has a size of 10,000, with a 0.05 probability of reinitializing a negative example using uniformly distributed noise. Training runs for 200 epochs with 703 iterations (updates) per epoch, using 64 positive and 64 negative examples per iteration. SGLD employs a step size of $\alpha = 0.0001$, and the training data is augmented with Gaussian noise with 0.03 standard deviation. We perform experiments using the default dataset, CIFAR-10.

The summary of training behavior using the default setup with different values of $\beta$ is presented in Figure 16. The first graph tracks training progress by measuring the difference between average $f_{\boldsymbol{\theta}}(\mathbf{x}^+)$ and $f_{\boldsymbol{\theta}}(\mathbf{x}^-)$ for each batch. We aggregate 200 consecutive values, representing them with their minimum and maximum (the transparent regions), and use the center of gravity of these points in the graph[11] as the representative value. The other graphs in Figure 16 display the evolution of Inception Score (IS) (Salimans et al., 2016), Fréchet Inception Distance (FID) (Heusel et al., 2017), and classification accuracy on the test set throughout training. IS and FID are calculated based on the content of the replay buffer. As shown in the graphs, JEM training cannot be completed with the standard loss ($\beta = 0$) due to an abrupt divergence around epoch 55, associated with a sudden drop in the quality of generated images, as illustrated in Figure 17.

---

[9]In some cases, divergence occurred earlier than detected, but the timing is not critical for these experiments.

[10]Nijkamp et al. (2019) used 100 SGLD steps.

[11]We use the center of gravity instead of the mean value to account for the symmetrical logarithmic scale of the vertical axis.

We can prevent this divergence by using a suitable $\beta$. We discovered that the number of malformed examples generated by the SGLD procedure increases as training progresses. This is not apparent for models trained with $\beta = 0$, as the training instabilities prevent us from observing this. In Figure 18, we provide images generated from the model trained with $\beta = 0.000025$ at different stages of training. Note that the use of these negative examples did not lead to training divergence, and there was a phase in training where the number of malformed examples temporarily dropped to zero. As $\beta$ increases, the proportion of generated examples that are still early in their MCMC chain also increases. These examples resemble noise and exhibit low likelihood values, as illustrated in Figure 19. In terms of the best IS and FID achieved during training, increasing $\beta$ reduces performance. Since the reported IS and FID are based on the content of the replay buffer, negative examples in the buffer that are early in their MCMC chain will cause overly pessimistic estimates for larger $\beta$. However, based on visual comparisons, there is still a decrease in the quality of images as $\beta$ increases, although the difference might not be as large as IS and FID suggest.

The test classification accuracy fluctuates around similar values, with minimal improvements as $\beta$ increases. Note that the learning rate is multiplied by 0.3 in epochs 160 and 180, which accounts for the behavioral changes around these epochs. We only experiment with very small values of $\beta$. For larger values, we observed a significant gap between $f_\theta(\mathbf{x}^+)$ and $f_\theta(\mathbf{x}^-)$, which hindered effective training of the generative part of the model, as SGLD failed to produce good $\mathbf{x}^-$, similar to the experiments discussed in Section 2. For the reported values of $\beta$, there should be no noticeable effect on the SGLD procedure[12] since $1 - \beta \approx 1$, indicating that our method effectively impacts only the loss computation.

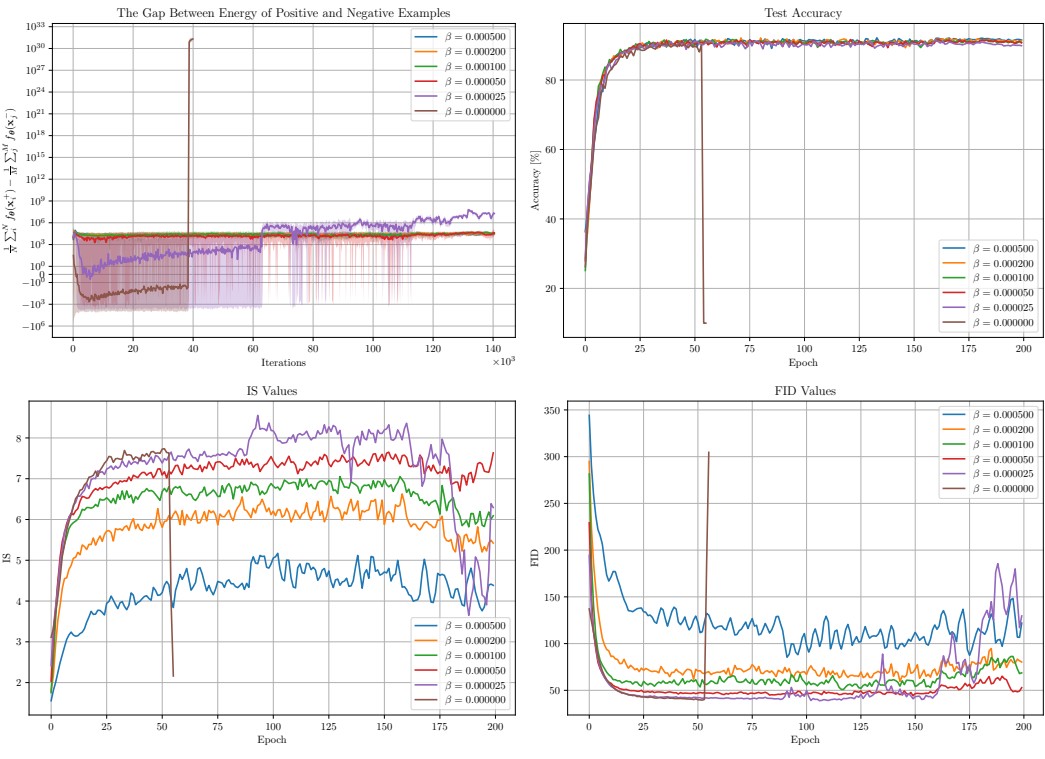

Figure 16: Training progress for the JEM in scenario 1 (default setup). We report the evolution of the difference between the mean of the positive and negative energies, test accuracy, IS, and FID.

To demonstrate that the observed behavior is not confined to the default setup, we introduce additional configurations by adjusting hyperparameters related to EBM training. Our goal is not to find the configuration with the best performance but to primarily assess the training stability under various

---

[12]Assuming the practical SGLD procedure approximates the intended theoretical distribution.

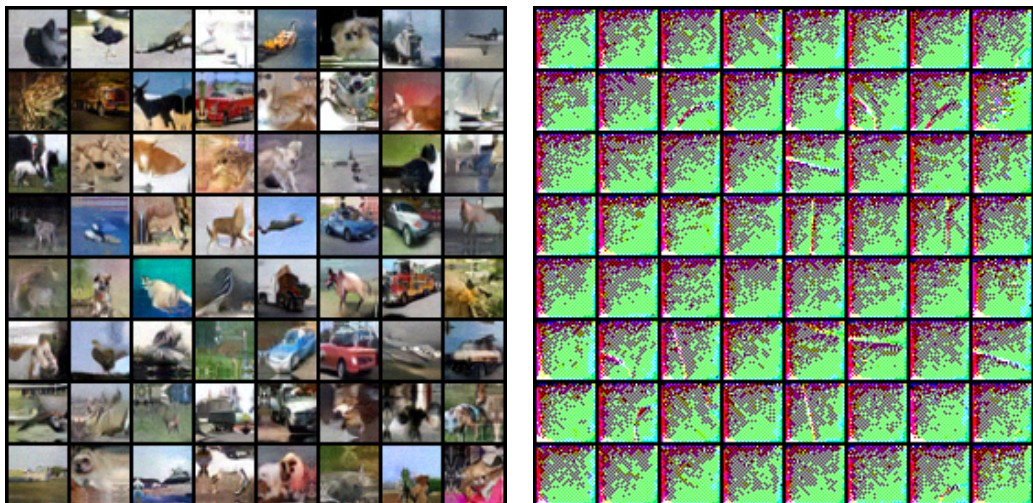

<table>
<tr><td>Epoch 54, Iteration 437</td><td>Epoch 54, Iteration 537</td></tr>
</table>

Figure 17: The default setup of JEM training with $\beta = 0$ results in a sudden change. The difference in the quality of images used as negative examples is shown 100 iterations apart.

conditions. Each configuration is referred to as a "scenario", with the default setting corresponding to the first scenario. For clarity, we number them as follows:

1. Default setup of JEM
2. Reducing the number of SGLD steps
3. Reducing the number of negative examples
4. Increasing the output scaling factor to 40,000
5. Decreasing the output scaling factor to 10,000
6. Increasing the SGLD step size $\alpha$ by a factor of 4
7. Decreasing the SGLD step size $\alpha$ by a factor of 4

The performance of the default setup is also summarized in the first line of Table 2 where we compare the best performance reached with $\beta = 0$ and $\beta \neq 0$. We report the test accuracy reached in the epoch with the best validation set performance and the best value reached throughout training for IS and FID with $\beta$ reported in Figure 16.

Table 2: Comparison of the baseline system performance with $\beta = 0$ and the best system with $\beta \neq 0$ for each metric and its corresponding $\beta$ value. We report $\beta^* = \beta/o$ rounded to 1 decimal place for better readability, where o is the output scaling factor.

| # | Setup Modification | $\beta = 0$ | | | Best Performance $\beta \neq 0$ | | | | | |
|---|---|---|---|---|---|---|---|---|---|---|
| | | ACC↑ | IS↑ | FID↓ | ACC↑ | IS↑ | FID↓ | $\beta^*_{\text{ACC}}$ | $\beta^*_{\text{IS}}$ | $\beta^*_{\text{FID}}$ |
| 1 | The default setup | 90.5% | 7.7 | 39.6 | **91.7%** | **8.6** | **38.9** | 4.0 | 0.5 | 0.5 |
| 2 | SGLD steps ($20 \rightarrow 5$) | 10.0% | 1.8 | 439.6 | **88.3%** | **7.9** | **48.0** | 2.0 | 0.5 | 0.5 |
| 3 | Neg. examples ($64 \rightarrow 8$) | 89.3% | 7.6 | **41.9** | **91.1%** | **7.9** | 42.0 | 2.0 | 0.5 | 0.5 |
| 4 | Output scale ($2\times$) | 91.0% | 7.4 | **41.7** | **92.8%** | **7.8** | 41.9 | 10.0 | 0.5 | 0.5 |
| 5 | Output scale ($0.5\times$) | 87.6% | 7.4 | 43.2 | **90.2%** | **8.2** | **41.8** | 2.0 | 0.5 | 0.5 |
| 6 | SGLD step size ($4\times$) | 36.2% | 2.9 | 154.1 | **92.4%** | **8.0** | **41.5** | 4.0 | 0.5 | 0.5 |
| 7 | SGLD step size ($0.25\times$) | 86.5% | 7.7 | **41.9** | **88.0%** | **8.1** | 42.7 | 1.0 | 0.1 | 0.1 |

In scenario 2, we reduce the number of SGLD steps from 20 to 5, with results summarized in Figure 20. The training instabilities for $\beta = 0$ were even more severe, leading to divergence in

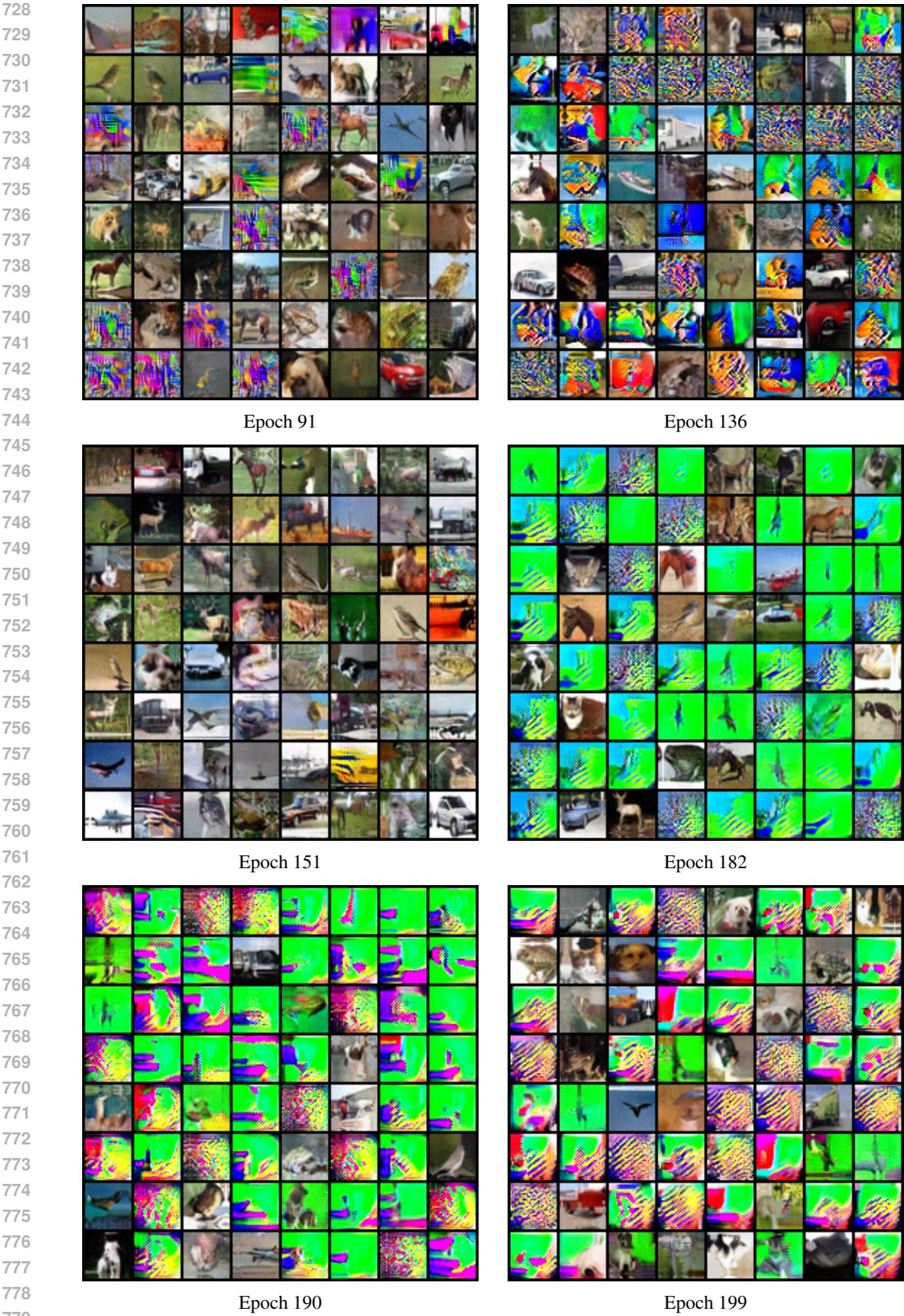

Epoch 91

Epoch 136

Epoch 151

Epoch 182

Epoch 190

Epoch 199

Figure 18: Generated images using the default JEM setup with $\beta = 0.000025$. Some images start being malformed (epoch 91) but the divergence is avoided. No malformed images are seen in epoch 151, although they reappear later. The epoch is indicated below the respective plot.

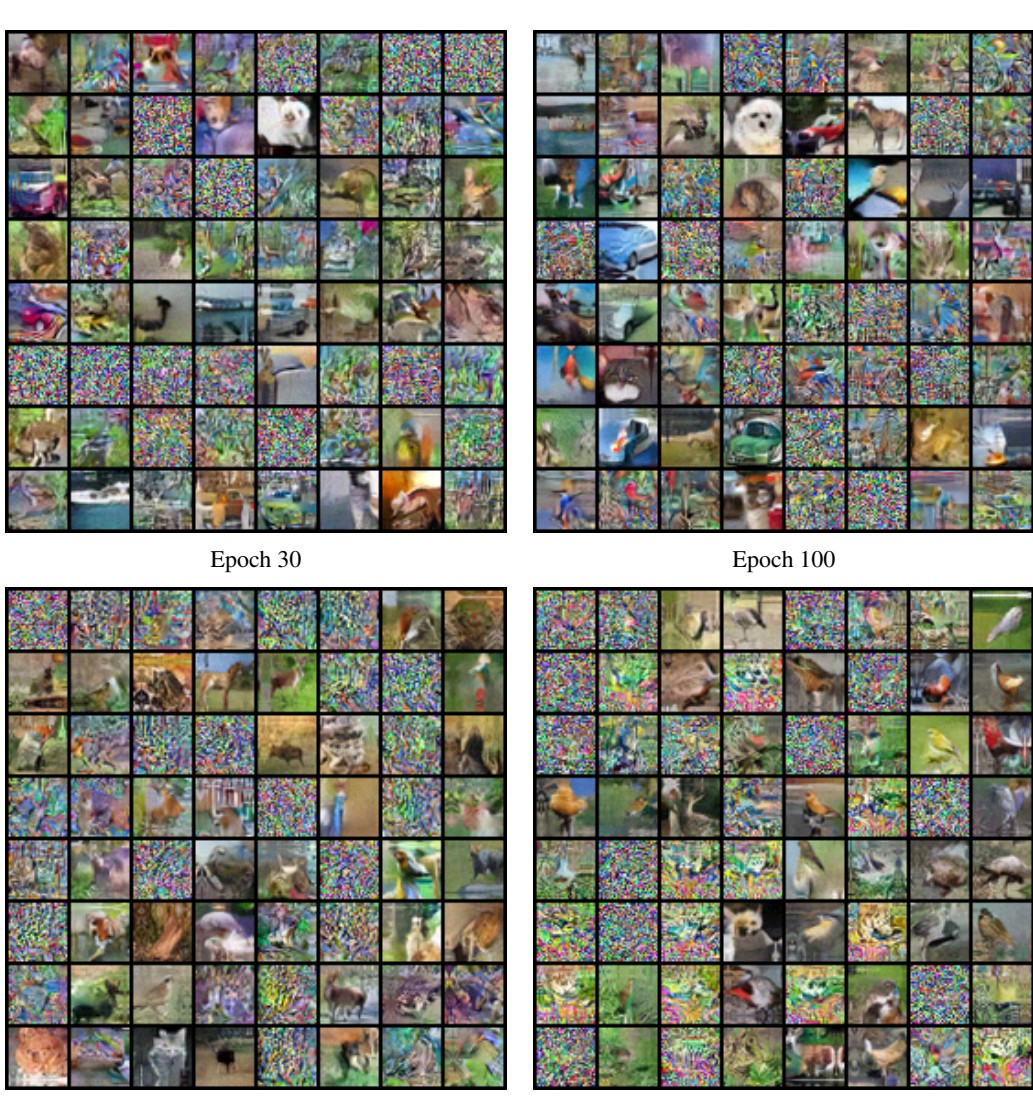

Epoch 30           Epoch 100

Epoch 150           Epoch 199

Figure 19: Quality of generated images using the default setup of JEM with $\beta = 0.0005$. As $\beta$ increases, no malformed images are presented, but some generated images appear to be still early in the MCMC chain. This effect gradually increases with increasing $\beta$. The epoch is indicated below the respective plot.

the first epoch. In scenario 3, we reduce the number of negative examples $M$ from 64 to 8, as an alternative way to limit computation. The behavior shown in Figure 21 leads to the same conclusions as in the default scenario.

To examine the effect of the output scaling factor set to 20,000, we test scenario 4 and scenario 5, adjusting it to 40,000 and 10,000, respectively. The results, shown in Figure 22 and Figure 23, both show earlier divergence for $\beta = 0$. Notably, in Figure 22, for $\beta = 0.00005$, the quality of negative examples begins to decrease rapidly around epoch 100, but the model maintains discriminative performance for over 30 epochs before eventually diverging. This also describes the behavior as $\beta$ decreases further compared to the default scenario. For small values of $\beta$, divergence is delayed but not eliminated, and as $\beta$ increases, the divergence becomes more gradual.

Finally, we adjust the SGLD step size from $\alpha = 0.0001$ to $\alpha = 0.0004$ and $\alpha = 0.000025$ in scenario 6 and scenario 7, respectively. This effectively scales the output of $f_{\theta}(\mathbf{x})$ by a factor of 4 and increases the standard deviation of the noise added in each SGLD step by a factor of 2. Figure 25 illustrates the effect of reducing the step size. For the increased step size, we also adjusted the output scaling factor from $20,000$ to $5,000$ to counteract the implicit scaling of $f_{\theta}(\mathbf{x})$ and reduced the learning rate by a factor of 4. Despite these adjustments, training with $\beta = 0$ still resulted in divergence by the third epoch, as shown in Figure 24.

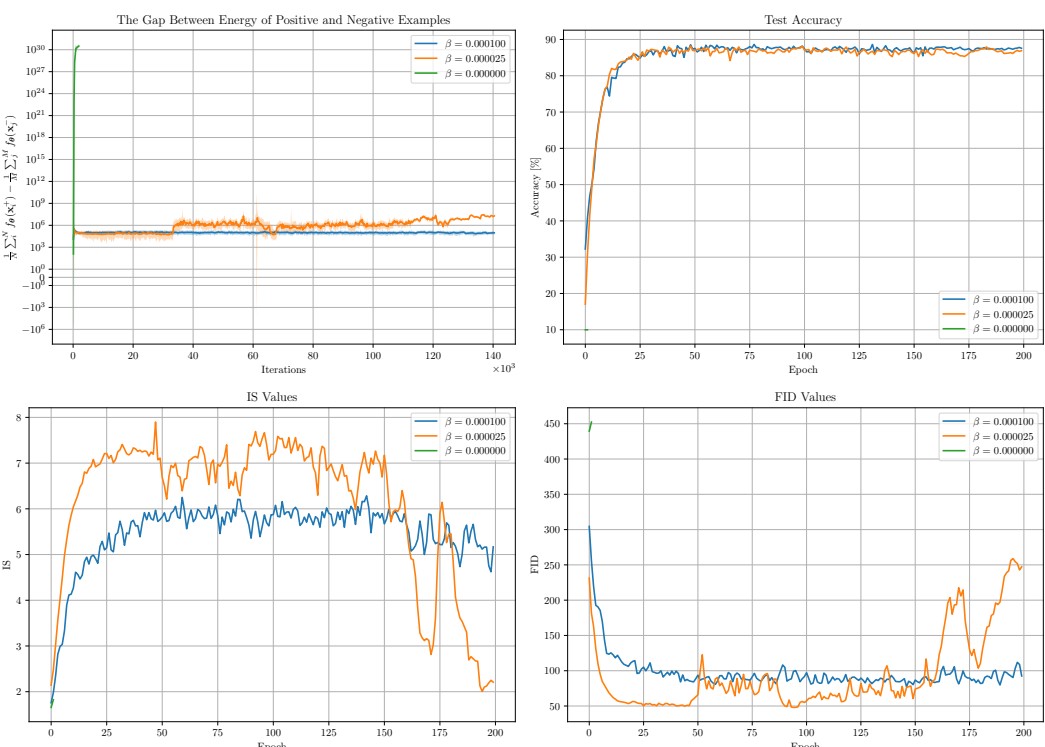

Figure 20: Training progress for the JEM in scenario 2, where 20 SGLD steps are reduced to 5. We report the evolution of the difference between the mean of the positive and negative energies, test accuracy, IS, and FID.

Based on the results in Table 2, we observe that in all scenarios, except for the default setup (where the second-largest $\beta$ achieved the highest classification accuracy), the largest $\beta$ consistently resulted in the best accuracy. In contrast, the smallest non-zero $\beta$ produced the best IS and FID scores among models trained with $\beta \neq 0$. This suggests that experimenting with even smaller $\beta$ could further improve FID, although this is not our primary goal. Notably, the test accuracy with $\beta = 0$ was the lowest across all scenarios. The results also indicate that increasing the output scaling factor improves test accuracy, even in the $\beta = 0$ setting.

The same approach is applied in JEM, where during SGLD sampling to maximize $\log p_{\theta}(\mathbf{x})$, the negative energy is scaled: $f_{\theta}(\mathbf{x}) = 20000 \log \sum_y e^{h_{\theta}(\mathbf{x})[y]}$ is used instead of $f_{\theta}(\mathbf{x}) = \log \sum_y e^{h_{\theta}(\mathbf{x})[y]}$.

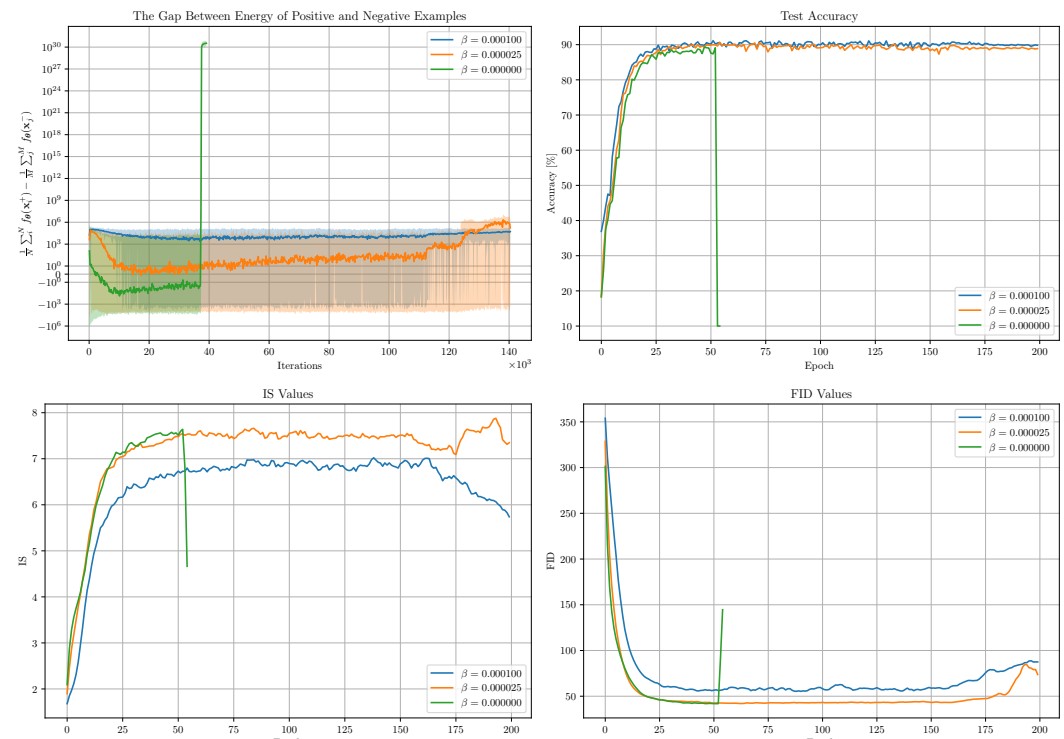

Figure 21: Training progress for the JEM in scenario 3, where 64 negative examples per update are reduced to 8. We report the evolution of the difference between the mean of the positive and negative energies, test accuracy, IS, and FID.

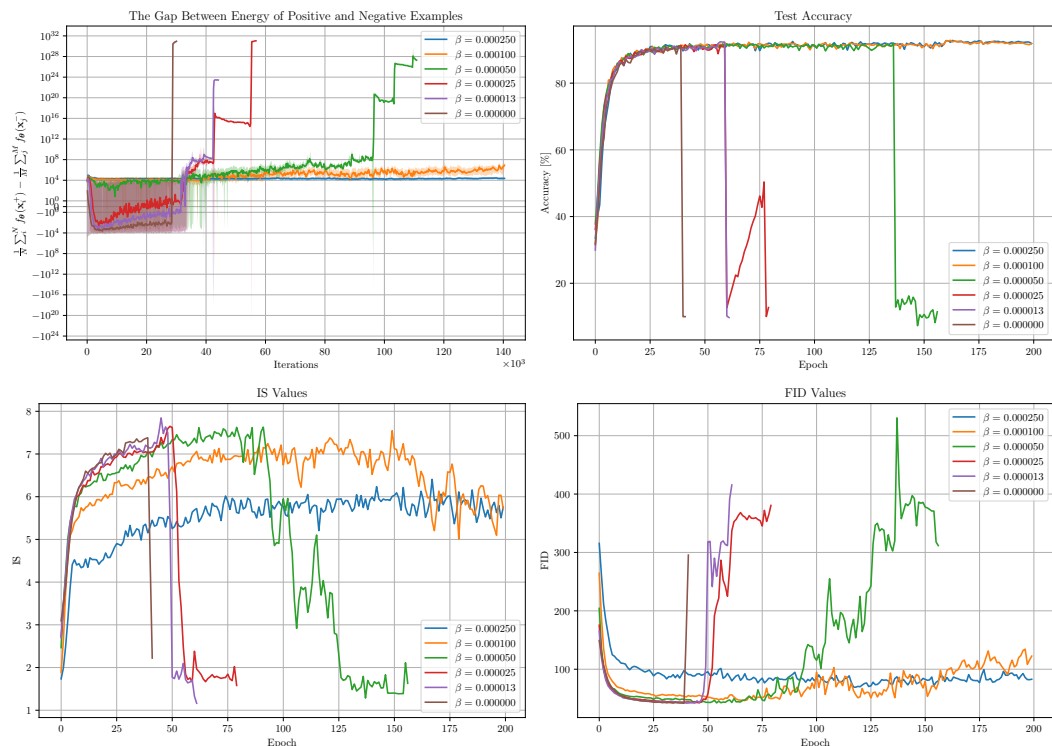

Figure 22: Training progress for the JEM in scenario 4, where the output scaling factor is set to 40,000 instead of the default 20,000. We report the evolution of the difference between the mean of the positive and negative energies, test accuracy, IS, and FID.

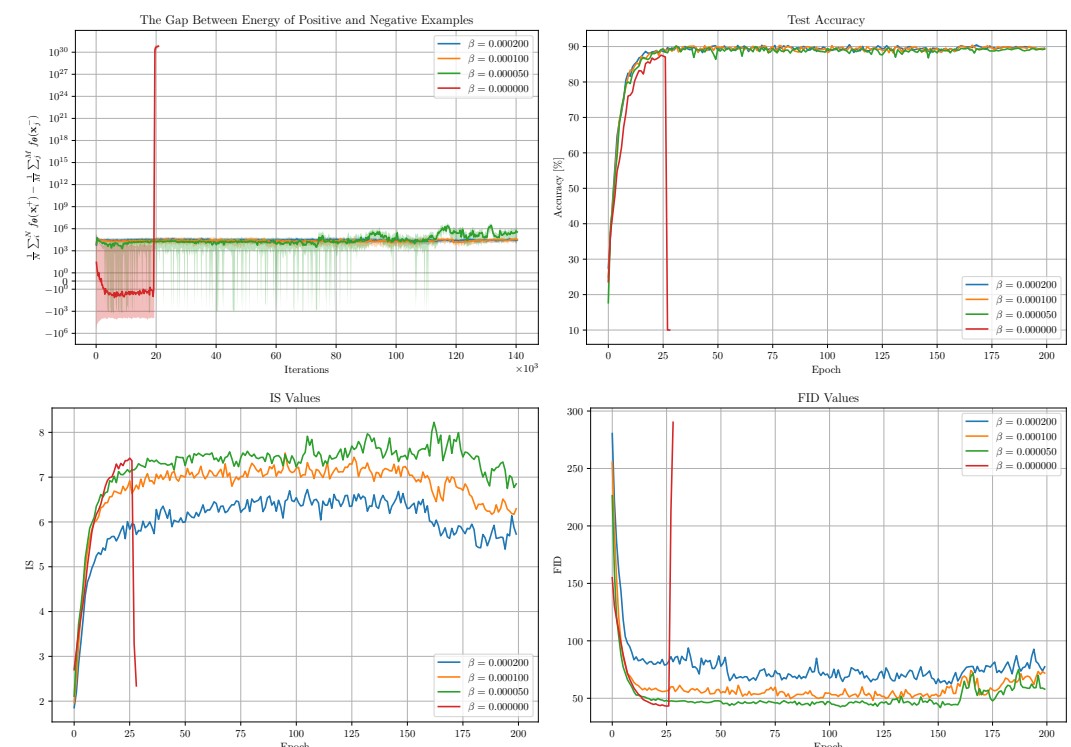

Figure 23: Training progress for the JEM in scenario 5, where the output scaling factor is set to 10,000 instead of the default 20,000. We report the evolution of the difference between the mean of the positive and negative energies, test accuracy, IS, and FID.

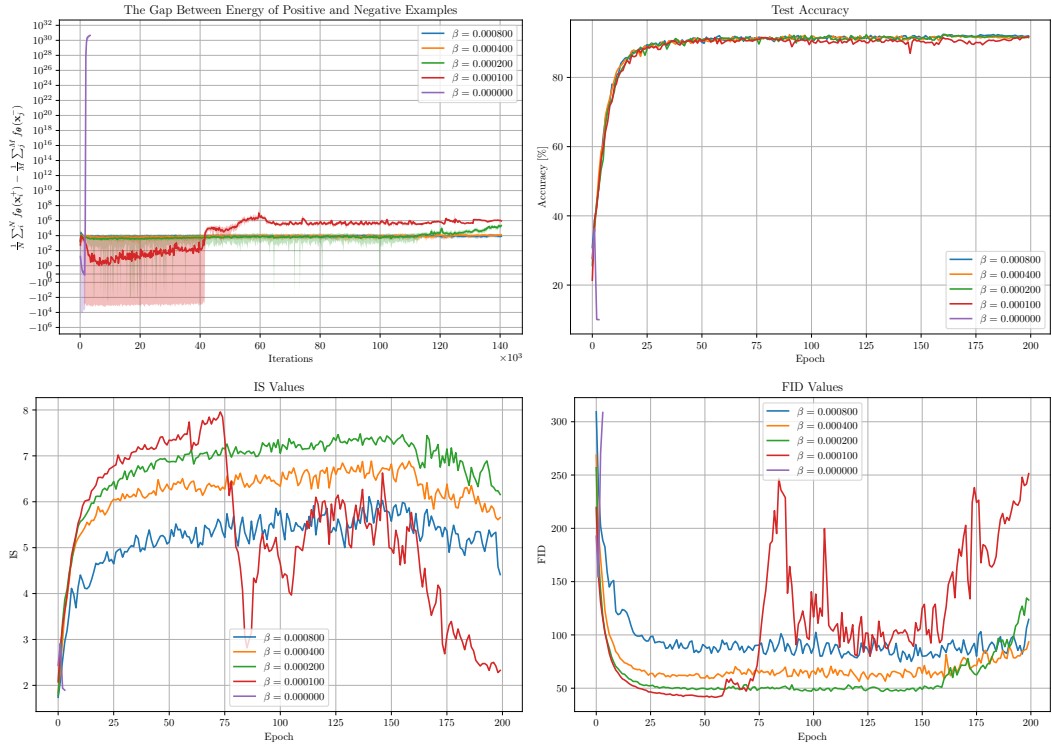

Figure 24: Training progress for the JEM in scenario 6, where SGLD step size $\alpha$ is scaled up by a factor of 4. We report the evolution of the difference between the mean of the positive and negative energies, test accuracy, IS, and FID.

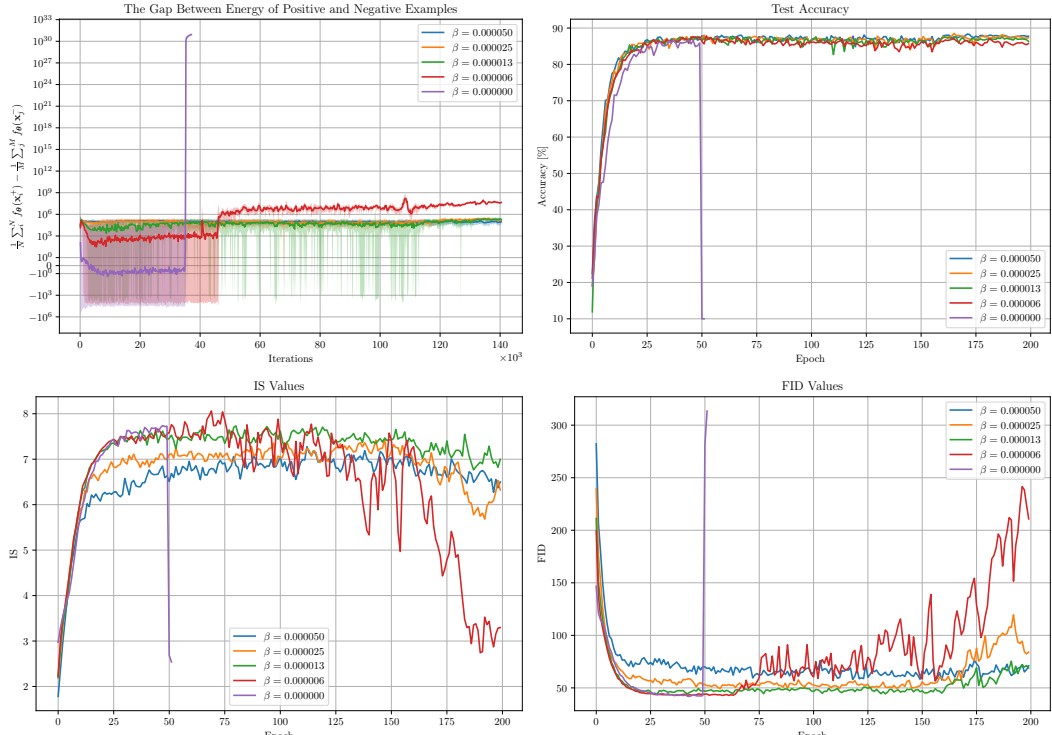

Figure 25: Training progress for the JEM in scenario 7, where SGLD step size $\alpha$ is scaled down by a factor of 4. We report the evolution of the difference between the mean of the positive and negative energies, test accuracy, IS, and FID.

## M.2 PRACTICAL IMPLEMENTATION

In Appendix G, we explain that, in practice, the amount of noise used in the SGLD procedure is reduced, which has minimal impact on EBM training. This change corresponds to a different parametrization of $f_{\boldsymbol{\theta}}(\mathbf{x})$, with $p_{\boldsymbol{\theta}}(\mathbf{x}) \propto e^{\mathrm{o}\, f(\mathbf{x})}$. We denote o as the output scaling factor, typically set to 20,000. The introduction of o has a more significant effect in the context of JEM. Consequently, the SGLD procedure generates samples based on the negative energy $f_{\boldsymbol{\theta}}(\mathbf{x}) = \mathrm{o}\log\sum_y e^{h_{\boldsymbol{\theta}}(\mathbf{x})[y]}$, instead of $f_{\boldsymbol{\theta}}(\mathbf{x}) = \log\sum_y e^{h_{\boldsymbol{\theta}}(\mathbf{x})[y]}$. The corresponding objective for $\log p_{\boldsymbol{\theta}}(\mathbf{x})$ then becomes

$$\nabla_{\boldsymbol{\theta}}\mathbb{E}_{p_d(\mathbf{x})}\left[\log p_{\boldsymbol{\theta}}(\mathbf{x})\right] \approx \frac{1}{N}\sum_i^N \nabla_{\boldsymbol{\theta}}\, \mathrm{o}\log\sum_y e^{h_{\boldsymbol{\theta}}(\mathbf{x}_i^+)[y]} - \frac{1}{M}\sum_j^M \nabla_{\boldsymbol{\theta}}\, \mathrm{o}\log\sum_y e^{h_{\boldsymbol{\theta}}(\mathbf{x}_j^-)[y]}. \quad (40)$$

However, the JEM implementation omits scaling by $\mathrm{o} = 20,000$. The introduction of o has two consequences. First, the reduction in SGLD noise alters the negative energy of $p_{\boldsymbol{\theta}}(\mathbf{x}, y)$ from $h_{\boldsymbol{\theta}}(\mathbf{x})[y]$ to $(\mathrm{o}-1)\log\sum_y e^{h_{\boldsymbol{\theta}}(\mathbf{x})[y]} + h_{\boldsymbol{\theta}}(\mathbf{x})[y]$. This can be derived from

$$\log p_{\boldsymbol{\theta}}(\mathbf{x}) + \log p_{\boldsymbol{\theta}}(y\mid\mathbf{x}) = \mathrm{o}\log\sum_y e^{h_{\boldsymbol{\theta}}(\mathbf{x})[y]} + h_{\boldsymbol{\theta}}(\mathbf{x})[y] - \log\sum_y e^{h_{\boldsymbol{\theta}}(\mathbf{x})[y]} + c, \quad (41)$$

where $c$ is a constant independent of $\mathbf{x}$ and $y$. Second, instead of maximizing $\log p_{\boldsymbol{\theta}}(\mathbf{x}, y)$, the objective becomes $1/\mathrm{o}\log p_{\boldsymbol{\theta}}(\mathbf{x}) + \log p_{\boldsymbol{\theta}}(y\mid\mathbf{x})$.

In our first experiment, we attempted to remove the scaling factor of 20,000 by setting it to $\mathrm{o} = 1$. However, this caused $\nabla_{\mathbf{x}}f_{\boldsymbol{\theta}}(\mathbf{x})$ to become too small, leading to ineffective movement during the SGLD procedure. Similarly, applying the scaling factor of 20,000 to $h_{\boldsymbol{\theta}}(\mathbf{x})[y]$ instead of $f_{\boldsymbol{\theta}}(\mathbf{x})$, which would only represent a different parameterization of JEM, resulted in poor discriminative performance. Intuitively, the SGLD procedure requires rapid changes in $f_{\boldsymbol{\theta}}(\mathbf{x})$, while maintaining uncertainty in $p_{\boldsymbol{\theta}}(y\mid\mathbf{x})$. Using the default setup with $\mathrm{o} = 20,000$ achieves this balance. Although

all experiments were conducted with small values of $\beta$, it is important to recognize that when $\beta = 1/20000 = 0.00005$, the reweighting becomes proportional to $\sum_y e^{h_{\boldsymbol{\theta}}(\mathbf{x})[y]}$.

The proposed method is incorporated into JEM by introducing the following modifications:

- The negative energy in the SGLD procedure is scaled by $1 - \beta$, i.e., we use $f_{\boldsymbol{\theta}}(\mathbf{x}) = (1 - \beta) \circ \log \sum_y e^{h_{\boldsymbol{\theta}}(\mathbf{x})[y]}$

- The calculation of the $\log p_{\boldsymbol{\theta}}(\mathbf{x})$ loss is based on Equation 8 rather than Equation 3.

