# OpenReview forum: "Energy-based Model Training Objective Robust to Inaccurate SGLD Samples"
_ICLR.cc/2025/Conference — Submitted to ICLR 2025_

### Official Review · Reviewer_hvyA · 2024-10-28

**Soundness:** 2
**Presentation:** 2
**Contribution:** 2
**Rating:** 3
**Confidence:** 5

**Summary:**

The present paper proposes a new training strategy for the maximum likelihood training (MLE) of Energy based Models. Namely,  the gradient of the MLE objective are estimated by combining a Langevin sampling of a “higher temperature” version of the model, and a self normalized importance sampling reweighting to recover an expectation according to beta=1. An additional empirical modification is done to bypass the importance sampling estimate when the proposed negative samples it relies on have very low likelihood according to the model.

Numerical results are presented on toy 2d systems, as well as for a variant called joint EBM on CIFAR10.

**Strengths:**

- The papers’ motivation, how to train an EBM with accurate likelihood, is a challenge relevant to the ICLR community.
- The paper honestly discussed experiments with negative results.

**Weaknesses:**

- The approach proposed in the paper lacks some theoretical grounding. First, for the self-normalized importance sampling estimator to be correctly implemented, the sampling procedure from the proposal $p_\theta^{1-\beta}$ distribution should be exact. Here the authors rely on a Langevin dynamics, which has discretization error (which can be moderate), but also still suffers from slow mixing if $p_\theta^{1-\beta}$ is multimodal. Second, there is a no thorough assessment in the main text of the impact of adding a positive sample to the negative samples, beyond the fact of effectively discarding all the negative samples in this case, which arguably does not yield a highly quality estimator of the MLE gradients.

- The approach proposed does not solve the issue of sampling the EBM once trained.

- The numerical results are limited and moderately convincing. If the phenomenology expected by the authors is present for the 8 Gaussian example, the algorithm does not appear to reproduce robustly the relative weights of the modes in Rings. This shortcoming is not discussed in the paper. The results in Table 1 do not have error bars, making it hard to asses their robustness/significance.

- The discussion of the Related works is incomplete, there is no section properly dedicated to it. In particular I would advise the authors to comment on other works attempting MLE training of EBM [1,2,3] and this work  [4] investigating the impact of non-mixing sampling in the EBM sampling.

- The writing of the paper needs to be improved.
	- Some statements lacks precisions or justification:
		- “Sampling from a more uncertain distribution can lead to improved mixing.” L193
		- “Notably, these two values do not necessarily need to sum up to 1; arbitrary values can be employed instead, corresponding to a different parameterization of the EBM.” L296 —> what would then be the justification?
	- A lot of arguments that the author seek to make to justify the approach are moved to appendix while some less interesting implementation details are kept in the main text.  Half a page is dedicated to explaining experiments that fail while the setting of the JEBM experiment, which is probably a positive result the author want to emphasize, is not in the main text.

[1] Grenioux, Louis, Eric Moulines, and Marylou Gabrié. “Balanced Training of Energy-Based Models with Adaptive Flow Sampling.” In ICML 2023 Workshop on Structured Probabilistic Inference {\&} Generative Modeling, 2023. https://openreview.net/forum?id=AwJ2NqxWlk&referrer=%5BAuthor%20Console%5D(%2Fgroup%3Fid%3DICML.cc%2F2023%2FWorkshop%2FSPIGM%2FAuthors%23your-submissions).

[2] Béreux, Nicolas, Aurélien Decelle, Cyril Furtlehner, and Beatriz Seoane. “Learning a Restricted Boltzmann Machine Using Biased Monte Carlo Sampling.” SciPost Physics 14, no. 3 (March 14, 2023): 032. https://doi.org/10.21468/SciPostPhys.14.3.032.

[3] Carbone, Davide, Mengjian Hua, Simon Coste, and Eric Vanden-Eijnden. “Efficient Training of Energy-Based Models Using Jarzynski Equality.” Advances in Neural Information Processing Systems 36 (December 15, 2023): 52583–614.

[4] Agoritsas, Elisabeth, Giovanni Catania, Aurélien Decelle, and Beatriz Seoane. “Explaining the Effects of Non-Convergent Sampling in the Training of Energy-Based Models.” arXiv, January 23, 2023. https://doi.org/10.48550/arXiv.2301.09428.

**Questions:**

Minor:
- There are quite a few misprints at the end of the introduction.
- Why the authors use the term SGLD? I do not believe that the gradients are stochastically estimated, they can be exactly computed with autodiff. A maybe more appropriate denomination would be ULA (Unadjusted Langevin dynamics).

---

> ### Author Response · Authors · 2024-11-27
> **Reply to Reviewer hvyA 1/2**
>
> 1. The approach proposed in the paper lacks some theoretical grounding. First, for the self-normalized importance sampling estimator to be correctly implemented, the sampling procedure from the proposal
> distribution should be exact. Here the authors rely on a Langevin dynamics, which has discretization error (which can be moderate), but also still suffers from slow mixing if
> is multimodal. Second, there is a no thorough assessment in the main text of the impact of adding a positive sample to the negative samples, beyond the fact of effectively discarding all the negative samples in this case, which arguably does not yield a highly quality estimator of the MLE gradients.
>    - We propose a generalized loss as an alternative to the standard loss (a particular case of generalized loss, when $\beta=0$). In both cases, the expectation over the modeled distribution is approximated using the same approach, and we believe that this provides exactly the same theoretical grounding. We justify how it can effectively deal with "failed" samples by de-weighting their contributions. We summarize the impact of adding a positive example to the negative examples in Section 3.2. Even though reviewers are not required to read the Appendix, we do not consider the fact that the detailed derivation and discussion are placed in the Appendix rather than the main text to be the weakness of this work. Effectively discarding all negative examples can be reasonable in some cases. As an example, imagine the standard training using mini-batches. Some mini-batches might contain all negative examples that correspond to failed instances, while others contain only some failed cases. We want to discard mini-batches with all failed instances. Another example is JEM, where multiple objective functions are optimized simultaneously, and in case of generating all failed cases, we would effectively want to discard them.
>
> 2. The numerical results are limited and moderately convincing. If the phenomenology expected by the authors is present for the 8 Gaussian example, the algorithm does not appear to reproduce robustly the relative weights of the modes in Rings. This shortcoming is not discussed in the paper. The results in Table 1 do not have error bars, making it hard to asses their robustness/significance.
>    - The paper's main point is that the stability and density improve as $\beta$ increases, which we show in all presented cases using 2D toy data. Our work focuses on instances where the sampler provides improper samples. In that case, we show that if the training converges, we approximately learn the desired distribution plus some biased reflecting bias introduced by the sampling procedure as discussed in Appendix I. As stated in the paper, our work mainly assesses the performance using this biased sampling procedure. Improving Langevin dynamics (better initial distribution, more steps, better-suited step size) improves the results. We also explain why relative weights were not appropriately learned in Figure 5; see line 1054. With biased sampling, the training will always result in bias in learned densities. At the same time, $\beta$ controls whether you prefer less biased distribution or less biased samples (bias with respect to the distribution of training data). We do not provide error bars, as the goal of that experiment is not to reach the best performance but to demonstrate improved stability. We consider degradation in performance at the cost of enhanced stability to be an acceptable result. We believe that the presented improved performance is caused by the standard MLE training diverging before reaching the best performance.
> 3. The discussion of the Related works is incomplete, there is no section properly dedicated to it. In particular I would advise the authors to comment on other works attempting MLE training of EBM [1,2,3] and this work [4] investigating the impact of non-mixing sampling in the EBM sampling.
>    - Thank you for pointing that out; we address that in the [Related literature] section above.
>
> 4. Some statements lacks precisions or justification: “Sampling from a more uncertain distribution can lead to improved mixing.” L193
>    - We thought it was an intuitive statement following the preceding discussion and imagining it for simple examples, such as a mixture of 2 Gaussians. Increasing $\beta$ can be approximately understood as increasing the variance, which improves the mixing. We will remove this statement.

---

> > ### Author Response · Authors · 2024-11-27
> > **Reply to Reviewer hvyA 2/2**
> >
> > 5. Some statements lacks precisions or justification: “Notably, these two values do not necessarily need to sum up to 1; arbitrary values can be employed instead, corresponding to a different parameterization of the EBM.” L296 —> what would then be the justification?
> >    - We understand that reviewers are not required to review the Appendix. Still, the following sentence states: "We derive this in Appendix G, based on an analysis of key aspects of the practical SGLD sampler, including its extension to $β \ne 0$ settings". In short, as a result, we will train EBM with an adjusted learning rate and different parameterization, e.g., instead of $p_{\theta}(x) \propto exp(f_{\theta}(x))$, we will have $p_{\theta}(x) \propto exp(2f_{\theta}(x))$ or $p_{\theta}(x) \propto exp(0.5f_{\theta}(x))$.
> >
> > 6. A lot of arguments that the author seek to make to justify the approach are moved to appendix while some less interesting implementation details are kept in the main text. Half a page is dedicated to explaining experiments that fail while the setting of the JEBM experiment, which is probably a positive result the author want to emphasize, is not in the main text.
> >    - We find section 3.2 to encapsulate a substantial contribution of our work; however, it is impossible to fit all analyses and arguments in the main paper. We provide the arguments in the main paper and reference parts of the Appendix that justify it and give more details. We discuss the importance of section Section 4.2 in the [Section 4.2] section above. As stated earlier, we cannot fit everything in the main text, but we discuss the most important results regarding JEM in the main text (Section 4.3), which occupies one full page.
> >
> >
> > Questions:
> > 1. There are quite a few misprints at the end of the introduction.
> >    - Thank you, we corrected the misprints.
> > 2. Why the authors use the term SGLD? I do not believe that the gradients are stochastically estimated, they can be exactly computed with autodiff. A maybe more appropriate denomination would be ULA (Unadjusted Langevin dynamics).
> >    - Thank you for pointing that out; we were following the literature on JEM, which denotes it as SGLD ([1],[2],[3]). After a careful review, we agree that ULA is the more suitable name, and we will replace all occurrences of SGLD with ULA.
> >
> >
> > [1] Grathwohl et al. Your classifier is secretly an energy-based model, and you should treat it like one, ICLR 2020
> >
> > [2] Yang, Xiulong, and Shihao Ji. "Jem++: Improved techniques for training jem." Proceedings of the IEEE/CVF International Conference on Computer Vision. 2021.
> >
> > [3] Yang, Xiulong, Qing Su, and Shihao Ji. "Towards Bridging the Performance Gaps of Joint Energy-based Models." Proceedings of the IEEE/CVF Conference on Computer Vision and Pattern Recognition. 2023.

---

### Official Review · Reviewer_WiqD · 2024-10-31

**Soundness:** 2
**Presentation:** 2
**Contribution:** 3
**Rating:** 5
**Confidence:** 4

**Summary:**

This paper revisits the issue of non-converged Stochastic Gradient Langevin Descent introducing biases in classical contrastive divergence and persistent contrastive divergence training of energy-based models. The paper makes two contributions: Firstly, debiasing of the parameter gradient through a decomposition of the model $p_\theta(\mathbf x)$ into two tempered distributions $q_\theta(\mathbf x) = p_\theta^{1-\beta}(\mathbf x)$ and  $r_\theta(\mathbf x) = p_\theta^{\beta}(\mathbf x)$ and subsequent self normalised importance sampling. Secondly, the paper includes a positive sample in the set of contrastive negative samples to stabilise training. The paper demonstrates stabilising effects but also comments on deprecating sample quality for large $\beta$.

**Strengths:**

- The derivations are correct
- The factorisation of the model into an importance distribution and an importance ratio is an interesting idea to improve algorithms that involve self-sampling from the model.
- I appreciate the honesty in reporting a deprecation of sample quality when $\beta>0$ is used. This reflects that the authors are sampling from a tempered, i.e. smoothed out model distribution, and shows that the approach taken by the authors demands a trade-off between training stability and sample quality, at least for high-dimensional distributions.
- The stabilisation can be used in any self-sampling based training method for energy-based models, and can thus be impactful if executed well.

**Weaknesses:**

- The experiments on toy data sets are missing a good baseline that helps putting the results in context. For example, I would expect an experimental result of contrastive divergence with the standard regularisation $f_\theta(x_+)^2 + f_\theta(x_-^2)$ for comparison which I know to produce okay results on toy data. (see, e.g. [1] for details on the stabilisation term)
- On image data, only very small values of the stabilisation parameter $\beta$ actually yield stable training of the EBM, thus changing the standard EBM training method minimally. Consequently, the stabilisation of JEM is only demonstrates marginal improvements of the generative (in terms of FID) and discriminative model (in terms of accuracy) over the base training method used.
- The work is missing a related work section to put this work into a broader context of stabilisation tricks for EBM training. For example, the biases of contrastive divergence have also been targeted by [2]. The trick of including a positive sample to the set of negative samples has been explored before. The trick has been used to stabilise EBM training in [3]. For example, equation 21 in the appendix in your paper closely resembles [3], section 4. The trick is also known in prior contrastive estimation [4] for Bayesian experimental design.

[1] Du, Yilun and Mordatch, Igor: Implicit Generation and Modeling with Energy-Based Models, NeurIPS 2019

[2] Du et al. Improved Contrastive Divergence Training of Energy Based Models, ICML 2021

[3] Schroeder et al. Energy Discrepancies: A Score-independent loss for energy-based models (see section 4)

[4] Foster et al. A Unified Stochastic Gradient Approach to Designing Bayesian-Optimal Experiments. PMLR 2020 (see equation 12)

**Questions:**

- Have you experimented with negative $\beta$ values? This is justified since the importance ratio does not need to be a distribution. I would be particularly curious about this for image data, where values of $\beta>0$ lead to noisy samples in the replay buffer. (you could also switch the factorisation to $q_\theta \propto \exp(\beta f_\theta))$ and choose $\beta\in \mathbb R_{\geq 0}$, which fits more closely to notations in statistical physics).
- Another reason to consider negative $\beta$ is the fact that [5] achieves good results by performing Langevin dynamics with small noise, effectively sampling from a negatively tempered distribution. This approach could potentially be debiased with your proposed methodology.

[5] Grathwohl et al. Your classifier is secretly an energy-based model, and you should treat it like one, ICLR 2020

---

> ### Author Response · Authors · 2024-11-27
> **Reply to Reviewer WiqD 1/2**
>
> 1. The experiments on toy data sets are missing a good baseline that helps putting the results in context. For example, I would expect an experimental result of contrastive divergence with the standard regularisation ($\cdot$) for comparison which I know to produce okay results on toy data. (see, e.g. [1] for details on the stabilisation term)
>
>    - We extended the existing recipe from [6], which we consider a standard setup used by many other works. Our baseline from [6] does not use the regularization you propose, and we are unaware of any other work that would compare learned 2D toy data densities using such regularization. As an example, we can even use a work that you refer to in your review [3], but also others, such as [7]. Additionally, we are comparing the effect of the loss function, while any effect of the regularizer would be orthogonal to this. Next, [1] employed this regularization together with spectral normalization. Still, they claim: "During a typical training run, we keep training until the sampler is unable to generate effective samples (when energies of proposal samples are much larger than energies of data points from the training data set)." suggesting that the proposed method does not entirely avoid the problem. [5] reported that they did not find a setting that would help stabilize the training and, at the same time, did not significantly hurt the performance when experimented with these regularizations. [8] reported that adding L2 regularization, on the contrary, caused instability in a particular model.
>
> 2. On image data, only very small values of the stabilisation parameter actually yield stable training of the EBM, thus changing the standard EBM training method minimally. Consequently, the stabilisation of JEM is only demonstrates marginal improvements of the generative (in terms of FID) and discriminative model (in terms of accuracy) over the base training method used.
>    - On the contrary, as demonstrated in the experimental part, the effect of even very small values of $\beta$ significantly affects the optimization, which is the main reason we only experiment with small values. Moreover, stable training is achieved even with larger values of $\beta$. At the same time, we do not consider that already very small values of $\beta$ are effective in preventing the training instabilities to be a weakness of this work. The main point of experiments on JEM is to demonstrate that we introduced an effective way of avoiding training divergence rather than to improve FID, IS, or accuracy. As discussed in our work, we expect worse performance in terms of FID or IS. Improved FID/IS are only the consequence of the fact that with $\beta=0$, the training diverged too early. We further trained JEM with more restricted resources (Table 2), demonstrating how significant problem the training instabilities are for JEM trained with the standard MLE loss ($\beta=0$).
> 3. The work is missing a related work section to put this work into a broader context of stabilisation tricks for EBM training. For example, the biases of contrastive divergence have also been targeted by [2]. The trick of including a positive sample to the set of negative samples has been explored before. The trick has been used to stabilise EBM training in [3]. For example, equation 21 in the appendix in your paper closely resembles [3], section 4. The trick is also known in prior contrastive estimation [4] for Bayesian experimental design.
>    - We agree that we should have included more related work and addressed that in the [Related literature] section above. However, we disagree that the "trick of including a positive sample into negative samples" has been used to stabilize EBM training. In [3], the context of what you call positive and negative samples is quite different. At the same time, they do not include a positive sample in negative ones but only use the scaled negative energy of these samples. As far as we understand, in [4], the additional "good sample" is added into the denominator, which is still quite different from what we do, as we simultaneously include it in the numerator. Because of the context of these works and the specificity of each approach, we believe the connection between them is too weak to be considered related.

---

> > ### Author Response · Authors · 2024-11-27
> > **Reply to Reviewer WiqD 2/2**
> >
> > Questions:
> > 1. Have you experimented with negative values? This is justified since the importance ratio does not need to be a distribution. I would be particularly curious about this for image data, where values of lead to noisy samples in the replay buffer. (you could also switch the factorisation to and choose, which fits more closely to notations in statistical physics). Another reason to consider negative is the fact that [5] achieves good results by performing Langevin dynamics with small noise, effectively sampling from a negatively tempered distribution. This approach could potentially be debiased with your proposed methodology.
> >
> >    - If we understand that correctly, you are suggesting substituting $\beta$ for $1-\beta$. It is possible, but it would complicate all equations, which we consider to negatively affect the readability. Moreover, $\beta$ (or $1-\beta$) is theoretically not restricted to only positive values, as you suggest. Even negative values could be considered; however, the idea is that we want to de-weight negative examples with low $p_{\theta}(x)$. Using a negative value of $\beta$ will result in the opposite effect (increased weight for samples with lower $p_{\theta}(x))$. We believe that the last paragraph in the conclusion section addresses the issue of negative values of $\beta$, see L529: "In this work, we addressed the issue of samplers frequently producing samples with low $f_θ (x)$. Similarly, if a sampler tends to produce samples with excessively high $f_θ (x^–)$ values (e.g., sampling from ∝ $p_θ (x^–)^4)$, the proposed approach could be adapted by using negative values of β." Based on your request, we ran experiments with JEM and negative values of $\beta$. Based on the epochs, when the divergence occurs, it follows the paper's narrative that smaller $\beta$ values result in less stable training even when extending to negative $\beta$ values. Here are the values of negative $\beta$ and the corresponding epoch when the training diverged. Note that, as discussed in the paper, we perform two extra epochs after the divergence occurs and start with epoch 0, i.e., epoch 2 means that the divergence occurred already in the first epoch of the training.
> >
> >      - $\beta= 0.000 \rightarrow$ ep. 56
> >      - $\beta=-0.005 \rightarrow$ ep. 31
> >      - $\beta=-0.001 \rightarrow$ ep. 65
> >      - $\beta=-0.002 \rightarrow$ ep. 26
> >      - $\beta=-0.050 \rightarrow$ ep. 2
> >
> >    - Unfortunately, the argument with JEM using negatively tempered distribution is not entirely precise. Sampling from negatively tempered distribution but incorporating the standard MLE procedure corresponds to the proper training of EBM parameterized differently with a modified learning rate. We explain this in detail in Appendix G, specifically in G.1. Regarding JEM, it is discussed in Appendix M.2 (in the new version) and Appendix M.1 in the submitted version.
> >
> >
> >
> > [1] Du, Yilun and Mordatch, Igor: Implicit Generation and Modeling with Energy-Based Models, NeurIPS 2019
> >
> > [2] Du et al. Improved Contrastive Divergence Training of Energy Based Models, ICML 2021
> >
> > [3] Schroeder et al. Energy Discrepancies: A Score-independent loss for energy-based models (see section 4)
> >
> > [4] Foster et al. A Unified Stochastic Gradient Approach to Designing Bayesian-Optimal Experiments. PMLR 2020 (see equation 12)
> > Questions:
> >
> > [5] Grathwohl et al. Your classifier is secretly an energy-based model, and you should treat it like one, ICLR 2020
> >
> > [6] Nijkamp, Erik, et al. "On the anatomy of mcmc-based maximum likelihood learning of energy-based models." Proceedings of the AAAI Conference on Artificial Intelligence. Vol. 34. No. 04. 2020.
> >
> > [7] Duvenaud, David, et al. "No MCMC for me: Amortized samplers for fast and stable training of energy-based models." International Conference on Learning Representations (ICLR). 2021.
> >
> > [8] Yang, Xiulong, Qing Su, and Shihao Ji. "Towards Bridging the Performance Gaps of Joint Energy-based Models." Proceedings of the IEEE/CVF Conference on Computer Vision and Pattern Recognition. 2023.

---

> > > ### Comment · Reviewer_WiqD · 2024-12-02
> > >
> > > I appreciate the effort the authors put into their response. I understand that EBM training is typically unstable and that the goal of this paper is not primarily to improve the performance of the learned EBM, but to stabilise the training dynamics. Thank you for the investigation of negative beta values, too. I understand the potential of this work better, now.
> > >
> > > Some comments on the authors response:
> > > >Our baseline from [6] does not use the regularization you propose, and we are unaware of any other work that would compare learned 2D toy data densities using such regularization.
> > >
> > > The code in [3] uses L2 regularisation in the CD loss, even though it seems to not be stated in the paper explicitely. However, I can understand the authors' point that the stabilisation technique proposed here is supposed to be orthogonal to L2 regularisation.
> > >
> > > >  At the same time, they do not include a positive sample in negative ones but only use the scaled negative energy of these samples.
> > >
> > > This may seem so because of various equivalent forms to express this idea mathematically. Rewriting equation (21) it can be seen that for each data point
> > >
> > > $$
> > > \begin{aligned}
> > > \beta f_\theta(x_+^i) - \log\sum_{j = 1}^{M+1} \exp(\beta f_\theta(x_-^j)) &= \log \exp(\beta f_\theta(x_+^i)) - \log\sum_{j = 1}^{M+1} \exp(\beta f_\theta(x_-^j))  \\\\
> > > &= -\log\left(\sum_{j = 1}^{M+1} \exp(\beta (f_\theta(x_-^j) - f_\theta(x_+^i)))\right) \\\\
> > > & = -\log\left(1 + \sum_{j = 1}^{M} \exp(\beta (f_\theta(x_-^j) - f_\theta(x_+^i)))\right)
> > > \end{aligned}
> > > $$
> > >
> > > where in the last equality it was used that one of the negative energy contributions $f_\theta(x_-^j)$ corresponds to the positive contribution $f_\theta(x_+^i))$, exactly. This then produces the same structure as in [3], once we identify $U_\theta = -f_\theta$. So in my opinion there are interesting connections between the two stabilisation methods.
> > >
> > > Since the authors seem to be interested in connections to discriminative training, the following work may also be interesting, making connections of a similar kind: Omer & Michaeli, Contrastive Divergence is a Time Reversal Adversarial Game (ICLR 2021)
> > >
> > > I apologise for not leaving enough time to respond to my new comments. I will increase my score to acknowledge the potential of this work in stabilising EBM and JEM training, and since it brings attention to undesirable restarts of JEM training.

---

> > > > ### Author Response · Authors · 2024-12-04
> > > >
> > > > Thank you for your reply. We appreciate the increase in your score.
> > > >
> > > > 	The code in [3] uses L2 regularisation in the CD loss, even though it seems to not be stated in the paper
> > > > Thank you for clarifying that.
> > > >
> > > > 	So in my opinion there are interesting connections between the two stabilisation methods.
> > > > We understand the connection you are trying to point out and agree that these are not completely unrelated, but we want to clarify a few things. However, these might be too specific and difficult to grasp for anybody unfamiliar with our work and [3], so we limit the visibility and provide it as a separate answer.

---

### Official Review · Reviewer_v2cL · 2024-11-04

**Soundness:** 2
**Presentation:** 3
**Contribution:** 2
**Rating:** 3
**Confidence:** 3

**Summary:**

The paper presented a novel technique for training Energy-based models (EBMs) to stabilize the EBM training provide an accurate estimate of the likelihood and generate good-quality samples. The proposed approach involves generalizing the standard EBM loss function by adding an inverse temperature parameter taking values between 0 and 1 for regularizing the learned distribution of the negative samples. The paper presented experiments to show that this modification has resulted in stabilized training of EBMs in real and simulated datasets.

**Strengths:**

The paper is presented clearly, with the authors offering essential background to understand their method. They provide an intuitive explanation that is easy to grasp. The idea that the negative samples can be OOD and correct the loss in this scenario using importance score seems novel. The authors included experiments demonstrating the method's effectiveness, along with ablation studies to highlight its key components.

**Weaknesses:**

1.	The main weakness of the paper is the lack of a competitive method. The experiment section presents an ablation study regarding the effect of the inverse temperature parameter. A key competitor of this approach can be Diffusion models which have shown to be highly accurate for likelihood estimation (look at “Variation Diffusion Models” by Kingma et al. 2023).
2.	The presentation of the experiment section needs improvement. What is the necessity of section 4.2? It seems to highlight issues in training the proposed approach on CIFAR-10 data. Then the authors change their framework to a Joint Energy-based Model (JEM) on CIFAR-10 and show that their method still only works when the temperature parameter is very small. Even with stabilized JEM, the approach seems to be inferior to diffusion models on CIFAR-10 (see FID scores in Kingma et al. 2023).
3.	The key argument for opting for EBMs instead of diffusion models is the former’s ability to estimate likelihood. However, there is no result regarding the accuracy of likelihood estimation (except for the visual representation in Fig 2). The authors are encouraged to include quantitative NLL estimates for their EBM and compare them to diffusion models (Table 1 in “Improved Denoising Diffusion Probabilistic Models” Nichol and Dhariwal 2021).

**Questions:**

1. Denoting observations as "x" in the abstract is not required.
2. The contribution section in the introduction needs improvement. The authors are encouraged to use bullet points to communicate the key contributions.
3. What does the solid line in Fig 1 represent? Is it $p_d$?
4. Sec 3.3 can go into the appendix.

---

> ### Author Response · Authors · 2024-11-27
> **Reply to Reviewer v2cL**
>
> 1. The main weakness of the paper is the lack of a competitive method. The experiment section presents an ablation study regarding the effect of the inverse temperature parameter. A key competitor of this approach can be Diffusion models which have shown to be highly accurate for likelihood estimation (look at “Variation Diffusion Models” by Kingma et al. 2023).
>    - The goal of the experimental section is not to build the best model but to evaluate the behavior under different settings ($\beta$) of the newly proposed loss. Therefore, we do not consider it to be an ablation study. We address the performance in the [Competitive method] section above.
>    - We believe EBMs are important models, and the development should not be abandoned because diffusion models currently exhibit better performance. We must perform many iterations to obtain approximate likelihood estimation for Variation Diffusion Models instead of a single forward propagation required to evaluate unnormalized likelihood for EBMs. The difference between the cost might be essential in some applications. Moreover, we do not try to solve some specific task for which we should justify the choice of a particular method. Because of that, we find little relevance for this discussion in the context of our work, which directly addresses EBM training.
>
> 2. The presentation of the experiment section needs improvement. What is the necessity of section 4.2? It seems to highlight issues in training the proposed approach on CIFAR-10 data. Then the authors change their framework to a Joint Energy-based Model (JEM) on CIFAR-10 and show that their method still only works when the temperature parameter is very small. Even with stabilized JEM, the approach seems to be inferior to diffusion models on CIFAR-10 (see FID scores in Kingma et al. 2023).
>    - We address the importance of Section 4.2 in [Section 4.2]. We disagree that our method "does not work" with a larger $\beta$. Our method is meant to deal with inaccurate samples, and it correctly de-weights their contributions. Unfortunately, the sampler used in that work consistently fails to provide any genuine sample. The sample quality improves with decreasing $\beta$, but the stability and learned density worsen. The result of the experiment suggests that it is not possible to have both a good sampler and, at the same time, respect the likelihood in that particular setup. However, we believe the performance should improve when incorporating a better sampler of negative examples. The argument with diffusion models seems unrelated to the focus of this paper, as discussed in [Competitive method].
>
> 3. The key argument for opting for EBMs instead of diffusion models is the former’s ability to estimate likelihood. However, there is no result regarding the accuracy of likelihood estimation (except for the visual representation in Fig 2). The authors are encouraged to include quantitative NLL estimates for their EBM and compare them to diffusion models (Table 1 in “Improved Denoising Diffusion Probabilistic Models” Nichol and Dhariwal 2021).
>    - Again, we can only repeat the argument given before, the goal of this work is to propose an alternative for EBM training loss, so we consider the discussion "EBM vs. diffusion models" to be inappropriate. We already commented on the amount of compute need for the evaluation of (unnormalized) log-likelihoods for EBM and diffusion models. Moreover, as EBM provide only unnormalized log-likelihoods, we are not aware of any approach that evaluates NLL and it is not used in works that we experimented with and at the same time, no EBM NLL is reported in the suggested table. Can you suggest us the method, which is applicable to models/setups that we experimented with?
>
>
> Questions:
>
> 2. The contribution section in the introduction needs improvement. The authors are encouraged to use bullet points to communicate the key contributions.
>    - Thanks for these suggestions. We can reflect them in the updated version of the paper. We further list our contribution in [Contribution].
>
> 3. What does the solid line in Fig 1 represent?
>    - It is $p_{\theta}(x)$, as denoted on the vertical axis of these plots.
>
> 4. Sec 3.3 can go into the appendix.
>    - Thank you, we address that in [Section 3.3].

---

> > ### Comment · Reviewer_v2cL · 2024-11-27
> > **Comparison with diffusion models**
> >
> > I disagree with the authors that comparison with the diffusion models is outside the scope of this study. The paper "How to Train Your Energy-Based Models" (which the authors cited) shows the connection between the score-matching objective in diffusion models and the likelihood estimation in energy-based models. I also didn't understand the authors' comment "We must perform many iterations to obtain approximate likelihood estimation for Variation Diffusion Models instead of a single forward propagation required to evaluate unnormalized likelihood for EBMs." Are the authors claiming that EBMs can accurately estimate (unnormalized) likelihood in a single forward pass? This seems unlikely as far as I understood.

---

> > > ### Author Response · Authors · 2024-11-27
> > > **Comparison with diffusion models**
> > >
> > > Thank you for your fast reply. Please let us clarify our statements further.
> > >
> > > We cannot agree more with the statement that EBMs, diffusion models, score-based, and other generative models are undoubtedly all related. The suggested comparison is essential to assess the system's performance in works focusing on achieving the best result (building the best system). However, our work falls outside that category. Our work introduces generalized loss for EBM MLE training using negative examples. As the introduced loss has one hyperparameter, whose specific setting ($\beta = 0$) corresponds to the current training approach, we aim to analyze and examine the influence of different values of $\beta$ and how it can address existing issues in EBM training. For that purpose, we chose particular models (we already provided motivation for choosing such models in our previous reply). What is essential for our work are the relative changes as $\beta$ changes. We honestly do not understand how the comparison across models helps determine that, as it would only provide insight into how good that particular model we chose is.
> > >
> > > As stated before, the discussion of EBM vs diffusion models is utterly unrelated to our work, but we would like to answer to clarify that. We claim that the unnormalized (relative) likelihood of $x$ can be exactly (not approximately) evaluated in a single forward pass of $f_{\theta}(\cdot)$ as $exp(f_{\theta}(x))$, which is by the definition of the EBM. This evaluates the unnormalized likelihood under $p_{\theta}(x)$. One of our claims in our work is that when using the standard MLE training $\beta=0$ with an inaccurate sampler of negative examples, then $p_{\theta}(x)$ will be far from $p_{data}(x)$. We demonstrate how proposed training with larger $\beta$ can narrow this gap on toy data.

---

> ### Comment · Reviewer_v2cL · 2024-12-02
> **Further Comparison Needed**
>
> I understand the authors' point of view and their contribution better. In my view, the idea proposed in the paper is promising but not novel enough (as also pointed out by reviewer J6iF) to change my evaluation. In addition, I sincerely disagree with the authors' view that the experiments comparing this method with the diffusion model or other SOTA methods (as pointed out by reviewer J6iF) are irrelevant. Therefore, I will leave it to the ACs who are more senior to evaluate the contribution of this work.

---

> > ### Author Response · Authors · 2024-12-04
> >
> > Thank you for your reply; we are sorry you didn't take into account our previous reply, where we explained that our method lies in the generalization of the objective function and that we only chose existing setups (in their default settings) to examine the effect of the proposed objective hyperparameter $\beta$ (as $\beta=0$ corresponds to the standard MLE training). You don't provide any argument as to why the comparison across models would be beneficial for our method, as it would mainly reflect the performance of the chosen model. We believe that the comment about the novelty of reviewer J6iF comes from a misunderstanding, as we explained in the reply to his review.

---

### Official Review · Reviewer_J6iF · 2024-11-04

**Soundness:** 2
**Presentation:** 1
**Contribution:** 1
**Rating:** 3
**Confidence:** 5

**Summary:**

This work presents a variation of EBM learning based on importance sampling. Negative samples are drawn from the current EBM at slightly higher temperature / flatter potential determined by a parameter $\beta \in [0, 1]$ where $\beta=0$ is standard EBM training, then reweighted to obtain an approximation of the expectation of the potential gradient with respect to the model distribution at each step. This is meant to reduce the influence of biased negative samples with especially low likelihood values that result from MCMC sampling, which can lead to unstable training. Experiments on toy datasets, and unconditional modeling/JEM modeling on CIFAR-10 investigate the proposed method.

**Strengths:**

* Sampling with a slightly higher temperature energy surface and reweighting during learning to reduce the instability of negative samples is an interesting idea which seems to provide some stability benefits. The math behind the reweighting method is sound.

**Weaknesses:**

* The experimental evaluation is very limited. There is essentially no quantitative comparison to prior works except for Table 1, which compares to the original JEM paper. There have since been several works revisiting JEM to improve stability and performance which should be used for comparison. There is no comparison to a wide variety of recent EBM works that explore unconditional CIFAR-10 modeling with significantly stronger FID scores than the ones presented in this work. Overall, the proposed method is not validated against relevant SOTA results.
* The proposed reweighting is fairly straightforward. Without strong experimental results, the limited technical innovation might not be a strong enough contribution.
* Sections 3.1 through 3.3 seem somewhat tangential and it is not clear whether the inclusion of positive samples among negative samples is ablated or used in the experimental section.

**Questions:**

* How does the proposed method compare relative to SOTA EBM methods for CIFAR-10 generation and for SOTA models in the JEM family?
* Can the importance of Section 3.1 be validated in an ablation study?

---

> ### Author Response · Authors · 2024-11-27
> **Reply to Reviewer J6iF**
>
> 1. The experimental evaluation is very limited. There is essentially no quantitative comparison to prior works except for Table 1, which compares to the original JEM paper.
>    - We incorporate the following evaluation:
>      - Stability and density on the 2D dataset - Similarly to other works, we focus on 2D toy datasets, where the learned densities can be compared visually, arguably the most reliable way of evaluating the learned density. As the difference is evident, we consider any quantitative evaluation unnecessary. We additionally reported training instability in Figure 6 that occurred in the standard MLE ($\beta=0$).
>      - Stability on EBM - as reported, we did not experience any training instabilities
>      - Learned density on EBM - More details in [Section 4.2] above
>      - Stability on JEM - The standard MLE typically diverges around or before epoch 50 (Figs. 21-25). The best performance is reported in Table 2. We believe reported behavior as the training progresses is more informative than a quantitative measure. We further elaborate on this in the [Stability evaluation] section above.
>
> 2. There have since been several works revisiting JEM to improve stability and performance which should be used for comparison. There is no comparison to a wide variety of recent EBM works that explore unconditional CIFAR-10 modeling with significantly stronger FID scores than the ones presented in this work. Overall, the proposed method is not validated against relevant SOTA results.
>    - We address the evaluation in the [Stability evaluation] section above and reached performance in [Competitive method].
>
> 3. The proposed reweighting is fairly straightforward. Without strong experimental results, the limited technical innovation might not be a strong enough contribution.
>    - We believe that our technique being straightforward is not a disadvantage as long as it is based on a novel idea. We address our contribution in [Contribution].
>
> 4. Sections 3.1 through 3.3 seem somewhat tangential and it is not clear whether the inclusion of positive samples among negative samples is ablated or used in the experimental section.
>    - On the contrary, we consider the inclusion of positive examples to be one of the key contributions of our work. It is necessary for EBM training stabilization, which we believe to be one of the most important results of this work. It is not tangential to SNIS described in Section 3.0; in fact, the application of SNIS enables the inclusion of positive examples, as explained in Sections 3.1 and 3.2. We clearly state that we use this technique, and we also ablate it, see L385: "Furthermore, in Appendix K.2, we compare the performance of different variants related to how positive examples are incorporated into negative ones. The results suggest that our default variant, corresponding to Equation 8, should be preferred over alternatives utilizing Equation 7, Equation 33, and Equation 34. Consequently, in the rest of this work, we will consider only the default variant."
>
> Questions:
>  - Q: How does the proposed method compare relative to SOTA EBM methods for CIFAR-10 generation and for SOTA models in the JEM family?
>     - A: If you consider the JEM family to be models that have both generative and discriminative capabilities, we believe that [1] and [2] reported the best FID/IS/accuracy. Our method could be applied to [2] since it incorporates the standard loss for EBM training. However, as stated in [Competitive method], this work does not aim to build/compete with the best-performing system but to evaluate the effect of different $\beta$ values.
>  - Q: Can the importance of Section 3.1 be validated in an ablation study?
>     - A: It is done in Appendix K.2 (Figs. 7, 8, 10, and 11); see L385 in the main text.
>
> [1] Guo, Qiushan, et al. "EGC: Image Generation and Classification via a Diffusion Energy-Based Model." Proceedings of the IEEE/CVF International Conference on Computer Vision. 2023.
>
> [2] Yang, Xiulong, Qing Su, and Shihao Ji. "Towards Bridging the Performance Gaps of Joint Energy-based Models." Proceedings of the IEEE/CVF Conference on Computer Vision and Pattern Recognition. 2023.

---

### Author Response · Authors · 2024-11-27
**Response to all reviewers 1/2**

We thank all reviewers for their critical assessment of our work. We uploaded an updated version of the paper, where we improved English on a sentence level and corrected typos ($\beta = 2.5 × 10^{−6}$ -> $\beta= 2.5 × 10^{−5}$). Additionally, we refactored Appendix M to introduce experiments more straightforwardly. We want to address some aspects of our work that are relevant to multiple reviews.


[Competitive method] The reviewers claim we do not present a competitive method for achieving state-of-the-art performance. We proposed a generalization of the standard MLE objective for robust training in cases when some generated examples have low likelihood values. The basic existing setups are known to suffer from these issues, which is our primary motivation to experiment with them. More advanced approaches typically focus on a better choice of initial distribution for the MCMC chain (for example, by using normalizing flows or VAEs), modeling distribution in lower-dimensional latent space, introducing additional regularizers, and improving or enlarging NN architecture. We propose to consider generalized loss (i.e., different $\beta$ values). We want to stress that most of these improvements are compatible with the proposed generalized loss. Nevertheless, some reviewers negatively evaluated our choice of setup due to the gap between the performance of the considered setup and the state-of-the-art performance. Because of that, we were asked to compare the performance across different setups, which we believe to be irrelevant due to the provided arguments.

[Section 4.2] The reviewers consider the experiments performed in Section 4.2 a failure. However, we obtained results aligned with the paper's claims. Increasing $\beta$ mitigates the influence of model parameter updates that decrease the log-likelihood. We do not report log-likelihood for EBM as we cannot estimate its normalized value, but Figure 14 suggests that models trained with larger $\beta$ result in better likelihoods. A surprising finding is that the influence of using a different $\beta$ is considerable even for very small $\beta$ values, which suggests that unnormalized likelihoods for models trained with $\beta=0$ might be completely uninformative. At the same time, it also demonstrates the significance of the proposed loss. As we observe the same behavior during JEM training, we consider it essential to discuss it.

[Section 3.3] - The reviewers question the importance of the Section 3.3. This section aims to demonstrate how simple it is to incorporate the proposed generalized loss, but we agree with the reviewers and will move this section to the appendix.

[Related literature] - The reviewers suggest a broader discussion of related literature. We agree that our work would benefit from the discussion about EBM extensions and possibly JEM extensions. These aim to reach better performance, which sometimes also improves training stability. We will include this section in our work. We want to thank reviewers for pointing out some additional recent works we were unaware of.

---

> ### Author Response · Authors · 2024-11-27
> **Response to all reviewers 2/2**
>
> [Stability evaluation] The reviewers complain about the extent of our evaluation. One of the main objectives of our work is to improve the stability of the training. Therefore, we experiment with JEM, notoriously known for its training instabilities. The authors of JEM did not find a setting that would complete without experiencing the divergence and retaining the performance. We demonstrate that we can successfully train the model not just with their default settings but also in more resource-restricted settings, for which the training diverges much faster using the standard MLE training. We demonstrate these in Figs. 21-25, and although we do not report training stability via some quantitative measure, the improvement in stability is evident. We believe that this should be sufficient to demonstrate improved stability. Based on our experience, EBM training instabilities are most severe during development when new architecture, modality, or hyperparameters are explored. We hope our method can address that and allow a much broader range of combinations to successfully finish the training, even if it slightly reduces the performance.
>
>
> [Contribution]  - The reviewers raised concerns about our contribution and would appreciate a section on contribution as part of the introduction. The following is the summary of our contribution:
> 1. We propose a novel generalized loss for training EBM consisting of 2 parts:
>     1. Applying SNIS that introduces hyperparameter $\beta$ (set to 0 in the standard MLE optimization).
>     2. Including positive example into negative ones to improve stabilization when $\beta \ne 0$.
> 2. We provide theoretical motivation as to why increasing $\beta$ trades off the quality of samples produced by a biased sampler for increased stability and more credible density.
>     1. We empirically verify improvements of learned density on 2D toy datasets in various conditions.
>     2. We empirically verify stability improvements on JEM trained on CIFAR-10 under various hyperparameter settings.
> 3. We prove that optimizing 1.2 leads to maximizing the lower bound of the original objective associated with 1.1.
> 4. Variants of 1.2 having similar stability properties are proposed. We show that the objective associated with 1.2 helps to learn the most credible densities on 2D toy data.
> 5. We analyze the influence of 1.2 compared to the standard MLE training by isolating three different effects:
>     1. Weights are associated with each negative example, effectively de-weighting contribution from negative examples with low likelihoods, preventing an unconstrained decrease of the likelihood for negative examples. These weights are already a consequence of 1.1.
>     2. The learning rate is adjusted for each mini-batch based on the difference between the likelihoods of negative and positive examples. As a consequence, the importance of parameter updates decreases when using inaccurate negative examples.
>     3. Weights are associated with each positive example. These weights prevent an unconstrained growth of the likelihood values of positive examples. The difference between the likelihood of positive and negative examples influences the size of this effect.
> 6. We show that 1.2 applied to the standard MLE loss ($\beta=0$) only affects the global learning rate with no other effect.
> 7. We demonstrate the similarity between 1.2 and the discriminative training, which allows straightforward implementation using libraries supporting automatic differentiation.

---

### Meta-Review · Area_Chair_bpbY · 2024-12-16

**Metareview:**

This paper introduces a new method for training energy-based models aimed at increasing the stability of training this class of model. The method draws samples from a higher-temperature version of the model distribution then corrects the tempered sampling with importance sampling.

The paper is well written and the reviewers thought method was interesting. Overall though, the reviewers were concerned about the paper's lack of comparison to more recent works in the EBM literature and thought the baseline methods used for comparison are relatively out of date.

Overall I agree with the reviewers that the experiments presented do not convince me of the utility of the method. The experimental scope was limited and the authors do not compare with method with other methods meant to improve upon standard SGLD sampling for EBM training. While I agree that the point of the work is to understand how the method improves upon standard SGLD comparing with other (potentially orthogonal methods) should be done to help contextualize the benefit provided by the proposed method.

**Additional Comments On Reviewer Discussion:**

Initially most reviewers had concerns about the papers limited experiments, lack of comparisons to recent methods, weak related work section, and questions about the method's theoretical foundations. Reviewers and authors went back and forth on these points and reviewers were left still wanted additional comparisons. Overall the rebuttal process did not change the reviewers' overall sentiment about the work.

---

### Decision · Program_Chairs · 2025-01-22

Reject